# Macrophage-induced enteric neurodegeneration leads to motility impairment during gut inflammation

Mona Breßer[1,8], Kevin D Siemens[1,8], Linda Schneider[1], Jonah E Lunnebach[1], Patrick Leven [1],
Tim R Glowka [1], Kristin Oberländer[2,3], Elena De Domenico [4], Joachim L Schultze[4,5,6],
Joachim Schmidt[7], Jörg C Kalff[1], Anja Schneider[2,3], Sven Wehner [1] & Reiner Schneider [1]✉

## Abstract

Current studies pictured the enteric nervous system and macrophages as modulators of neuroimmune processes in the inflamed gut. Expanding this view, we investigated the impact of enteric neuron–macrophage interactions on postoperative trauma and subsequent motility disturbances, i.e., postoperative ileus. In the early postsurgical phase, we detected strong neuronal activation, followed by transcriptional and translational signatures indicating neuronal death and synaptic damage. Simultaneously, our study revealed neurodegenerative profiles in macrophage-specific transcriptomes after postoperative trauma. Validating the role of resident and monocyte-derived macrophages, we depleted macrophages by CSF-1R-antibodies and used CCR2$^{-/-}$ mice, known for reduced monocyte infiltration, in POI studies. Only CSF-1R-antibody-treated animals showed decreased neuronal death and lessened synaptic decay, emphasizing the significance of resident macrophages. In human gut samples taken early and late during abdominal surgery, we substantiated the mouse model data and found reactive and apoptotic neurons and dysregulation in synaptic genes, indicating a species' overarching mechanism. Our study demonstrates that surgical trauma activates enteric neurons and induces neurodegeneration, mediated by resident macrophages, introducing neuroprotection as an option for faster recovery after surgery.

**Keywords** Enteric Neurons; Neuroimmune Interaction; Postoperative Ileus; Synaptic Damage; Macrophages
**Subject Category** Neuroscience

## Introduction

The enteric nervous system (ENS), consisting of enteric neurons and enteric glia, is a branch of the autonomous nervous system that governs various functions throughout the alimentary tract, such as gastric motility, epithelial barrier, and intestinal homeostasis, appointing the ENS also as a key player in disease pathogenesis (Sharkey and Mawe, 2023). Intestinal pathological processes are often expedited by acute or chronic inflammation, leading to enteric neuropathies (Lakhan and Kirchgessner, 2010; Holland et al, 2021). Neuropathies and neuroinflammation with the resulting neurodegeneration are extensively discussed in numerous CNS disease states (Chitnis and Weiner, 2017; Zhang et al, 2023b) with an increasing focus on the neuronal phenotype (Zhang et al, 2023b). Although multiple forms of intestinal pathologies are known to impair ENS functions (Niesler et al, 2021), detailed investigations focusing particularly on the role or response of enteric neurons in inflammatory mechanisms are still lacking (Margolis and Gershon, 2016).

A prominent example of an acute neuroinflammatory condition in the gut is postoperative ileus (POI). POI is a frequent transient GI-motility impairment and a complication of abdominal surgery. Patients with POI suffer from, e.g., nausea, abdominal distension pain, and reduced food tolerance, leading to an extended hospitalization phase and a strong medico-economic burden for healthcare systems (Sommer et al, 2021). Early studies by our group and others primarily defined the role of immune cells in POI and identified resident macrophages (Wehner et al, 2007; Wehner et al, 2009) and infiltrating monocytes (Farro et al, 2017) as key players in the relevant immune processes (Boeckxstaens and de Jonge 2009; Buscail and Deraison, 2022; Sui et al, 2022; Stein et al, 2018). Recent advances expanded this view by investigating the role of enteric glial cells, a neurosupportive cell type of the ENS, revealing their impact on disease development (Linan-Rico et al, 2023) and their essential part in the inflammatory response of the gut (Schneider et al, 2020; Stakenborg et al, 2022; Schneider et al, 2022; Leven et al, 2023; Progatzky et al, 2021).

As part of the ENS, enteric glia intertangle closely with enteric neurons, and both lie in close proximity to resident macrophages. Similar to enteric glia, enteric neurons can be triggered by numerous mediators of an inflamed gut (Margolis and Gershon, 2016) and react to cytokines (Wang et al, 2022), neurotransmitters (Delvalle et al, 2018), and growth factors (Soret et al, 2020). However, besides the observation of disturbed postoperative

[1]Department of Surgery, University Hospital Bonn, Bonn, Germany. [2]German Center for Neurodegenerative Diseases (DZNE), Bonn, Germany. [3]University of Bonn Medical Center, Dept. of Neurodegenerative Disease and Geriatric Psychiatry/Psychiatry, 53127 Bonn, Germany. [4]Deutsches Zentrum für Neurodegenerative Erkrankungen (DZNE). PRECISE Platform for Genomics and Epigenomics at DZNE and University of Bonn, Bonn, Germany. [5]Systems Medicine, Deutsches Zentrum für Neurodegenerative Erkrankungen (DZNE), Bonn, Germany. [6]Genomics and Immunoregulation, Life & Medical Sciences (LIMES) Institute, University of Bonn, Bonn, Germany. [7]Department of General, Thoracic and Vascular Surgery, University Hospital Bonn, Bonn, Germany. [8]These authors contributed equally: Mona Breßer, Kevin D Siemens. ✉E-mail: Reiner.Schneider@ukbonn.de

motility, less is known about the response of enteric neurons to surgical trauma and the mechanisms affecting them in this inflammatory environment.

Herein, we aimed to investigate the molecular and cellular changes in enteric neurons during POI, as well as the impact of macrophages on neuronal changes. By reanalyzing our published transcription datasets, we focused on neuronal activity and function and discovered a CNS-like neurodegenerative phenotype with synaptic decay and neuronal death, which was confirmed by comprehensive transcriptional and translational analysis, together with immunohistochemical quantifications. Furthermore, we applied $ChAT^{Cre/+}/Rpl22^{HA/+}$ mice to extract cell-specific mRNA from hemagglutinin-labeled ribosomes of enteric neurons (Leven et al, 2021) within POI development and found evidence of strong enteric neuronal activity in the immediate postoperative phase with dysregulation in synaptic signaling. At a later stage, detrimental neuronal conditions with upregulation of apoptosis programs and downregulation of cellular homeostasis and synaptic structures were observed. Macrophage-depletion studies and the usage of CCR2-knockout mice revealed that resident inflammatory macrophages are imperative for initiating these neurodegenerative processes and that their depletion reduces neuronal decay, synaptic damage, and POI symptoms. Validating our findings in the clinical situation, we learned that similar mechanisms are also induced in patients' gut samples after abdominal surgery.

## Results

### Intestinal manipulation and inflammation activate enteric neurons

Neuroinflammation in the gut involves and is regulated by various cell types (Mazzotta et al, 2020). Therein, enteric neurons in the *muscularis externa* (*ME*) are initially affected by proximal reactive enteric glia (Leven et al, 2023) and inflammatory macrophages (Wehner et al, 2007), driving them towards an activated cellular state further aggravated by infiltrating leukocytes (Fig. 1A). To understand the pathogenesis of surgery-associated GI dysfunction, we applied a mouse model of postoperative ileus to study the impact of intestinal manipulation during surgery on myenteric neurons. Induction of POI can be accomplished in a standardized manner using surgical trauma by intestinal manipulation (IM) to trigger acute intestinal neuroinflammation, strong CD45+ immune cell infiltration (Appendix Fig. S1a,b), and neuronal dysfunction resulting in gastrointestinal motility impairment in the small intestine (Appendix Fig. S1c). Our analysis focused on two time points, 3 h, and 24 h after the intestinal manipulation, generating data from the disorder onset and manifestation, respectively (Fig. 1B). We evaluated the first neuronal reactions after IM by immunofluorescence for cFOS, a classical early neuronal activation marker (Lara Aparicio et al, 2022), and discovered a strong increase in cFOS+ neurons (ANNA1+ cells) within myenteric ganglia (Fig. 1C), which we also visualized in an overview image (Appendix Fig. S1d). Quantification of c-FOS+/ANNA1+ cells in the early postoperative phase (IM3h) revealed activation of more than 50% of enteric neurons, whereas, at IM24h, almost no double-positive cells were detected (Figs. 1C and EV1A). Next, we investigated the transcriptional reaction of the *ME* by bulk RNA-Seq. As expected,

after surgical trauma, a principal component analysis (PCA) showed a clear clustering of control (naive animals), IM3h, and IM24 groups (Fig. EV1B). Within the list of differentially expressed genes (DEGs) that are more than fivefold (IM3h) and more than tenfold (IM24h) induced, we detected well-known genes associated with enteric neuronal function highly dysregulated in both stages (Fig. 1D,E). These included, among others, markers for cellular activation (IM3h: *Fos*, *Arc*, *Bdnf*), neuronal destruction (IM24h: *Tubb4a*, *Evol4*), and cell death (IM24h: *Bax*, *Bcl3*). To understand these neuronal changes, we performed gene ontology (GO) analysis of DEGs in mice undergoing surgery at specific time points compared to their control (naive) mice and detected strong enrichment of gene clusters connected to "neuron death", "synapses", and "neurogenesis" for both time points (Fig. 1F). Interestingly, these neuronal gene clusters were similarly enriched as classical POI hallmarks "inflammation" and "immune cell migration" (Fig. 1F). Hierarchical clustering of these neuron-related GO terms showed a precise patterning for control, IM3h, and IM24h samples. Major fluctuations within these clusters indicate an immense gene expression switch in neurons during POI progression (Fig. 1G). Distinct gene expression patterns after trauma also recur in heatmaps connected to the POI hallmark gene clusters "inflammatory response" and "migration" (Fig. EV1C). Intrigued by the transcriptomic changes related to "synapses", we further validated specific genes of this GO term in previously generated RNA-Seq datasets from POI mice (Schneider et al, 2020; Schneider et al, 2022; Leven et al, 2023; Mallesh et al, 2021). The gene expression of synapse proteins (*Vcan*, *Syt12*, and *Shank3*) and synaptic function proteins (*Lyn*, *Arc*, *Vgf*, *Bdnf*, *Shc4*, and *Rtn4*) were found dysregulated during POI, mainly at IM3h (Fig. EV1D,E). However, genes highly expressed in control animals for basic synaptic transmission (*Nrxn1*, *Lynx1*, *Gria1*, *Cplx2*, *Chrm1*, and *Oprk1*) were strongly downregulated in POI mice at both stages (Fig. EV1F). Subsequent mass spectrometry analysis confirmed the significant regulation of genes related to neuron death and intestinal neuroinflammation on protein level. The PCA plot showed clear differences between controls and POI mice (Fig. EV1G). Volcano plots (Fig. 1H) and a heatmap for the most regulated proteins (Appendix Fig. S1e) further confirmed distinct dysregulation at the peak of the disease, showing protein regulation of synapses (SYN1 + 2. SNAP25, SYNM), macrophage activation (CD63, ARG1, MPO, YM1), cell death (BCLAF1) and inflammation (S110A8 + A9, HMOX1, SERPINE1, PTGS1, SPARCL1). Using the significantly up- and downregulated proteins for pathway enrichment analysis, we detected enrichment in cascades related to inflammatory signaling and migration, as expected in POI (Fig. EV1H), and neuronal functions during POI (Fig. 1I), emphasizing the profound alteration in the cellular state we have observed on the transcriptomic level.

In this first experimental set, we discovered that enteric neurons become activated in the early phase of POI and that genes connected to enteric neuron function are severely dysregulated throughout the disease course.

### Intestinal manipulation and inflammation induce enteric neurodegeneration

Following up on the detrimental enteric neuronal phenotype in POI, we explored synaptic function by immunofluorescence and

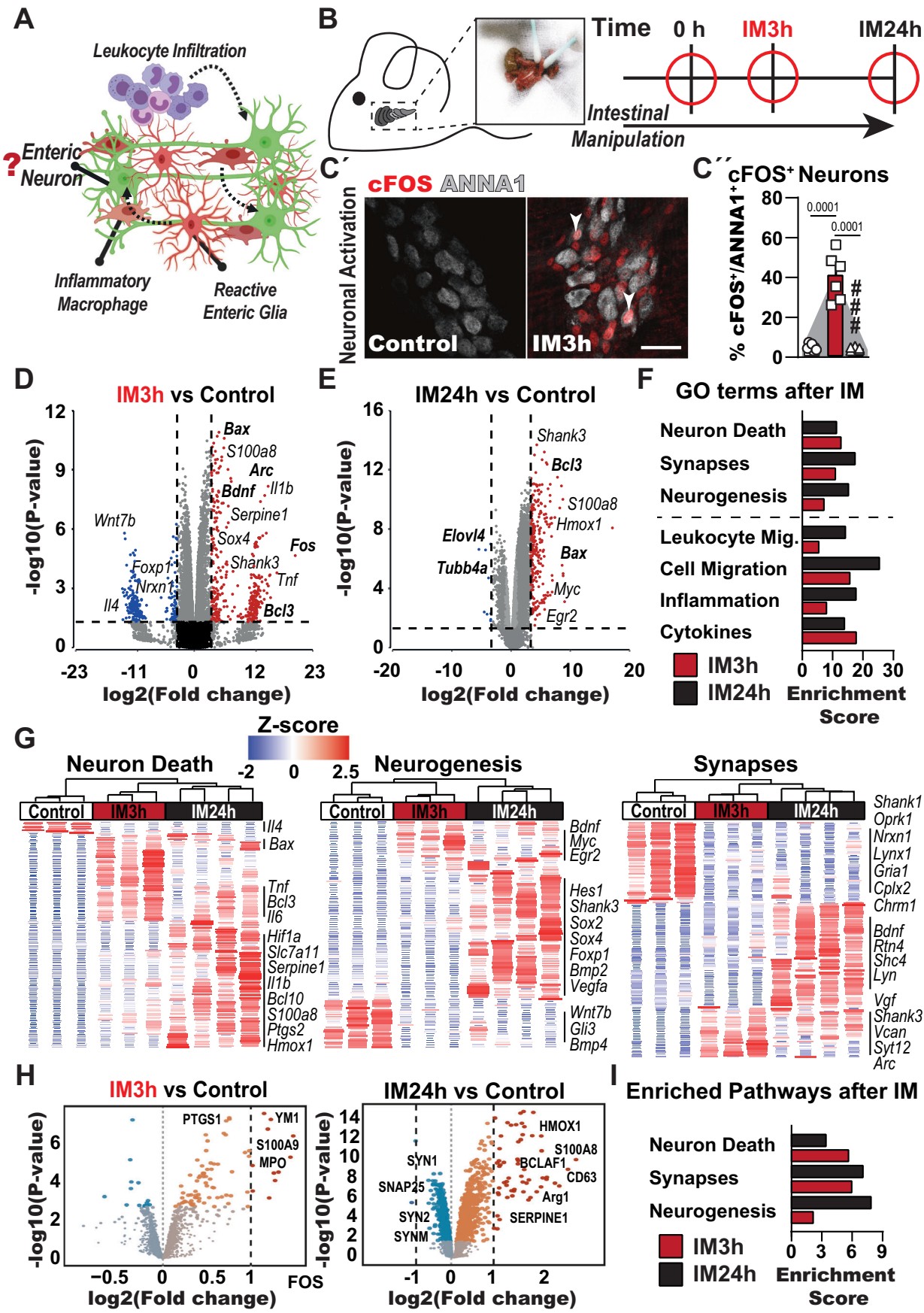

◄ **Figure 1. Intestinal manipulation and inflammation activate myenteric neurons.**

(A) Schematic overview of cell types involved in the inflammatory reactions during POI. (B) Overview of analysis time points in the POI animal model after intestinal manipulation (IM). Acute activation of resident cells in the small intestine *muscularis externa (ME)* is initiated at IM3h. At IM24h, the disease peak, an immune cell infiltrate (Appendix Fig. S1a,b) is present in the *ME* and, in concert with the ENS, evokes typical symptoms of POI. (C′) Immunofluorescence analysis of control and IM3h jejunum whole mounts for myenteric neurons (ANNA1+, gray) and an activation marker (cFOS+, red). At IM3h, a population of double-positive cells (white arrowheads) was detected. Scale bar 50 µm. (C′′) Quantification of cFOS+/ANNA1+ cells after IM. Bar graphs show the mean % of double-positive cells normalized to the total number of ANNA1+ cells. n = 7 (control), 6 (IM3h), 5 (IM24h). (D–I) 3′ Bulk RNA-Seq analysis of *ME* samples isolated from IM and control mice at the indicated time points. (D) Volcano plot showing significantly changed genes between the IM3h and control group. The plot depicts 384 up- and 260 downregulated genes with a fold change ≥5 and P < 0.05. (E) Volcano plot shows significantly changed genes between the IM24h and control group. The plot depicts 272 up- and 9 downregulated genes with a fold change of ≥10 and P < 0.05. (F) Gene ontology (GO) analysis of significantly changed genes (P < 0.05, fold change ≥1.5) shows GO terms connected to neuronal functions and POI hallmarks. (G) Heatmaps of genes connected to neuron death, neurogenesis, and synapses in IM and control animals. (H, I) Mass spectrometry analysis of control and POI mice. (H) Volcano plot shows significantly changed proteins between the IM24h (842 up- and 567 downregulated) and IM3h (78 up- and 14 downregulated) compared to the control group. (I) Pathway analysis of significantly changed proteins in POI (P < 0.05). n = 6. Statistical analysis is based on one-way ANOVA (C) and Fisher's exact t test (D–I). Standard deviations are presented as SEM. Source data are available online for this figure.

gene expression analyses. Synapsins (Syn1-3), postsynaptic protein 95 (PSD95), and choline acetyltransferase (ChAT) are essential proteins participating in synaptic function (Rosenberg et al, 2014; Margiotta et al, 2021). Histology and quantification of synaptic proteins showed no apparent changes at IM3h but a diminished expression at IM24h (Figs. 2A and EV2A; Appendix Fig. S2a). The expression of corresponding transcripts (*Syn1-3*, *Dlg4*, and *ChAT*) was decreased accordingly (Fig. 2B), and genes involved in synaptic transmission (*Snph*, *Syp*, *Scn2a*, *Scgn*, *Kcna2*, and *Syngr1*) were robustly downregulated in POI at IM24h with a partial decrease already 3 h after IM (Fig. EV2B). Subsequent mass spectrometry analysis confirmed the downregulation of various synaptic proteins in POI mice (Fig. 2C). In addition, we performed a staining procedure commonly used to visualize CNS structures, the Golgi-cox staining (Louth et al, 2017), in the gut to image dendritic structures of control and POI mice. In line with the neurodegenerative molecular patterns, Golgi-cox stained Swiss roles showed fewer dendrites and fewer axonal structures in the gut of IM24h compared to control mice (Fig. 2C).

Next, we investigated whole mounts of the *ME* during POI for potential changes in neuronal numbers and found reduced numbers of ANNA1+ neurons 24 h after surgery (Fig. 2D). Moreover, the amount of cleaved caspase-3 (CASP3)+ apoptotic neurons doubled three hours after surgery and persisted even at IM24h compared to controls (Fig. 2E; Appendix S2b). Accordingly, *Bax* transcript and BCLAF1 protein levels, both classical markers for apoptosis (Slupe et al, 2021; Lee et al, 2012), showed a similar expression pattern (Fig. 2F,G). Both results reiterate the previously shown induction of the GO term "neuron death" during POI (Fig. 1F,K) and are in line with enriched pathways connected to neurodegeneration from our proteomics analysis (Fig. EV2D). Simultaneously, we discovered an upregulation of Ki67+ neurons at IM24h (Fig. EV2E) with an analogous increase of *mKi67* transcripts (Fig. EV2F) and Proliferating-Cell-Nuclear-Antigen (PCNA) protein levels (Fig. EV2G) in POI mice. Together, these data indicate a simultaneous induction of neuronal apoptotic cell death and proliferative programs. The latter might be a potential compensatory reaction of the ENS to neuronal deterioration. Therefore, we examined mice 21 days after POI induction (IM21d), expecting the neuronal loss to be compensated at this late time point. Similar to the previously observed levels at IM72h (Leven et al, 2023; Schneider et al, 2022), motility impairment and leukocyte infiltration had reached homeostatic levels again and did not differ

from naive animals in IM21d mice (Fig. EV2H). Strikingly, IM21d mice displayed no changes in neuronal numbers (ANNA1+, Fig. 2H) and synaptic structures (SYN1/2, Appendix Fig. S2c), suggesting recovery from enteric neurodegeneration.

As a final proof of concept, we also tested blood plasma samples using a SIMOA assay for the *Quanterix* system to determine the presence of neuronal markers indicating neurodegeneration of the ENS. Elevation of these mediators, namely neurofilament light chain (NfL), GFAP, and beta-amyloid-40 and 42 (Aβ40 + Aβ42), has been observed in CNS neurodegeneration, e.g., in Alzheimer's disease models (5xFAD mouse model (Forner et al, 2021)). Remarkably, we detected high levels of NfL in the blood plasma of mice after surgery at IM24h (Fig. 2H), but no visible changes in the levels of the other markers (Appendix Fig. S2c). Again, IM21d mice showed no elevation of these markers and did not differ from controls (Fig. 2H), supporting that ENS neurodegeneration is recoverable.

These findings introduce ENS neurodegeneration as a transient event in POI, suggesting compensation for early neuronal loss and, ultimately, a recovery from enteric neurodegeneration.

## Inflammation-triggered neuronal activation leads to enteric neurodegeneration

While the previous observations originated from full tissue *ME* specimens, we next aimed to decipher neuron-specific transcriptional programs. Therefore, we utilized a *RiboTag* approach to acquire neuron-specific transcriptomes of control and POI mice by bulk RNA-Seq (Fig. 3A, (Leven et al, 2021)). Crossing Chat^Cre/+ with Rpl22^HA/+ mice produced a transgenic line expressing the *RiboTag* in about 80% of all ganglionic enteric neurons (Fig. EV3A). PCA plot indicated a clear clustering between IM and control groups, showing sample variations mainly in the controls (Fig. 3B). First, focusing on the overall neuronal transcript levels, we detected only half of the number of transcripts by RNA-Seq analysis in the IM24h samples compared to other time points (Fig. EV3B). This transcriptional phenotype matches with the neurodegenerative phenotype described in Figs. 1 and 2. Accordingly, 95% of all DEGs in IM24h samples were significantly downregulated, including *Syn1* and *ChAT*, also depicting neurodegeneration in enteric neurons at the peak of the disease. At the same time, enriched expression of *Bax* supported the neuronal decay phenotype (Fig. EV3C). Therefore, we centered the study on the early IM3h time point, which

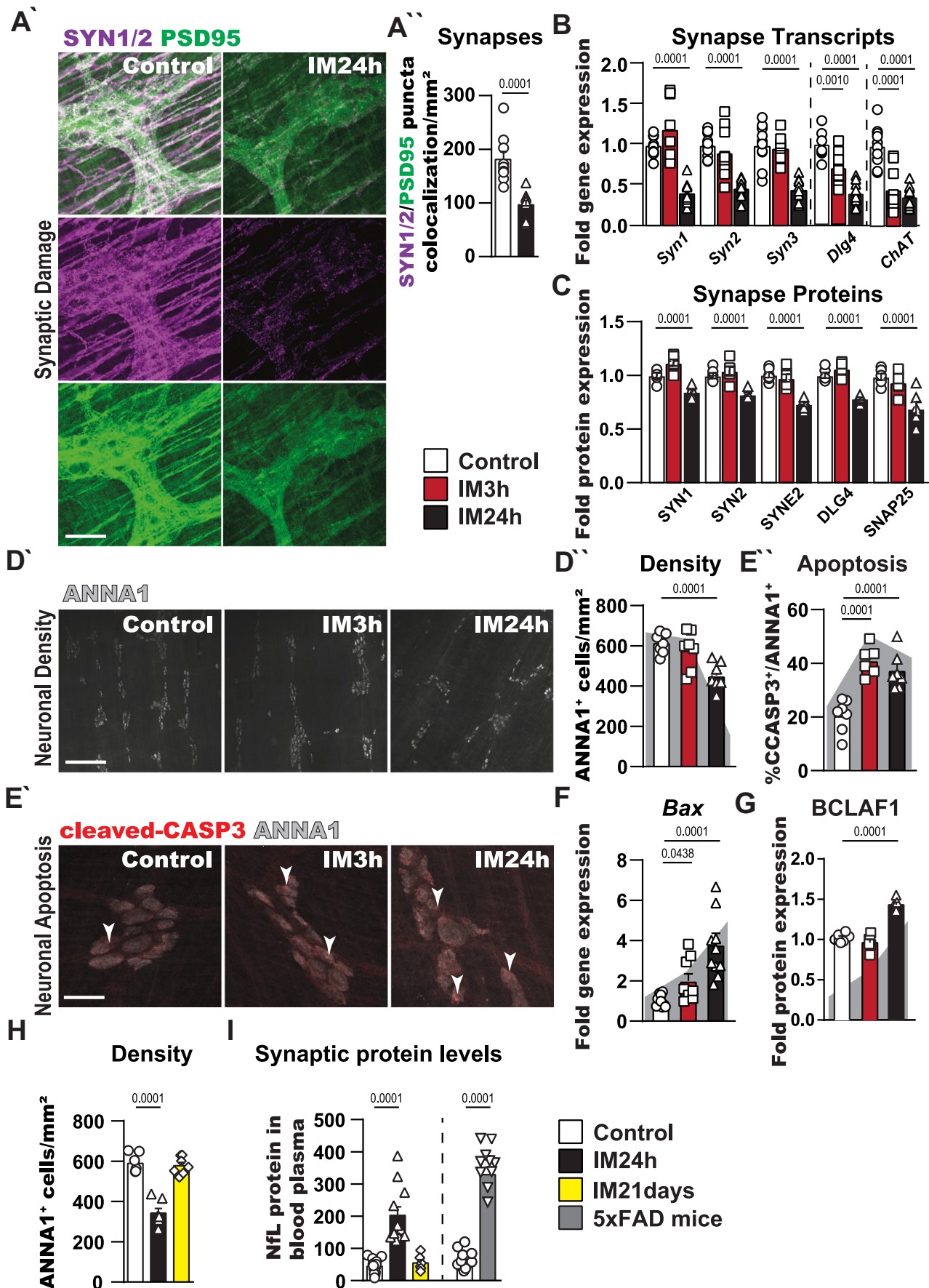

**Figure 2.  Intestinal manipulation and inflammation induce enteric neurodegeneration.**

(A′) Immunofluorescence analysis of synaptic structures (SYNAPSIN 1/2, violet), postsynaptic density protein 95 (PSD95, green) 24 h post IM and in control. Scale bar 50 μm. (A′′) Immunofluorescence quantification of synapse puncta co-localization intensity per mm$^2$ in jejunum ME tissue after IM and control. $n = 9$ (control), 10 (IM24h). (B) Gene expression analysis of synaptic structures in IM and control animals. Bar graphs show the fold gene induction normalized to control mice. $n = 10$ (control), 9 (IM3h), 10 (IM24h). (C) Protein expression analysis of synaptic structure proteins in IM and control animals. Bar graphs show the fold protein induction normalized to control mice. $n = 6$. (D′) Immunofluorescence analysis of myenteric neurons (ANNA1$^+$, gray) 3 and 24 h post IM and in control. Scale bar 50 μm. (D′′) Quantification of ANNA1$^+$ cells per mm$^2$ jejunum ME tissue after IM. Bar graphs show the mean number of ANNA1$^+$ cells normalized to the ME area. $n = 8$ (control), 7 (IM3h), 7 (IM24h), with each analyzed animal, one whole-mount jejunum specimen was used for the quantification. (E′) Immunofluorescence analysis of apoptotic (cleaved-CASP3$^+$, red) myenteric neurons (ANNA1$^+$, gray) 3 and 24 h post IM and in control. In all conditions, double-positive cells (white arrowheads) were detected. Scale bar 50 μm. (E′′) Quantification of ANNA1$^+$/cleaved-CASP3$^+$ cells after IM. Bar graphs show the mean % of double-positive cells normalized to the total number of ANNA1$^+$ cells. $n = 7$ (control), 6 (IM3h), 7 (IM24h). (F) Gene expression analysis of the apoptosis marker Bax in IM and control. Bar graphs show the fold gene induction normalized to control mice. $n = 10$ (control), 9 (IM3h), 10 (IM24h). (G) Protein expression analysis of the apoptosis marker BCLAF1 in IM and control. Bar graphs show the fold protein induction normalized to control mice. $n = 6$. (H) Quantification of ANNA1$^+$ cells per mm$^2$ jejunum ME tissue after IM24h and IM21 days. Bar graphs show the mean number of ANNA1$^+$ cells normalized to the ME area. $n = 7$ per group, with each analyzed animal, one whole-mount jejunum specimen was used for the quantification. (I) Blood plasma levels of neurofilament light chain (NfL) in POI, control, and 5xFAD (Alzheimer′s) mice. Plasma samples were analyzed with a SIMOA assay (Neurology 4-Plex E Advantage Kit) using the Quanterix system. Bar graphs show the NfL protein levels (pg/ml) in the blood plasma of all groups. $n = 13$ (control1), 15 (control2), 11 (IM24h), 6 (IM21 days). Statistical analysis is based on Student′s t test (A) and one-way ANOVA (B–I). Standard deviations are presented as SEM. Source data are available online for this figure.

revealed an activated neuronal state with more than 600 upregulated genes, including genes connected to cell death (e.g., Bax, Bcl3). In addition, genes associated with cell activation (e.g., Fos, Egr4) and neurogenesis (e.g., Sox9, Sox4) were upregulated, while genes associated with synaptic transmission (e.g., Arc, Shank1, Nrxn1) were differentially regulated, hinting at neuronal activation in the early disease state (Fig. 3C). In line, GO term analysis of DEGs showed enrichment scores connected to "neuron death", "synapse", "neurogenesis", and "proliferation" at IM3h (Fig. 3D) and IM24h (Fig. EV3D). Exploring only GO terms associated with cell death processes, we recognized the "apoptotic process" gene cluster as the one with the highest enrichment score (Fig. EV3E), further underlining the cleaved caspase-3 increase in neurons during POI. Simultaneous upregulation of the gene clusters "inflammation" and "response to cytokine" indicated a direct reaction of enteric neurons with their inflammatory environment. Subsequent heatmap clustering of neuron-related GO terms reiterated the transcriptional shift during the early POI phase with oppositely expressed gene clusters compared to controls (Fig. 3E). Further investigating synaptic signaling in neurons, we detected an early-on dysregulation of the previously evaluated synaptic genes (Downregulated: Syp, Scn2a, Kcna2, Synpo, Synj1, Oprk1, Lynx1, Nrxn1, Cplx2, Gria1, Chrm1; (Fig. 3F). Upregulated: Syngr2, Vcan, Sy12, Arc, Vgf, Rtn4, Synm, Synpo2, Chrm2; (Fig. 3G)), predicting functional changes in neuronal transmission in the inflamed ME. The expression changes of synaptic genes (Syn1-3, Dlg2, Shank1-3, and ChAT) at IM3h strengthened these observations (Fig. EV3F).

The neuron-specific transcriptome analysis confirmed POI-induced enteric neurodegeneration and suggested an impact of the inflammatory environment on enteric neuron homeostasis.

## Inflammatory macrophages are able to cause enteric neurodegeneration

In the CNS, neurodegeneration involves actions of various cell types. One prominent but non-exclusive driver are reactive microglia (Li et al, 2020; Gao et al, 2023; Bennett and Viaene, 2021), by inflicting inflammatory (Zhang et al, 2023b; Kwon and Koh, 2020) and metabolic stress (Muddapu et al, 2020; Jha et al, 2017) on neurons. As

microglia and macrophages share many functional characteristics (Verheijden et al, 2015), and the activation state of resident macrophages is crucial for the onset of POI (Wehner et al, 2007), we aimed to assess whether similar interactions occur in the ENS. Therefore, we first investigated the localization of macrophages in relation to enteric neurons during the POI. To better visualize the possible interaction of these two cell types in intestinal inflammation, we subjected a macrophage fluorescent reporter strain, the Cx3cr1$^{gfp/+}$ mice (Jung et al, 2000), to IM and quantified the number of total and ganglia-associated macrophages by immunofluorescence. At IM24h, the ganglia-associated macrophages assembling around ANNA1$^+$ enteric neurons significantly increased (Figs. 4A,B and EV4A), suggesting a possible interaction between both cell types. To further validate this neuron–macrophage interaction, we performed additional stainings for F11R, a recently published marker, labeling neuron-associated macrophages in the gut (Viola et al, 2023) and quantified these cells in control and POI mice. CX3CR1$^+$/F11R$^+$ macrophage numbers increased during POI, indicating possible inflammation-induced cell-cell interaction between enteric neurons and macrophages (Fig. 4C). Moreover, ganglion-associated and interganglionic macrophages of IM3h mice display increased levels of CD68, a marker for inflammatory activation, compared to control mice, suggesting an early macrophage activation after IM (Fig. 4D). As gut inflammation culminates in this disease stage, at IM24h, the CD68 expression peaked there (Fig. EV4B). To further define the early activation, we performed FACS analysis of resident macrophages at IM3h (gating scheme, Appendix Fig. S3a) and confirmed this early inflammatory activation by upregulation of several surface markers indicative of reactive macrophages, such as CD16/32 (Hamzei Taj et al, 2016), CD36 (Chen et al, 2019), MHCII (Buxadé et al, 2018), and CD11c (Tang et al, 2021) (Fig. 4E). Next, we aimed to elucidate the macrophage activation status in more detail by FACS-sorting of the infiltrating (Ly6C$^{high}$/CX3CR1-GFP$^+$) and the resident (Ly6C$^{low}$/CX3CR1-GFP$^+$) macrophages (gating scheme, Appendix Fig. S3b) from POI and control mice to analyze their transcriptomes by SMART-Seq2 (Fig. EV4C). The PCA plot of all three time points showed substantial differences in the mRNA expression between resident macrophages (Fig. 4F) and monocyte-derived macrophages (Fig. EV4D). Moreover, Volcano plots revealed notable transcriptional changes in resident macrophages with 2524 up- and 244

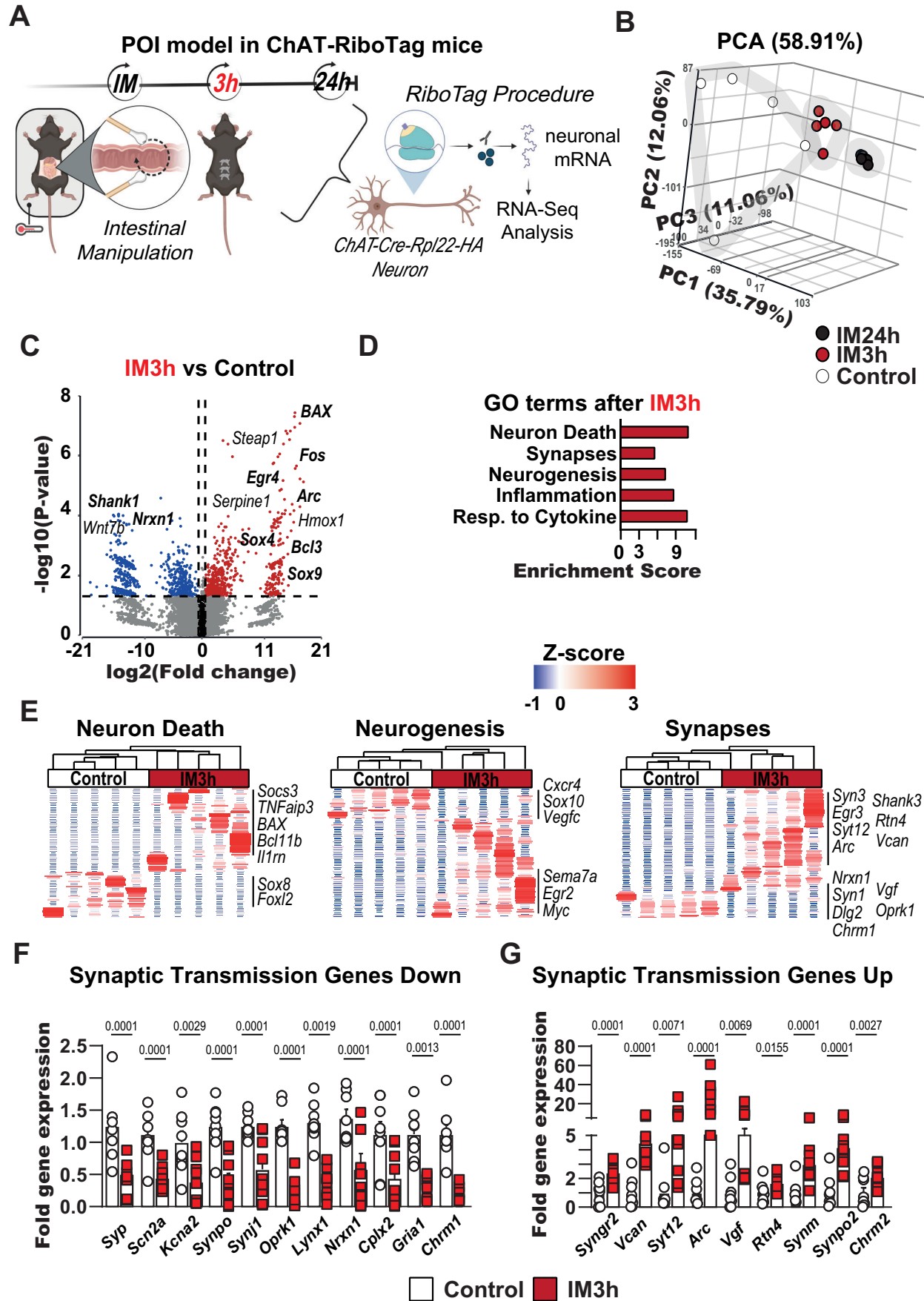

◀ **Figure 3. Inflammation-triggered neuronal activation leads to enteric neurodegeneration.**

(A) Schematic overview of the experimental setup and *RiboTag* procedure to generate the specific transcriptome of enteric neurons in the POI animal model. ChAT[cre]/Rpl22[HA/+] mice were subjected to IM, and *ME* samples were collected from IM3h, IM24h, and control animals to isolate RNA and subsequently performed RNA-Seq analysis from enteric neurons. (B) The principal component analysis (PCA) of the neuronal RiboTag samples from POI mice presents a separation of the three groups. (C) The volcano plot shows significantly changed genes between the IM3h and control group. The plot depicts 607 up- and 786 downregulated genes with a fold change ≥1.5. $n = 5$ per group. (D) Gene ontology (GO) analysis of significantly changed genes ($P < 0.05$) shows induction of GO terms connected to neuronal functions, inflammation, and proliferation. (E) Heatmaps of genes significantly changed in myenteric neurons connected to neuron death, neurogenesis, and synaptic structures from IM3h and control animals. (F, G) Gene expression analysis of factors involved in synaptic transmission (F) downregulated and (G) upregulated in IM3h animals. Bar graphs show the fold gene induction normalized to control mice. $n = 8$ (control), 9 (IM3h). Statistical analysis is based on Fisher's exact $t$ test (C, D) and Student's $t$ test (F, G). Standard deviations are presented as SEM. Source data are available online for this figure.

downregulated genes at IM3h and 2010 up- and 856 downregulated genes at IM24h (Fig. EV4E). Notably, at IM24h, a volcano plot also showed 2357 up- and 41 downregulated genes between infiltrating monocyte-derived and resident macrophages with inflammatory genes in the group of the highest fold changes (Appendix Fig. S3c). The robust inflammatory activation during POI is also documented by a heatmap for inflammatory response genes (Fig. EV4F). Analysis of all DEGs discovered an apparent neurodegenerative macrophage phenotype, concluded from the enrichment of GO terms for prominent neurodegenerative diseases (Parkinson's, Alzheimer's, and Huntington's) throughout POI development, elevated already at IM3h (Fig. 4G). A common characteristic of neurodegenerative diseases is the phagocytosis of damaged neurons by microglia (Butler et al, 2021). To corroborate a similar mechanism in POI, we applied a published approach to quantify phagocytosis in vivo (Viola et al, 2023). Subjecting double transgenic mice (*Cx3cr1[gfp/+]/Chat[Cre]-Ai14-floxed* (tdTomato)) that label macrophages (green) and neuronal structures (red) to our POI model and performing flow cytometry at IM24 h revealed a vast increase in GFP[+]/tdTomato[+] cells in the *ME*, indicating that macrophages had phagocyted neuronal structures (Fig. 4H and gating strategy in Appendix Fig. S4a). Phagocytosis was confirmed by confocal microscopy (Appendix Fig. S3d) by an intracellular (= phagocyted) localization of the tdTomato[+] material in GFP[+] macrophages. A closer look revealed that the phagocytic cell population consisted of almost equal amounts of resident and infiltrating macrophages (Fig. 4I), although the transcriptional data indicated a more pronounced phagocytic activity in resident cells (Fig. EV4G). Further characterizing resident macrophages in POI, we found a more than fivefold increased expression of mediators known to participate in neuronal damage, such as *S100a6* (Filipek and Leśniak, 2020), *S100a8* (Ghavami et al, 2010; Ha et al, 2021), *S100a9* (Ghavami et al, 2010; Wang et al, 2018), and *Lgals3* (Galactin3, (Xue et al, 2023; Soares et al, 2021)), and related to neuroinflammation including the cytokines *Il6* (Luo et al, 2019; Kozina et al, 2022), *Tnfaip2* (Zhang et al, 2023b; Kia et al, 2018; Zhao et al, 2018), *Il1a* (Zhang et al, 2023b; Guttenplan et al, 2020), *Il1b* (Zhang et al, 2023b; Pott Godoy et al, 2008) together with the PGE$_2$-producing enzyme (*Ptgs2,* (Nango et al, 2023; Bartels and Leenders, 2010)) (Fig. 4J,K). These findings imply a high production of potentially "neuro-destructive" factors by cells in close proximity to enteric neurons. In addition, our transcriptional data hint at a inflammatory and neuro-deteriorating phenotype in resident macrophages as presented in heatmaps by genes involved in "regulation of neuron death" and "inflammatory response" (Fig. EV4H; Appendix S4b). These findings, therefore, point to a more pronounced role of resident macrophages compared to infiltrating cells in promoting enteric neurodegeneration. To further examine this

hypothesis, we subjected CCR2 knock-out mice, a mouse line characterized by reduced numbers of circulating monocytes and infiltrating macrophages, to our POI animal model. Previous work has already verified reduced amounts of infiltrating immune cells and a prolonged motility impairment in CCR2$^{-/-}$ mice after IM (Farro et al, 2017), thereby allowing us to focus on the impact of resident macrophages on enteric neurodegeneration. At IM24h, we validated the decreased infiltration (fewer MPO$^+$ cells: Fig. 4I) and fewer macrophages around ganglia (Fig. EV4J). However, no differences were detected between ANNA1$^+$ cell numbers (Fig. EV4K) or synaptic damage (Fig. EV4L) in CCR2$^{-/-}$ and WT mice, indicating no or only a weak impact of infiltrating monocytes and monocyte-derived macrophages on neuron death in POI. These findings further pronounce the significance of resident macrophages in this context.

Validating additional similarities between reactive resident macrophages and microglia in the cause of neurodegeneration, we focused on cellular metabolism, as microglia can damage neurons by modulating their redox status (Muddapu et al, 2020; Jha et al, 2017). Interestingly, in the list of highly enriched GO terms, we detected changes in gene clusters connected to cellular metabolic processes, such as "metabolic pathways", "glycolysis", "carbon metabolism", and "HIF1 pathway", spiking particularly at the early disease stage (Appendix Fig. S4c). As metabolic changes in tissue-resident macrophages can directly affect the metabolic status of surrounding cells, including neurons (Bennett and Perona-Wright, 2023; Wculek et al, 2022), we reviewed the neuronal *RiboTag* transcriptome data to examine the metabolism of enteric neurons on gene expression level. At IM3h, we detected high enrichment in GO terms connected to primary and cellular metabolic processes as well as gene clusters regulating mitochondrial membrane biology and respiratory chain processes (Appendix Fig. S4d), whose dysregulations can induce neurodegeneration (Esteras et al, 2023). In line with the GO term data, prominent metabolism regulator genes (*Nfe2I2* (Esteras et al, 2023), *Keap1* (Esteras et al, 2023), *Hmox1* (Li et al, 2021), and *Hif1a* (Zhang et al, 2023a; Merelli et al, 2018; Zhang et al, 2011): Appendix Fig. S4e) presented an early and continuous upregulation, indicating simultaneous metabolic changes in enteric neurons and resident macrophages in POI development, possibly tied to neurodegeneration.

Our results suggest that activated resident macrophages are able to create and maintain a neurodegenerative environment for enteric neurons after POI induction, altering their gene expression profiles and inducing neuronal death programs.

## Depleting resident macrophages in the inflamed gut reduces enteric neurodegeneration

Given that activated resident macrophages contribute to neurotoxic actions in POI, we hypothesized that their depletion could reduce

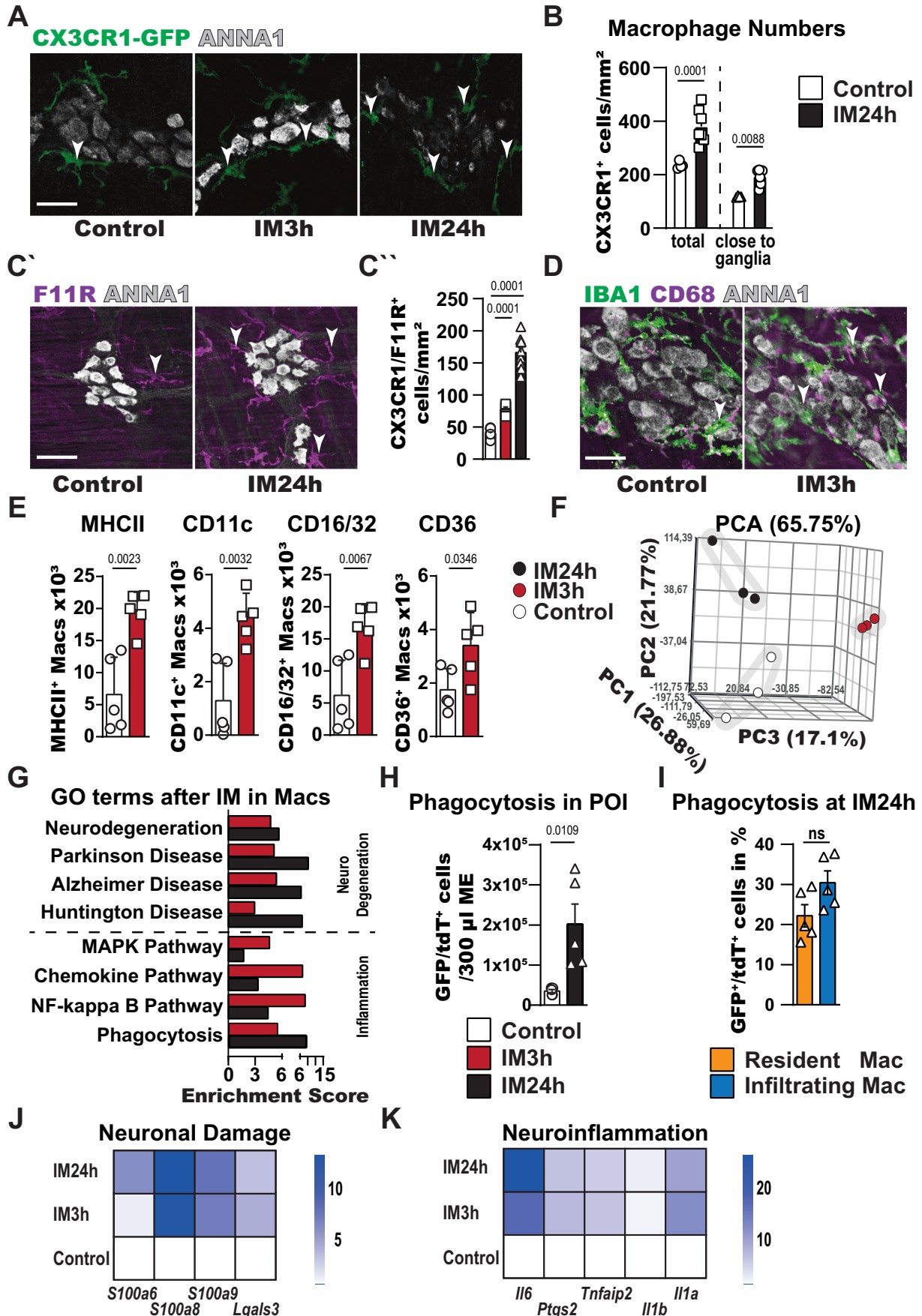

**Figure 4. Inflammatory macrophages are involved in enteric neurodegeneration.**

(A) Immunofluorescence analysis of myenteric neurons (ANNA1+, gray) and resident macrophages (CX3CR1-GFP+, green, white arrowheads) 3 and 24 h post IM and in control. Scale bar 50 μm. (B) Quantification of CX3CR1-GFP+ cells per mm² *ME* after IM in the whole jejunum *ME* tissue and surrounding ENS ganglia. CX3CR1-GFP+ macrophage number increased in total and around ENS ganglia in IM24h mice. Bar graphs show the mean cell number of CX3CR1-GFP+ cells normalized to the jejunum *ME* area. n = 4 (control), 7 (IM24h). (C´) Immunofluorescence analysis of neuron-associated macrophages (F11R+, white arrowheads) 24 h post IM and in control. Scale bar 50 μm. (C´´) Immunofluorescence quantification of neuron-associated macrophages (F11R+/CX3CR1+)3 of the jejunum *ME* tissue and 24 h post IM and in control. Quantification shows a clear increase in neuron-associated macrophages during POI. n = 10 (control), 5 (IM3h), 10 (IM24h), with each analyzed animal, one whole-mount jejunum specimen was used for the quantification. (D) Immunofluorescence analysis of myenteric neurons (ANNA1+, gray) and activated (CD68+, violet, white arrowheads) resident macrophages (Iba1+, green) 3 h post IM and in control. Scale bar 50 μm. (E) FACS analysis of IM3h animals and controls to validate the inflammatory activation of resident macrophages. MHCII, CD11c, CD16/32 and CD36 showed upregulation in IM3h animals. Bar graphs show the mean cell number of macrophages positive for the indicated inflammatory marker normalized to counting beads in 300 μl *ME* cell suspension. n = 5. (F) Principal component analysis (PCA) of macrophage samples from POI mice, representing a separation of the three groups. (G) GO analysis of significantly changed genes (P < 0.05) shows strong induction of GO terms connected to neurodegeneration and inflammation. (H, I) FACS analysis of POI animals and controls to validate phagocytosis of neuronal structures from resident and infiltrating macrophages. Cx3cr1gfp/+/ChatCre-Ai14-floxed mice were used in the POI animals model to quantify double-positive macrophages. (H) Macrophages (CX3CR1-GFP+, green) phagocyte more neuronal structures (tdTomato+, red) during POI. (I) At IM24h, resident and infiltrating macrophage populations show similar numbers for phagocytic cells. n = 5. (J, K) Heatmaps of genes significantly changed in resident macrophages connected to neuronal damage (J) and neuroinflammation (K) from IM and control animals. Heatmap values were generated by normalization to resident macrophage gene expression from control mice. Statistical analysis is based on Fisher's exact *t* test (G), Student´s *t* test (C, E, H, I), and one-way ANOVA (B, J, K). Standard deviations are presented as SEM. Source data are available online for this figure.

neuronal death and ameliorate disease symptoms. Therefore, we i.p. injected *Cx3cr1gfp/+* reporter mice with CD115-antibodies to reduce resident macrophage numbers in the intestine (Matheis et al, 2020; Pendse et al, 2023; Muller et al, 2014) and subsequently performed IM (Fig. 5A). Immunofluorescence revealed a strong reduction in CX3CR1-GFP+ macrophage numbers 48 h after injection, with more than 60% depletion in unoperated controls and around 50% depletion in IM24h animals (Figs. 5B and EV5A,B), which we confirmed by flow cytometry of colon *ME* (Fig. EV5C) and *Cx3cr1* gene expression analysis of jejunal *ME* (Fig. EV5D). In addition, quantification of MHCII+ macrophages displayed a similar depletion after CD115 injection in both control and IM24h mice (Fig. EV5E). Importantly, we also observed this depletion in CX3CR1-GFP+ macrophages, specifically surrounding ENS ganglia (Fig. 5C). Furthermore, while unoperated mice remained unaffected, depletion led to changes in two major hallmarks of POI: a reduced infiltration of leukocytes into the *ME* (Fig. 5D) and an ameliorated GI-transit time after IM (Fig. 5E). Accordingly, expression levels of prominent POI disease genes (*Arg1*, *Egr1*, *Ccl2*, *Il6*, (Schneider et al, 2020; Schneider et al, 2022)) showed less upregulation at IM24h in depleted animals and no changes in controls (Fig. EV5F). We quantified the neuron numbers after IM24h and, in line with our hypothesis, found more ANNA1+ neurons in the *ME* of CD115-depleted animals, indicating a smaller decrease in neuronal numbers and less neurodegeneration (Fig. 5F). Gene expression analysis supported these findings, as apoptosis genes (*Bcl-xL*, *Bcl2*) were less, and synaptic structure genes (*Syn1*, *Dlg4*, *Chat,* and *Snap25*) were more expressed in the macrophage-depleted in total RNA and neuronal *RiboTag* RNA samples of the *ME* at IM24h (Figs. 5G,H and EV5G,H).

Together with the observations in CCL2−/− mice, our data affirm that resident macrophages are the main contributors to enteric neurodegeneration, as their depletion—but not a strong reduction of infiltrating monocytes protects mice from inflammation-driven neuronal loss and, at the same time, improves POI hallmarks.

## Gut surgical trauma causes enteric neurodegeneration in patients

Having gained mechanistic insights into inflammation-driven enteric neuronal death in a murine model of gut trauma, we aimed to find supportive translational evidence for our conclusions in surgical patients. By collecting paired jejunal biopsies during a pancreatectomy, an extensive operation with surgical jejunal manipulation, at an early and late stage of the operation, we were able to investigate the effect of surgery-induced IM in patients (Fig. 6A). Histological analysis of ENS ganglia revealed activated, cFOS+ (Fig. EV6A) and dying, cleaved CASP3+ (Fig. 6B) neurons (ChAT+, white arrow) next to intraganglionic IBA1+ macrophages (white asterisk) already in the early patient samples after surgical trauma. Similar to the murine datasets, we detected significant upregulation of genes associated with cell death (*BCL2*, *BCL2A1*, *CASP3*, *CASP4*), neuroinflammation (*IL1B*, *PTGS2*, *PTGES*, *IL6*, *TNFAIP2*, *TNFAIP3*, *S100A8*, *S100A9*), neurogenesis (*SOX2*, *SOX9*, *FOXP1*, *BMP2*) and proliferation (*MYC*, *CCND1*, *CCNE1*) in late compared to early patient samples (Fig. 6C–F). To get a broader overview of the activation pattern in the jejunal *ME* of patients, we then performed bulk RNA-seq analyses. A PCA showed a distinct clustering of both groups, representing an expected shift in gene expression after surgery (Fig. EV6B). The corresponding volcano plot highlighted DEGs previously discussed in murine POI, like *ARC*, *BCL3*, *S100A9, FOSL1*, and *MYC*, indicating the activation of similar pathways in patients after surgical trauma (Fig. EV6C). GO analysis applied to the DEG list confirmed the enrichment of gene clusters of POI hallmarks, such as inflammation, migration, and proliferation (Fig. EV6D). Notably, we also observed enrichment in the GO terms "synapse", "neurogenesis", "neuron differentiation", and "neuron apoptosis" (Fig. 6G). Concurring heatmaps expanded on these changes by depicting unique patterns for early and late samples with distinct up- and downregulated gene groups, including genes like *S100A8 + A9*, *BCL3*, *EGR2 + 3*, *SYN3* and *DLG4* (Fig. 6H,I). Venn diagrams of all three GO terms showed the number of overlapping DEGs with the murine POI data set, indicating a conserved gene core related to inflammation-induced neurodegeneration in the gut across species (Fig. EV6E). Next, we added patient samples to validate the RNA sequencing data and investigated specific synaptic structures (*SYN1-3*, *DLG2*, *SHANK1-3*) and transmission (*ARC*, *VCAN*) gene expression. We discovered a severe dysregulation partially overlapping with data from POI mice (Fig. 6J,K), with the majority of analyzed genes, such as *SYN3*, *SHANK1*, *ARC*, and *VCAN*, being upregulated at the late time point. Following our hypothesis about metabolic changes leading to enteric neurodegeneration, we also discovered a strong overlap between the human and murine datasets for genes involved in

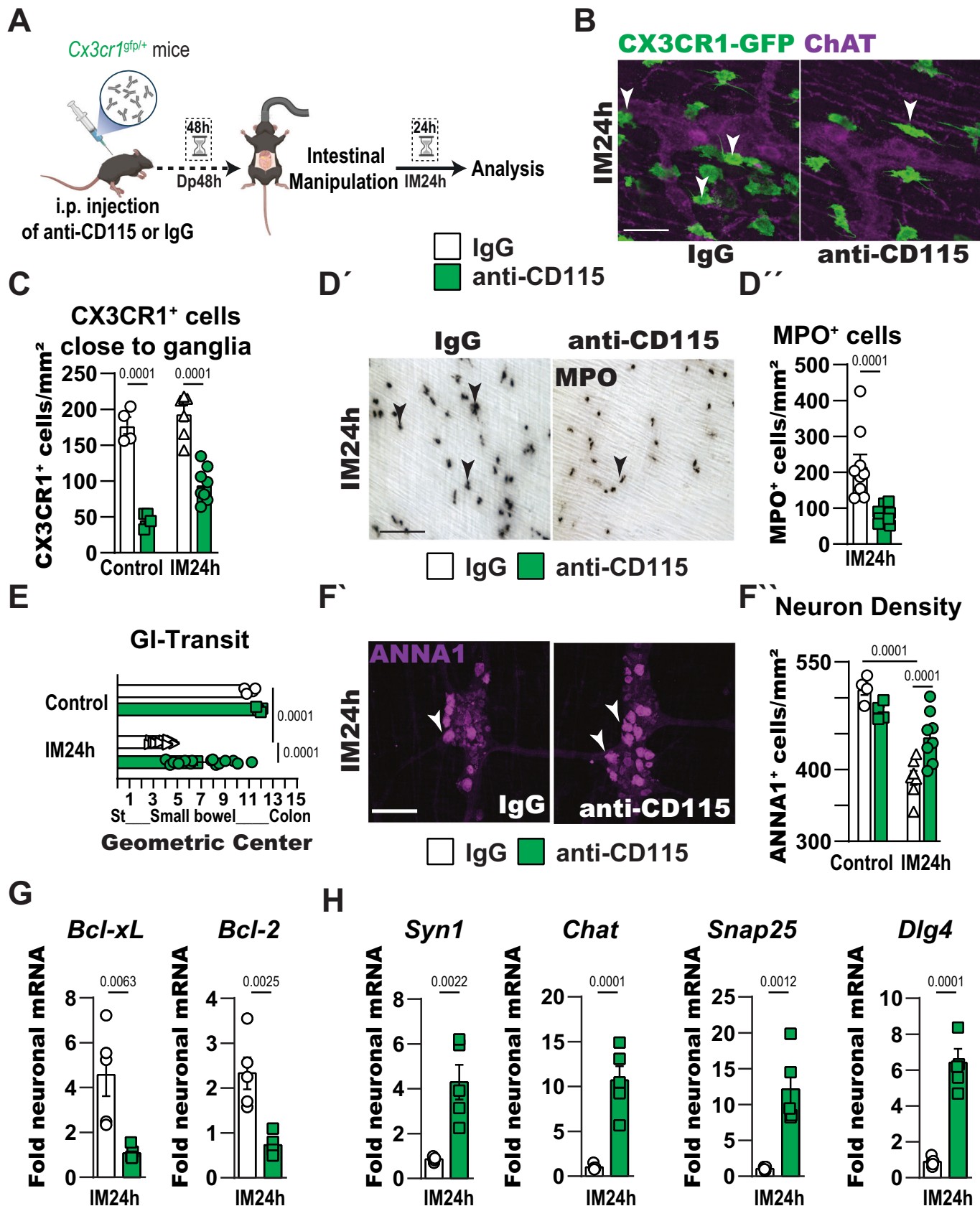

**Figure 5. Depleting macrophages in the inflamed gut reduces enteric neurodegeneration.**

(A) Schematic overview of the experimental setup to deplete macrophages in the *ME* of POI mice. CX3CR1$^{gfp/+}$ mice were i.p. injected with anti-CD115 antibodies or IgG-control and subjected 48 h later to IM. POI animals with and without depletion of macrophages were analyzed 24 h post IM. (B) Immunofluorescence analysis of myenteric neurons (ChAT$^+$, violet) and resident macrophages (CX3CR1-GFP$^+$, green, white arrowheads) after depletion treatment, 24 h post IM. Scale bar 50 µm. (C) Quantification of CX3CR1-GFP$^+$ cells per mm² *ME* surrounding ENS ganglia with and without CD115-depletion in control and IM animals. Bar graphs show the mean cell number of CX3CR1-GFP$^+$ cells normalized to the *ME* area. $n = 4$ (both controls), 7 (IM24h_IgG), 8 (IM24h_CD115). (D′) Immunohistological analysis of activated (MPO$^+$, black, black arrowheads) leukocytes in animals 24 post IM after previous anti-CD115 or IgG treatment. Scale bar 100 µm. (D′′) Quantification of MPO$^+$ cells per mm² jejunum ME in control and IM animals with and without CD115-depletion. Bar graphs show the mean cell number of MPO$^+$ cells normalized to the jejunum *ME* area. $n = 9$. (E) Gastrointestinal (GI) transit analysis with FITC-dextran in animals 24 post IM and controls after previous anti-CD115 or IgG treatment. Bar graphs show the mean of the geometric center calculated by compiling the fluorescent data from the whole GI tract. $n = 4$ (both controls), 15 (IM24h_IgG), 15 (IM24h_CD115). (F′) Immunofluorescence analysis of myenteric neurons (ANNA1$^+$, violet) after anti-CD115 or IgG treatment 24 h post IM and in control. Scale bar 50 µm. (F′′) Quantification of ANNA1$^+$ cells per mm² jejunum *ME* tissue after anti-CD115 or IgG treatments in IM or control mice. Bar graphs show the mean number of ANNA1$^+$ cells normalized to the jejunum *ME* area. CD115 treatment significantly reduced the number of enteric neurons (myenteric plexus, ANNA1$^+$, white arrowheads) in IM24h but not control mice, implying that resident macrophages have an important role in postoperative neurodegeneration. $n = 4$ (both controls), 7 (IM24h_IgG), 8 (IM24h_CD115). (G, H) Neuronal mRNA expression analysis of factors involved in cell death (G) and synaptic structures (H) in Chat-Cre RiboTag animals undergoing anti-CD115 or IgG treatment with IM. Bar graphs show the fold gene induction normalized to their respective IgG-treated counterparts. $n = 5$. Statistical analysis is based on Student´s $t$ test (D, G, H) and one-way ANOVA (C, E, F). Standard deviations are presented as SEM. Source data are available online for this figure.

metabolic processes, represented by upregulation of redox target genes (*NFE2L2*, *HIF1A*, *HMOX1*, *HMOX2*, *SOD2*: Fig. EV6F), enrichment of matching gene clusters ("response to ROS", "metabolic process", Fig. EV6G) and visibly strong induction of multiple genes related to metabolic processes (Fig. EV6H).

Together, our data from surgical patients uncovered a neurodegenerative phenotype similar to the murine one, characterized by an apparent neuronal activation and cell death, indicating a conserved neuroimmune mechanism during gut inflammation across species.

## Discussion

In this study, we focused on the impact of acute intestinal inflammation on the neuronal network of the ENS, thereby discovering an immediate enteric neuron activation in the early inflammatory phase followed by a switch towards a neurodegenerative phenotype. We identified resident macrophages as the responsible players mediating this acute enteric neurodegeneration.

Enteric neurons and glia are known to coordinate propulsive motility patterns in the GI tract, ensuring a functional oral to aboral transport of ingested food (Holland et al, 2021). In POI, a common post-operative complication observed after abdominal surgery, this coordinated motility is disturbed (Sommer et al, 2021). Meanwhile, clear evidence exists that a surgery-induced inflammation of the *muscularis externa* (*ME*) contributes to POI development. The exact cellular sequence triggering intestinal neuroinflammation in the *ME* is still elusive; despite this, adrenergic-dependent enteric glia reactivity (Leven et al, 2023) as well as resident muscularis macrophage activation (Wehner et al, 2007) are seen as disease initiators, infiltrating neutrophils then promote the inflammatory cascade (Stein et al, 2018), while infiltrating monocyte-derived macrophages mediate disease resolution (Farro et al, 2017). Together, these cells exert a motility-disturbing effect by impairing enteric neuronal function. Even though enteric neuron involvement in motility disruption is known, investigations of the molecular responses and cellular fate of enteric neurons in the neuroinflammatory environment have been largely disregarded. Recently, we and others expanded the view on ENS involvement in POI by showing that enteric glia become reactive by enhanced adrenergic

stimulation (Leven et al, 2023), extracellular ATP (Schneider et al, 2020), or IL-1 (Stakenborg et al, 2022; Schneider et al, 2022), which in turn shapes the immune environment and triggers the disease (Schneider et al, 2022; Leven et al, 2023). As enteric glia are closely intertangled with enteric neurons and are in direct contact with macrophages, we anticipated simultaneous enteric neuron activation and subsequent molecular and cellular changes during the development of POI.

Indeed, abdominal surgery leads to a variety of cell activations. Previous work showed early response gene-1 (EGR-1) induction within the immediate post-operative phase in smooth muscle cells and enteric neurons (Schmidt et al, 2008), depicting neuronal activation as part of the early trauma phase. Herein, we found that the reliable cellular activation marker cFOS (Lara Aparicio et al, 2022) is expressed in enteric neurons early on, confirming their activation at POI onset, simultaneously with other cell types, including immune cells (Sui et al, 2022), smooth muscle cells, and enteric glia (Leven et al, 2023). Given that immediate neuronal activation is part of the postsurgical response, we reanalyzed our published RNAseq datasets (Schneider et al, 2020; Schneider et al, 2022; Leven et al, 2023; Mallesh et al, 2021) and generated new proteomic data from post-operative *ME* samples to review the molecular changes in enteric neurons. Interestingly, even in full-tissue *ME* specimens, we discovered a considerable amount of differentially expressed genes and proteins pertaining to neuronal dysfunction, death, neurogenesis, and neuronal proliferation. Even though a neuronal response is highly expected due to the impaired GI-motility and its direct connection to neuron density (Boschetti et al, 2019), the sheer magnitude was unforeseen. Recent studies highlighted several distinct interactions of neurons during gut inflammation (Margolis and Gershon, 2016; Populin et al, 2021; Yang et al, 2023), such as their communication with mast cells (Forsythe, 2019), macrophages (Meroni et al, 2019), and ILCs (Cardoso et al, 2017; Klose et al, 2017). In addition, they produce inflammatory mediators after TLR activation (Burgueño et al, 2016) and control the inflammatory microenvironment by releasing neuropeptides (Aresti Sanz and El Aidy, 2019) and neurotransmitters (Jakob et al, 2020; Zhu et al, 2022). Notably, Margolis et al uncovered that elevated enteric neuronal density correlates positively with a more severe progression of colitis (Margolis et al, 2011), while in the POI model, neuronal loss was associated

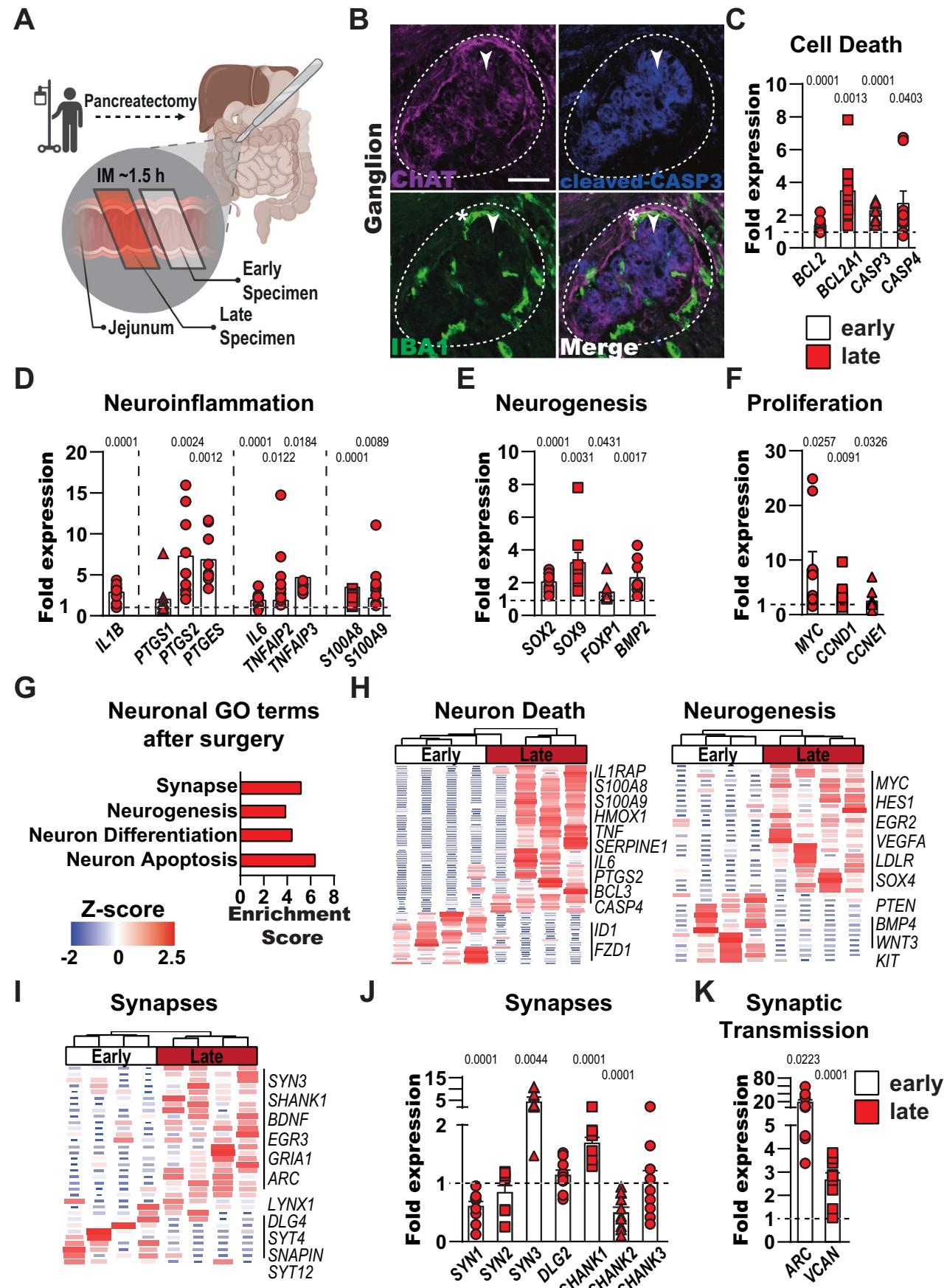

**Figure 6. Gut surgical trauma causes enteric neurodegeneration in patients.**

(A) Schematic overview of human sample collection. Patients undergoing pancreatectomy had two jejunal biopsies taken during surgery (early and late specimens with ~1.5 h between collections). (B) Immunofluorescence analysis of apoptotic (cleaved-CASP3$^+$, blue) myenteric neurons (ChAT$^+$, violet) and surrounding resident macrophages (IBA1$^+$, green, white asterisk) in the *ME* of the early human jejunal specimen. After surgery double-positive neurons (white arrowheads) in the enteric ganglia (white dashed line) are detected, indicating apoptotic myenteric neurons. Scale bar 50 μm. (C–F) Gene expression analysis of cell death markers (C), neuroinflammation (D), neurogenesis (E), and proliferation (F) in human jejunum *ME* samples. Bar graphs show the fold gene induction normalized to early jejunum *ME* samples. $n = 9$. (G–I) Bulk 3′ RNA-Seq analysis of early and late jejunal samples. (G) GO analysis of significantly changed genes ($P < 0.05$) shows induction of GO terms connected to neuronal functions such as synapse, neurogenesis, and neuron death. (H) Heatmaps of genes connected to neuron death and neurogenesis in early and late jejunal specimens. (I) Heatmap of genes connected to synapses in early and late jejunal specimens. (J, K) Gene expression analysis of factors involved in synaptic structures (J) and transmission (K) in early and late jejunal specimens. Bar graphs show the fold gene induction normalized to early jejunum *ME* samples. $n = 9$. Statistical analysis is based on Fisher's exact *t* test (G) and Student´s *t* test (C–F, J, K). Standard deviations are presented as SEM. Source data are available online for this figure.

with worse disease outcomes, indicating a significant but controversial role of neurons in acute and chronic inflammatory cascades in the gut. Further confirmation of neuron-specific effects came from our *RiboTag* approach (Leven et al, 2021), presenting enriched GO terms related to neuronal death, neurogenesis, and proliferation that further underlined the neurodegenerative phenotype already in the early disease stage.

Interestingly, we found a simultaneous induction of these signatures, suggesting compensatory mechanisms counteracting the pending neuronal loss. Notably, the existence of enteric neurogenesis in the intestine is still under debate, as only a few studies have investigated it (Grundmann et al, 2019), and published results range from high rates of neuron proliferation (Kulkarni et al, 2017) to no detection at all (Virtanen et al, 2022). Compensation of the enteric neuron loss during POI by generation of new neurons is very likely, as we found induction of neuronal proliferation by upregulation of Ki67-positive myenteric neurons and neurogenesis gene patterns in our sequencing data of the diseased mice. Interestingly, recent publications also discuss additional functions of Ki67, such as its role in cell cycle arrest and/or cell synchronization (Miller et al, 2018; Sun and Kaufman, 2018). These functions may play an alternative role in the recovery of neurons during intestinal inflammation in POI.

Our long-term approach with animals 21 days after surgery validated the enteric neurogenic potential or renewal. Following the activation of relevant signaling cascades and the disturbance in neuronal homeostasis after IM, we detected a clear recovery from the POI symptoms and the neurodegenerative phenotype. In gut injury models (Laddach et al, 2023), neurogenesis has been detected with Sox2$^+$ enteric glia giving birth to new neurons (Belkind-Gerson et al, 2017), and recent scRNA-Seq studies confirmed an extraordinary potential of intraganglionic glia in neurogenesis (Guyer et al, 2023). In accordance, our latest work revealed an elevated proliferative state in intraganglionic glia after POI induction (Schneider et al, 2020; Leven et al, 2023), further highlighting glia-based neurogenesis as a mechanism of neuronal replenishment.

In the ENS, the term neurodegeneration is reasonably novel. The terminology has been proposed in various ENS conditions associated with Parkinson´s Disease (PD) (Virga et al, 2018), DSS-induced colitis (Sun et al, 2021), rheumatoid arthritis (Piovezana Bossolani et al, 2019), trimethyltin toxicity (Septyaningtrias et al, 2023), intestinal obstruction (Herdewyn et al, 2014), and aging (Camilleri et al, 2008). Neurodegeneration, initially described in the CNS (Chitnis and Weiner, 2017), is a central element of disease progression (Zhang et al, 2023b; Ransohoff, 2016) and recovery

(Cheng et al, 2022). In PD, for example, enteric neurodegeneration has been recently introduced to describe synaptic damage and neuron death in the ENS (Virga et al, 2018; McQuade et al, 2021; Schaffernicht et al, 2021; Garretti et al, 2023; Montanari et al, 2023) corresponding to the CNS phenotype (Wilson et al, 2023); however, more studies are needed to confirm the enteric neuron phenotype and discover novel similarities between the injured brain and gut.

In addition to the loss of enteric neurons and the presence of neurodegenerative signatures, we also see a massive dysregulation of synaptic gene expression patterns in enteric neurons, characterized by downregulation of Synapsins, ChAT, and PSD-95 in myenteric ganglia. While the synaptic integrity is mostly undisturbed at the protein level at POI onset, it becomes dramatically damaged after 24 h, proven by histological, transcriptional, and translational validation, including a specialized measurement of neurodegenerative markers in blood plasma samples. This aspect is especially relevant in light of the apparent neuronal death, as this dysregulation indicates an obstruction of synaptic signaling in the surviving enteric neurons. ENS synaptic signaling has been thoroughly defined (Fung and Vanden Berghe, 2020; Vanden Berghe et al, 2008), and its synaptic dysfunction is often associated with extrinsic innervating neurons during gastrointestinal inflammation (Brierley and Linden, 2014) or preluding neurological disorders (Rao and Gershon, 2016). Here, we connect acute inflammation to a synaptic dysfunction of enteric neurons during POI. Therefore, we suggest pharmacological maintenance or restoration of synaptic signaling and integrity as a valuable strategy for promoting POI recovery. Accordingly, previous work showed that motility disturbances normalized after 72 h (Stein et al, 2018; Schneider et al, 2022; Leven et al, 2023) in wild-type animals, proving that endogenous mechanisms favor a return to vital neuronal function, resulting in regular propulsive motility even though neuron numbers are still reduced. As post-operative motility disturbances to some degree occur in all patients and are accepted as a temporary and "normal" response in POI pathophysiology, prolonged disturbances might show insufficient recovery of neuronal and myenteric synaptic function.

In terms of identifying the cellular and molecular source triggering post-operative neurodegeneration, we suspected the ganglia-neighboring resident macrophages since they have been established previously as initiators of the inflammatory cascade after gut trauma (Wehner et al, 2007). Our histological examinations showed enrichment of macrophages around ganglia in the *ME*, including the recently proven neuron-associated F11R$^+$ subpopulation (Viola et al, 2023), indicating a possible interaction during POI. Macrophages share many functional characteristics

with microglia (Verheijden et al, 2015), which transition to a reactive state and are an important therapeutic target for fighting CNS neurodegeneration (Gao et al, 2023). As shown for microglia, which are able to actively cause neuronal death (Brown and Vilalta, 2015; Butler et al, 2021), we also found the induction of "neurotoxic" factors such as neuroinflammatory cytokines (Luo et al, 2019; Kozina et al, 2022; Kia et al, 2018; Zhao et al, 2018; Guttenplan et al, 2020; Pott Godoy et al, 2008; Nango et al, 2023; Bartels and Leenders, 2010; Zhang et al, 2023b) and neuro-damaging proteins (Filipek and Leśniak, 2020; Ghavami et al, 2010; Ha et al, 2021; Wang et al, 2018; Xue et al, 2023; Soares et al, 2021), all related to the gene ontology term "regulation of neuron death", in resident macrophages immediately after surgery, persisting at least 24 h. During the disease, resident macrophages showed a clear inflammatory phenotype with an increased phagocytosis of neuronal structures. Furthermore, they change their metabolic activity, inducing the gene expression of glycolysis genes, which is known to induce a proinflammatory state (Wculek et al, 2022). Certainly, these factors are not only produced by macrophages, as other immune cells (Sui et al, 2022) and enteric glia (Schneider et al, 2020; Leven et al, 2023) have also been shown to release them in the context of POI.

In addition, comparing resident and infiltrating macrophages at the peak of the disease, we understood that resident cells are the driver of enteric neurodegeneration. On the transcriptional level, the resident cells induced more genes related to "neuron death", including neuro-damaging factors, and showed a strong inflammatory activation pattern compared to the infiltrating ones. Using CCR2$^{-/-}$ mice in our POI model, we could further support the crucial role of resident cells. CCR2$^{-/-}$ animals lack infiltrating monocytes and monocyte-derived macrophages after IM (Farro et al, 2017), making them a perfect tool to compare the roles of resident and infiltrating cells in enteric neurodegeneration after IM. Hence, we validated neuron death and synaptic damage and discovered that the knock-out mice developed the same neurodegenerative phenotype as their WT littermates, indicating only a weak impact of the infiltrating cells on neuronal damage in POI. This conclusion sides with the accepted neurosupportive, pro-resolving connotation of infiltrating monocyte-derived macrophages (Farro et al, 2017; Kim et al, 2022; Guilliams et al, 2018). On the other hand, a reduction of resident macrophages before surgery led to protection from POI disease hallmarks and showed less associated neurodegeneration, represented by a smaller amount of neuron death and a reduced transcriptional downregulation of synaptic genes in enteric neurons. Findings by Dora et al showed similar protective effects of disease symptoms after macrophage depletion with L-clodronate in a DSS-induced colitis model (Dora et al, 2021). However, resident macrophage actions might not be per se detrimental but rather disease-specific, as they were shown to exert neuro-supportive functions in gut infection (Matheis et al, 2020).

Finally, we found translational evidence of enteric neurodegeneration in jejunal samples from patients undergoing pancreatectomy collected at an early and later stage during surgery. Despite the earlier time point of sample collection compared to murine POI, we found a clear enrichment of neuronal death-associated genes as well as a reduction in genes indicative of synaptic dysfunction, highlighting conserved pathways. Although we were not able to test the presence of these pathways selectively in human enteric

neurons, these data support the existence of species' overarching pathways regarding enteric neuronal fate. Future studies on single neurons from human gut tissue are needed to identify key genes for important neuronal functions in inflammation. Importantly, detecting similar neurodegeneration in the murine POI model offers new treatment opportunities. First, neuroprotective treatments established for the CNS can be extended to the ENS to ameliorate motility impairments and, hopefully, speed up ENS recovery after surgical trauma. For many scheduled surgeries, neuroprotective pretreatment and follow-up doses could be administered to the patient. For CNS neurodegeneration, treatments exist in PD (Hirsch and Hunot, 2009), intriguingly also with the metabolic factor NRF2 (Zhang et al, 2019), whose expression was permanently induced in enteric neurons of POI mice. In this regard, recent studies in the gut showed some promising targets for neuroprotective therapies. For instance, VEGF signaling effectively prevented neuronal death in the ENS (Hecking et al, 2022). In addition, the steroid progeterone shielded the ENS from oxidative stress (Stegemann et al, 2023), and vitamin D saved enteric neurons from palmitic acid-induced damage (Larsson and Voss, 2018). A recent review also highlighted the neuroprotective properties of omega-6 oxylipins in the gut (Mantel et al, 2023). Interestingly, a more clinically available approach, prucalopride, a commercially available 5-HT4 receptor agonist, protected human enteric neurons from H$_2$O$_2$-induced cell death (Bianco et al, 2016), confirming former studies that already reported 5-HT4 receptor-mediated neuroprotection in mice (Liu et al, 2009). These studies promote the idea of future studies focusing on the maintenance of neuronal function and integrity as a supportive measure in POI prevention.

In summary, our data show that enteric neurons participate in the inflammatory cascade after surgical gut trauma and are driven toward a neurodegenerative program by resident macrophages. Certainly, our study is only a starting point in defining mechanisms behind neurodegeneration in the gut and comprehending the recovery of neuronal function after POI. Follow-up work has to identify specific pathways and markers in neurons and resident macrophages to get the complete story. However, counteracting these neurodegenerative processes could be a new strategy to protect patients from surgery-induced ENS complications, thereby positioning enteric neuroprotection as a valuable intervention strategy for treating inflammatory diseases of the gut.

## Methods

**Reagents and tools table**

| Reagent/resource | Reference or source | Identifier or catalog number |
|---|---|---|
| **Experimental models** | | |
| Postoperative Ileus Model | PMID: 33332729 | na |
| CSF1-Ab depletion Model | PMID: 31923400 | na |
| *RiboTag* Model | PMID: 34939271 | na |
| **Antibodies** | | |
| For FACS | Various companies | Appendix Table S1 |
| For Histology | Various companies | Appendix Table S1 |
| **Oligonucleotides and other sequence-based reagents** | | |

| Reagent/resource | Reference or source | Identifier or catalog number |
|---|---|---|
| qPCR primers | Selfmade | Appendix Table S3 |
| **Chemicals, enzymes, and other reagents** | | |
| QIAzol lysis Reagent | Qiagen | Cat. No. 79306 |
| Golgi-cox Reagent | NeuroTechnologies INC | Cat. No. PK401 |
| Hanker–Yates Reagent | Polyscience Europe | Cat. No.: 08661-5 |
| 70kDA FITC-dextran | Sigma Aldrich | Cat. No.: 46945 |
| Ecosurf | ThermoScientific | Cat. No.: 17217056 |
| Epredia Shando Immu-Mount | ThermoScientific | Cat. No.: 9990402 |
| TissueTek O.C.T. Compound | Sakura | Cat. No.: 4583 |
| SYBR Green PCR Master Mix | Applied Biosystems | Cat. No.: A46012 |
| **Software** | | |
| Prism 10 | GraphPad | na |
| Partek Flow | Partek Inc | na |
| FlowJo 9 | BD | na |
| *FACSDiva* software | BD | na |
| NIS-Elements version AR 5.42.04 software | Nikon | na |
| BioRender | BioRender | na |
| Adobe CS2024 | Adobe | na |
| ImageJ software | ImageJ | na |
| **Other** | | |
| Neurology 4-Plex E SIMOA assay | Quanterix | Cat. No. 103670 |
| Tape station 4200 system | Agilent | Cat. No. G2991BA |
| Qubit HS dsDNA assay | ThermoScientific | Cat. No. Q33230 |
| miRNeasy Micro Kit | Qiagen | Cat. No. 217084 |
| QuantSeq 3′ RNA-Seq Library Prep Kit | Lexogen | Cat. No.: 015.384 |
| NextSeq 500/550 High Output Kit v2.5 | Illumina | Cat. No. 20024904 |
| High-Capacity cDNA Reverse Transcription Kit | Applied Biosystems | Cat. No. 4368814 |
| Quantstudio 5 machine | ThermoScientific | Cat. No. A34322 |
| Leica SP8 AOTF confocal microscope | Leica | na |
| Nikon Eclipse Ti2 | Nikon | na |

## Animals

$ChAT^{Cre/+}$ (B6;129S6-Chattm2(cre)Lowl/J) /$Rpl22^{HA/+}$ (*B6N.129-Rpl22tm1.1Psam/J, abbreviated as RiboTag*) and CCR2-KO (B6.129S4-Ccr2tm1Ifc/J) mice were used for IM and following histological and transcriptional analyses. $CX3CR1^{GFP/+}$ (B6.129P2(Cg)-Cx3cr1tm1Litt/J) mice were used for macrophage depletion experiments and transcriptional analyses. Animals were housed under SPF conditions in the central and our satellite housing facility (University Hospital Bonn, Bonn, Germany). Male mice (10–18 weeks) were used in the intestinal manipulation

experiments, and mice of both sexes (10–20 weeks) were used for macrophage depletion experiments (female mice only for establishing the depletion method, while only male mice were used for POI experiments) carried out under German federal law (AZ.: 81 02.04.2020.A357). All animal studies were performed in a blinded manner to reduce subjective bias from the experimentators.

## Post-operative ileus model

The POI mouse model was induced by intestinal manipulation, as described previously (Leven et al, 2023). Animals were sacrificed 3 h and 24 h after manipulation. In short, 15 min prior to surgery, animals received pain medication (Tramadol 30 mg/kg (Grünenthal, Aachen, NRW, DE); s.c.). The animals received isoflurane inhalation anesthesia and were laid on heating pads to stabilize body temperature. Laparotomy (2 cm) was performed along the *linea alba*, and the abdominal cavity was held open by clamps while the small bowel was gently eventrated and placed on moist gauze. The small bowel was then mechanically stimulated with light pressure of two opposing moist cotton swaps in a rolling motion toward the *cecum* twice. The intestine was gently placed back, and the opened cavity closed with running sutures. Animals were returned to their cages, and for post-anesthesia recovery, they were placed under a red-light lamp for 30 min. Animals received oral pain medication (Tramadol, 1 mg/ml (Aliud Pharma, Laichingen, BW, DE)) ad libitum via their drinking water.

## Gastrointestinal transit measurement

Gastrointestinal transit (GIT) was assessed by measuring the intestinal distribution of gavaged fluorescently labeled dextran 90 min after administration, as described previously (Leven et al, 2023). Briefly, animals were received a gavage of 100 µl 70kDA FITC-dextran (Sigma Aldrich, St. Louis, MO, USA) and rested for 90 min without additional food or water. Then, animals were sacrificed, intestines extracted, and the gastrointestinal tract separated into 15 segments. The stomach (st) correlates with a geometric center (GC) of 1, the small bowel correlates with a GC of 2–11, the cecum (c) correlates with a GC of 12, and the colon correlates with a GC of 13–15. Segments were flushed with Krebs-Henseleit buffer, contents were centrifugated at $5000 \times g$ for 5 min, and eluates were analyzed for FITC fluorescence. The GC was calculated to generate GI-transit for naive, IM24h, IM21 days, and animals previously treated with Anti-CD115 Antibody (Dp48h) or IgG.

## Flow cytometry (FACS)

FACS analysis was performed on isolated *ME* samples of the small bowel or colon 24 h or 3 h after IM and/or 48 h after CD115-depletion. The *ME* was isolated by sliding small bowel segments onto a glass rod, removing the outer muscularis circumferentially with moist cotton applicators, and cutting the *ME* tissue into small pieces. *ME* tissue was digested with a 0.1% collagenase type II (Worthington Biochemical, Lakewood, NJ, USA) enzyme mixture, diluted in PBS, containing 0.1 mg/ml DNase I (La Roche, Germany), 2.4 mg/ml Dispase II (La Roche, Germany), 1 mg/ml BSA (Applichem, Germany), and 0.7 mg/ml trypsin inhibitor

(Applichem, Germany) for 40 min in a 37 °C shaking water bath. Afterward, single-cell suspension was obtained using a 70 μm filter mesh, and cells were stained for 30 min at 4 °C with the appropriate antibodies. Antibodies used in this study are shown in Appendix Table S1. Flow cytometry analyses were performed on *FACSCanto II* (BD Biosciences, Franklin Lakes, NJ, USA) using *FACSDiva* software, and data were analyzed with the latest *FlowJo* software (Tree Star, Ashland, OR, USA).

## Neurology 4-Plex E SIMOA assay

NfL, GFAP, Aβ40 and Aβ42 levels were determined from the blood plasma of naive and POI mice using the human *NEUROLOGY 4-PLEX E* Advantage Kit (*Quanterix*, USA) on a SIMOA HD-X analyzer, software v.3.1 (*Quanterix*) by a blinded experimentator following the manufacturer's instructions. Samples were thawed on ice and randomized on plates. Plasma samples were measured in unicates in an on-bench 1:2.5 dilution.

## Immunofluorescence

Immunofluorescence was performed on longitudinally opened whole-mount specimens of ileum mounted on Sylgard gel-covered Petri dishes with the mucosa facing upwards. Following fixation in 4% PFA at 4 °C overnight, the mucosal layer was mechanically removed, allowing the permeabilization (1% Ecosurf (ThermoScientific)/PBS, RT, 15 min) and blocking (5% donkey serum, 0.25% Ecosurf/PBS, RT, 1 h) of the *ME* before incubation with antibodies (primary antibody: 4 °C, overnight; secondary antibody: 1.5 h, RT; Appendix Table S1).

For human samples of early and late time points, *ME* tissue was separated from the mucosa and fixed overnight in 4% PFA at 4 °C. Fixed tissues were incubated overnight in 30% sucrose at 4 °C and subsequently embedded in TissueTek O.C.T. Compound (Sakura, Germany). Cryosections of human *ME* were cut at 14 μm thickness on SuperFrost Plus Adhesion slides (Epredia, NH, USA). Sections were blocked and stained as described above. Specimens were mounted in Epredia Shando Immu-Mount (ThermoScientific) and imaged on a Leica SP8 confocal imaging system, Nikon Eclipse 2000TE, or Nikon T*i*2 fluorescent microscope.

## Hanker–Yates histology

Staining with Hanker–Yates reagent (Polyscience Europe, Germany) was performed on mucosa-free *ME* whole mounts of the terminal ileum. The samples were fixated in 100% ethanol for 10 min before mechanical separation of the mucosal and *ME* layers. Hanker–Yates myeloperoxidase staining solution was administered to whole mounts for 15 min before mounting. Myeloperoxidase-expressing cells (MPO[+]) were quantified by determining the mean number of MPO[+] cells/mm$^2$ for five random areas per animal.

## Golgi-cox histology

Staining with Golgi-cox reagents (FD NeuroTechnologies INC, USA) was performed on cryo-embedded Swiss rolls of the small bowel in accordance with the provided manual. Sectioned specimens were imaged at a Nikon Eclipse T*i*2 (see "Microscopy" section).

## Microscopy

Quantitative analyses of proliferation, cell death, and numbers of ANNA1[+], CX3CR1-GFP[+], or MPO[+] cells were performed with widefield images obtained on a Nikon Eclipse TE2000-E with a magnification of ×200 and a field of view of 397 μm × 317 μm or a Nikon Eclipse T*i*2 with a magnification of ×200 and a field of view of 769 μm × 769 μm. Representative images are confocal slices obtained with a Leica SP8 AOTF confocal microscope using a ×400 objective.

For quantification of CX3CR1[−] and MHCII[+] cells adjacent to enteric ganglia of the *ME*, five random widefield images per whole-mount specimen were taken, and immunolabeled cells overlapping or being closely associated to enteric ganglia were counted per enteric ganglion. Data for each animal is composed of the mean of immunolabelled cells adjacent to all ganglia found in the five images. All quantifications were performed by using ImageJ software counting plugins.

Human specimens were imaged using the Nikon Eclipse T*i*2 and Nikon NIS-Elements version AR 5.42.04 software.

## Macrophage depletion

Anti-CD115 antibodies were produced with hybridoma cell cultures cultivated from rat plasma cells, as mentioned before (Sudo et al, 1995). In all, 50 mg/kg (Matheis et al, 2020) of rat anti-CD115 or rat isotype control (Cat.#I4131 (MERCK, NJ, USA)) was administered to each mouse per intraperitoneal injection 48 h before the planned analysis. Dosage and treatment schedules were adopted from previous studies (Matheis et al, 2020). After depletion (Dp48h) or control treatment, mice underwent intestinal manipulation.

## Human surgical specimens

The collection of patient surgical specimens was approved by the ethics committee of North–Rhine–Westphalia, Germany (*Accession number: 266_14*). Informed consent was obtained to procure human surgical tissue from the small bowel (jejunum) of patients undergoing a pancreatectomy at an early and a late time point during the surgical procedure (Appendix Table S2). Human samples were collected and used for RNA-Seq and histology analysis.

Written consent was obtained for all patient samples, and the study conformed to the principles set out in the WMA Declaration of Helsinki and the Department of Health and Human Services Belmont Report.

## *RiboTag* method

*RiboTag* Immunoprecipitation was performed according to Leven et al, 2021 (Leven et al, 2021). Briefly, the muscle layer of the whole small bowel tissue was mechanically separated from the mucosal layer and placed in liquid nitrogen. The *ME* tissue was lysed on a Precellys homogenizer (Bertin Instruments, France) (3 × 5000 rpm, 45 s; 5 min intermediate incubation on ice) in pre-cooled homogenization buffer (50 mM Tris/HCl, 100 mM KCl, 12 mM MgCl$_2$,1% NP-40, 1 mg/mL Heparin, 100 μg/mL Cycloheximide, 1 mM DTT, 200 U/mL RNAsin, 1× Protease Inhibitor P8340, (all

SIGMA Aldrich)), centrifuged (10 min, 10,000 × g, 4 °C), and supernatants saved. Control samples labeled "Input" were generated from 50 μl of the cleared lysate. The supernatants were incubated with anti-HA antibody coupled to magnetic beads (Cat.#HY-K0201 (MedChemExpress Company, NJ, USA), 5 μl; 1 mg/mL; Appendix Table S1; 4 h, 4 °C, 7 rpm) and incubated (overnight, 4 °C, 7 rpm). Before elution of the cell-specific mRNA and following mRNA extraction (Qiagen RNeasy micro kit, (Qiagen, Germany)), high salt buffer (50 mM Tris/HCl, 300 mM KCl, 12 mM MgCl₂,1% NP-40, 100 μg/mL Cycloheximide, 0.5 mM DTT, (all SIGMA Aldrich)) was used to wash the beads thrice from the remaining supernatant while magnetically fixed.

## cDNA synthesis and quantitative PCR analysis

Total RNA was extracted from *ME* specimens at indicated time points after IM and control using the RNeasy Mini Kit (Qiagen, Germany), followed by deoxyribonuclease I treatment (Ambion, Austin, TX). Complementary DNA was synthesized using the High-Capacity cDNA Reverse Transcription Kit (Applied Biosystems, Darmstadt, Germany). High-Capacity cDNA Reverse Transcription Kit was used to transcribe the purified RNA (1 μg) according to the manufacturer's instruction manual. The mRNA expression was quantified by real-time RT-PCR with primers shown in Appendix Table S3. Quantitative polymerase chain reaction was performed with SYBR Green PCR Master Mix (Applied Biosystems, Darmstadt, Germany) on a Quantstudio 5 machine (ThermoScientific, USA).

## RNA-Seq analysis

Libraries prepared with QuantSeq 3' RNA-Seq Library Prep Kit (Lexogen, Greenland, NH, USA) were sequenced (single-end 50 bp, 10 M reads) on an Illumina Hiseq 2500. RNA-Seq data was analyzed with Partek Flow software (Partek Inc., MO, USA) using Lexogen pipeline *12112017*, which included two adapter trimming and a base-trimming step with subsequent quality controls (QC). Reads were aligned with star2.5.3, followed by a post-alignment QC and quantification to an annotation model. Normalized counts were subjected to principal component and gene set analysis. Pipeline information can be found within our uploaded sequencing files.

## SMART-Seq2 analysis

Flow cytometry sorted cells were resuspended in Trizol and preserved in QIAzol lysis reagent (Qiagen, Germany). For RNA extraction, the miRNeasy Micro Kit (Qiagen, Germany) was used following the manufacturer´s instructions. Quantity and integrity of RNA were assessed with an HS RNA assay on a tape station 4200 system (Agilent, CA, USA). The RNA sequencing was performed by generating non-strand-specific, full transcript sequencing libraries using the Smart-seq2 protocol, published by Picelli et al (Picelli et al, 2014). The Qubit HS dsDNA assay (ThermoScientific, USA) was used to quantify libraries. The distribution of fragment sizes was assessed by a D1000 assay on a tape station 4200 system (Agilent, CA, USA). The sequencing was carried out in a single-end mode (75 cycles) on a NextSeq 500 System (Illumina) with NextSeq 500/550 High Output Kit v2.5 (150 cycles) chemistry. "Partek Flow" software was used to analyze RNA-Seq data (Lexogen pipeline 12112017) as described in "RNA-Seq analysis".

## Mass spectrometry analysis

### Sample preparation for LC/MS measurements

In all, 50 μg of protein per sample were mixed with Lyse buffer (Preomics iST-NHS kit). Reduction, alkylation, and digestion were performed according to the Preomics iST-NHS kit protocol. 0.25 mg of TMTpro isobaric labeling reagent (18 plex) were added to each peptide sample and incubated at room temperature for 1 h, followed by quenching with 10 μL 5% hydroxylamine. Pooled peptides were dried in a vacuum concentrator and dissolved in 20 mM ammonium formate. Peptides were separated by reversed-phase fractionation at elevated pH on a Xbridge Shield RP18 column, 3.5 μm particles, 1.0 × 100 mm, Waters GmbH, Eschborn, Germany). Fractions were combined into six pools and dried in a vacuum concentrator.

### LC/MS measurements

Peptides were dissolved in 0.1% formic acid (FA) and separated on a Dionex Ultimate 3000 RSLC nano HPLC system (Dionex GmbH, Idstein, Germany). 1 μg was injected onto a C18 analytical column (self-packed 400 mm length, 75 μm inner diameter, ReproSil-Pur 120 C18-AQ, 3 μm, Dr. Maisch). Peptides were separated during a 180 min gradient from 2% to 35% solvent B (90% acetonitrile, 0.1% FA) at 300 nL/min. The nanoHPLC was coupled online to an Orbitrap Fusion Lumos mass spectrometer (ThermoFisher Scientific, Bremen, Germany). Peptide ions between 330 and 1600 *m/z* were scanned in the Orbitrap detector. In a top-speed method, peptides were subjected to collision-induced dissociation for identification (CID: 0.7 Da isolation, normalized energy 30%) and fragments analyzed in the linear ion trap with AGC target 50% and maximum fill time 35 ms, rapid mode. Fragmented peptide ions were excluded from repeat analysis for 30 s. Top 10 fragment ions were chosen for synchronous precursor selection and fragmented with higher energy CID (HCD: 2 Da MS2 isolation, 65% collision energy) for detection of reporter ions in the Orbitrap analyzer (range 100–500 *m/z*, resolution 50,000, maximum fill time 86 ms, AGC target 200%).

### Data processing

Raw data processing and database search were performed with Proteome Discoverer software 3.1.0.638 (ThermoFisher Scientific). Peptide identification was done with an in-house Mascot server version 2.8.3 (Matrix Science Ltd, London, UK). MS data were searched against Uniprot mouse reference proteome database (2023/04) and contaminants database (Hao lab (Frankenfield et al, 2022)). Precursor ion *m/z* tolerance was 10 ppm, fragment ion tolerance 0.5 Da (CID). Tryptic peptides with up to two missed cleavages were searched. C6H11NO-modification of cysteines (delta mass of 113.08406) and TMTpro on N-termini and lysines were set as static modifications. Oxidation was allowed as dynamic modification of methionine. Mascot results were evaluated by the Percolator algorithm version 3.05 (The et al, 2016) as implemented in Proteome Discoverer. Spectra with identifications above 1% *q* value were sent to a second round of database search with semi-tryptic enzyme specificity (one missed cleavage allowed). Protein N-terminal acetylation, methionine oxidation, TMTpro, and cysteine alkylation were then set as dynamic

**The paper explained**

**Problem**

Current studies provide compelling evidence that the enteric nervous system, together with resident macrophages, modulate neuroimmune processes in the gut after abdominal surgery. While substantial knowledge exists about macrophage–enteric glia interactions, the role of the enteric neurons, indispensable in gut homeostasis and gastrointestinal motility, has not been examined in detail during intestinal neuroinflammation. Therefore, we investigated the impact of enteric neuron–macrophage interactions on postoperative trauma and subsequent motility disturbances, i.e., postoperative ileus.

**Results**

We detected strong neuronal activation in the early postsurgical phase, followed by induction of neuronal proliferation, neuronal death, and synaptic degradation, all validated on transcript and protein level. A neuronspecific transcriptome analysis highlighted all mentioned neuronal responses to the inflammatory environment. Simultaneously, our study revealed a neurodegenerative profile in macrophage-specific transcriptomes during disease development. Depletion of resident macrophages by CSF-1R antibody treatment prior to surgical manipulation led to decreased neuronal death, less synaptic decay, and improved GI motility. Surgical manipulation of CCR2$^{-/-}$ mice, animals with low infiltration of monocyte-derived macrophages, showed no effect on enteric neurodegeneration, emphasizing the essential role of resident macrophages in this detrimental neuronal phenotype during intestinal neuroinflammation. In human jejunal gut samples, taken early and late during abdominal surgery, we substantiated the findings of our murine model and detected surgery-induced reactive and apoptotic neurons with dysregulated gene expression of synaptic signaling and neurogenic processes, revealing a species' overarching mechanism.

**Impact**

Our study demonstrates that surgical trauma and acute intestinal inflammation activate enteric neurons and induce neurodegeneration with severe synaptic decay, predominantly mediated by resident macrophages. This enteric neuron–macrophage interaction in gut inflammation paths the way for future studies focusing on neuroprotective mechanisms to dampen neurodegeneration and promote faster recovery after gastrointestinal surgery.

modifications. Actual FDR values were 0.6% (peptide spectrum matches) and 1.0% (peptides and proteins). Reporter ion intensities (most confident centroid) were extracted from the MS3 level, with SPS mass match >65%.

## Software

The software tools used for this study include Partek Flow, available from https://www.partek.com/partek-flow/#features; Gene Set Enrichment Analysis, available from https://www.partek.com/partek-flow/#features and http://bioinformatics.sdstate.edu/go/; Prism 10 for preparation of bar graphs and corresponding statistical analysis for the generated data; Biorender for generating the schematic illustrations and Adobe Illustrator and Photoshop for graphical adaption of the data for figure preparations.

## Statistical analysis

As indicated in the figure legends, statistical analysis was performed with Prism 10.0 (GraphPad, San Diego, CA, USA) using Student's *t* test and/or one-way ANOVA. In all figures, *P* values are indicated as *$P < 0.05$, **$P < 0.01$ and ***$P < 0.001$ when compared to control or #$P < 0.05$, ##$P < 0.01$, and ###$P < 0.001$ when compared to indicated samples. All plots are mean ± SEM. Animals for experiments were age- and sex-matched and randomly assigned to the experimental groups.

Partek Flow software was used for all analyses of all RNAseq data shown, and Fisher's exact test was applied, providing *P* values with multiple testing corrections.

Sample sizes for animal studies were chosen following previously reported studies that have used the POI animal model; at least six to ten independent mice per experimental setup. All animals were handled by standardized housing procedures and kept in precisely the same environmental conditions. They were genotyped at six weeks of age and received a randomized number by which they were identified. Age- and sex-matched animals were grouped randomly and used in the POI animal model. All the control or experimental mice in each experimental set were treated with the same procedure and manipulation. As a result of this, we avoided any group or genotype-specific effects due to the timing of experiments or handling of animals.

## Data availability

The datasets produced in this study are available upon reasonable request, and all transcriptional data is included in GEO databases: GSE263125, GSE198889, GSE205610, GSE134943 and GSE149181.

The source data of this paper are collected in the following database record: biostudies:S-SCDT-10_1038-S44321-024-00189-w.

## Peer review information

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

## Acknowledgements

The authors thank the Next Generation Sequencing Core Facility, the Mass Spectrometry Core Facility (DFG project number 386936527), the Bioinformatics Core Facility, and the Institute for Genomic Statistics and Bioinformatics of the University Clinics Bonn for supporting the RNA-Seq analysis. In addition, the authors thank the Flow Cytometry Core Facility (DFG project number 216372401) of the University Clinics Bonn for supporting all FACS experiments. The authors thank the technicians Patrik Efferz, Mariola Lysson, Jana Müller, Melissa Jürgens, and Bianca Schneiker for their support with the readouts like GIT, FACS, and qPCR and for handling the transgenic mouse lines. The authors thank Prof. Vanda A Lennon for sharing the ANNA1 antibody to visualize enteric neurons. Graphical visualizations were created with BioRender software. The authors thank the following funding organizations for supporting our research: Bonfor Postdoc (Instrument 2A) Grant: O-112.0060 (RS). Bonfor medical student Grant: O-112.0061 (RS + SW). ImmunoSensation2 Cluster Innovation Grant (RS), BonnNI medical student Grant: Q-611.0754 (SW). ImmunoSensation2 Cluster of Excellence: EXC 2151–390873048 (SW).

## Author contributions

**Mona Breßer**: Investigation; Visualization; Methodology; Writing—original draft; Writing—review and editing. **Kevin D Siemens**: Investigation; Visualization; Methodology; Writing—original draft; Writing—review and editing. **Linda Schneider**: Investigation; Visualization; Methodology; Writing—review and editing. **Jonah E Lunnebach**: Investigation; Methodology; Writing—review and editing. **Patrick Leven**: Supervision; Validation; Investigation; Methodology; Writing—review and editing. **Tim R Glowka**: Resources; Writing—review and editing. **Kristin Oberländer**: Resources; Investigation; Writing—review and editing. **Elena De Domenico**: Resources; Writing—review and editing. **Joachim L Schultze**: Resources; Writing—review and editing. **Joachim Schmidt**: Resources; Funding acquisition; Writing—review and editing. **Jörg C Kalff**: Resources; Funding acquisition; Methodology; Project administration; Writing—review and editing. **Anja Schneider**: Resources; Investigation; Writing—review and editing. **Sven Wehner**: Conceptualization; Supervision; Funding acquisition; Methodology; Project administration; Writing—review and editing. **Reiner Schneider**: Conceptualization; Supervision; Funding acquisition; Investigation; Visualization; Methodology; Writing—original draft; Project administration; Writing—review and editing.

Source data underlying figure panels in this paper may have individual authorship assigned. Where available, figure panel/source data authorship is listed in the following database record: biostudies:S-SCDT-10_1038-S44321-024-00189-w.

## Funding

## Disclosure and competing interests statement

SW and JCK receive royalties from Wolter Kluwer for contributing to the postoperative ileus section of the *UpToDate* library. The remaining authors declare no competing interests.

# Expanded View Figures

**Figure EV1. Intestinal manipulation and inflammation activate enteric neurons.**

(A) Immunohistochemistry analysis of enteric neurons (myenteric plexus, ANNA1$^+$, gray) in an activated (cFOS$^+$, red) state 24 h post intestinal manipulation (IM). Scale bar 50 μm. (B, C) RNA-Seq analysis of *ME* samples isolated from IM and control mice at the indicated time points. (B) Principal component analysis (PCA) of *ME* RNA samples from POI mice shows a separation of the three groups. (C) Heatmaps of genes connected to inflammatory response and migration in IM and control animals. (D–F) Gene expression analysis of factors involved in synapses (D) synaptic transmission (E) and synaptic signaling (F) in IM and control animals. Bar graphs show the fold gene induction normalized to control (naive) mice. $n = 10$ (control), 9 (IM3h), 10 (IM24h). (G, H) Mass spectrometry analysis of control and POI mice. (G) PCA of *ME* protein samples from POI and control mice shows a clear separation of the three groups. (H) Pathway analysis of significantly changed proteins in POI ($P < 0.05$) shows induction of pathways connected to inflammation and migration. $n = 6$. Statistical analysis is based on Fisher's exact *t* test (H) and one-way ANOVA (D–F). Standard deviations are presented as SEM. Source data are available online for this figure.

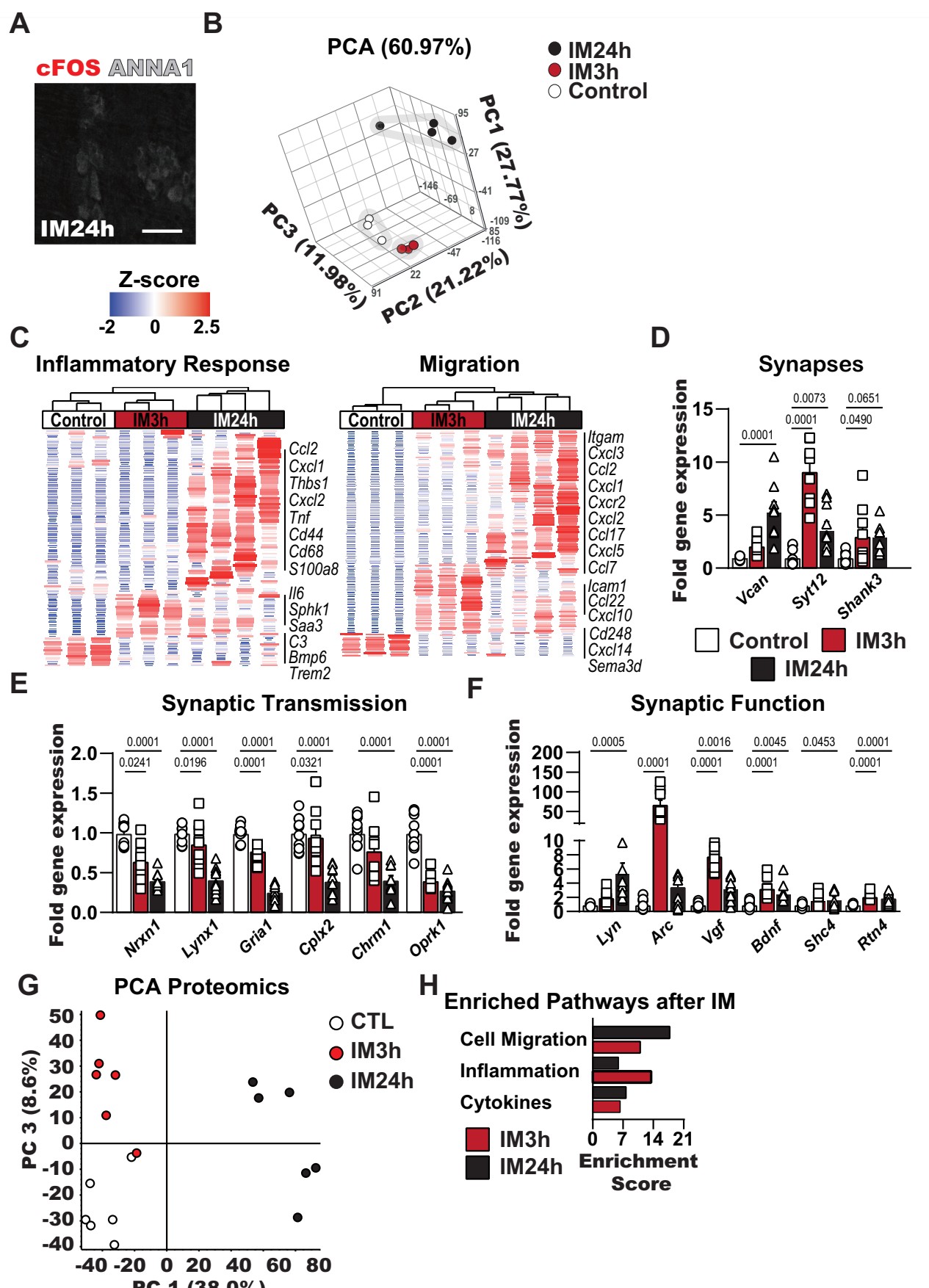

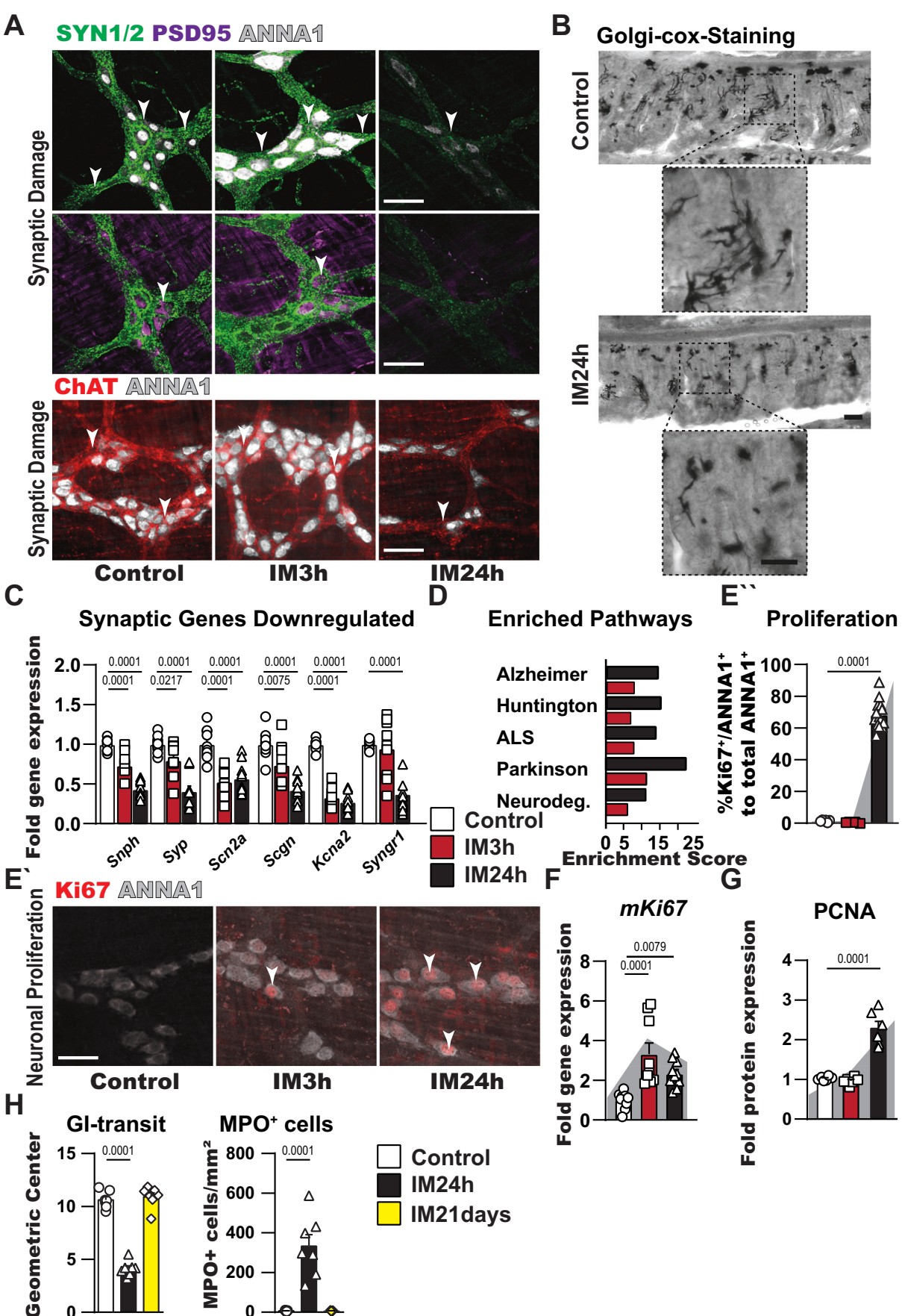

**Figure EV2.   Intestinal manipulation and inflammation induce enteric neurodegeneration.**

(**A**) Immunohistochemistry analysis of myenteric neurons (ANNA1 +, gray), synaptic structures (SYNAPSIN 1/2, green), postsynaptic density protein 95 (PSD95, violet), and choline acetyltransferase (ChAT, red) 3 and 24 h post IM and in control. Scale bar 50 μm. (**B**) Golgi-vox staining of swiss roles from POI and control animals. Scale bar 25 μm. (**C**) Gene expression analysis of factors involved in synaptic transmission in IM and control animals. Bar graphs show the fold gene induction normalized to control mice. $n = 10$ (control), 9 (IM3h), 10 (IM24h). (**D**) Mass spectrometry analysis of control and POI mice. Pathway analysis of significantly changed proteins in POI ($P < 0.05$) shows induction of pathways connected to neurodegenerative diseases. (**E′**) Immunohistochemistry of enteric neurons myenteric plexus, ANNA1[+], gray) in a proliferative state (Ki67[+], red) 3 and 24 h post IM and in control. A distinct population of double-positive cells (white arrowheads) was detected in POI. Scale bar 50 μm. (**E′′**) Quantification of ANNA1[+]/Ki67[+] cells after IM. Control mice showed almost no double-positive cells, whereas, at IM3h, a low, and at IM24h, many proliferating myenteric neurons were detected. Bar graphs show the mean % of double-positive cells normalized to the total number of ANNA1[+] cells. n = 6 (control), 3 (IM3h), 14 (IM24h). (**F**) Gene expression analysis of the proliferation marker *mKi67* in IM and control animals. At IM3h and IM24h, a significant upregulation of *mKi67* gene expression was detected. Bar graphs show the fold gene induction normalized to control mice. $n = 10$ (control), 9 (IM3h), 10 (IM24h). (**G**) Protein expression analysis of the proliferation marker PCNA in IM and control animals. At IM24h, a significant upregulation of PCNA protein was detected by mass spectrometry. Bar graphs show the fold protein induction normalized to control mice. $n = 6$. (**H**) Analysis of POI hallmarks in IM24h, IM21 days and control mice. Gastrointestinal transit and leukocyte (MPO[+] cells) infiltration peaks at IM24h. $n = 7$. Statistical analysis is based on Fisher's exact *t* test (**D**) and one-way ANOVA (**E–H**). Standard deviations are presented as SEM. Source data are available online for this figure.

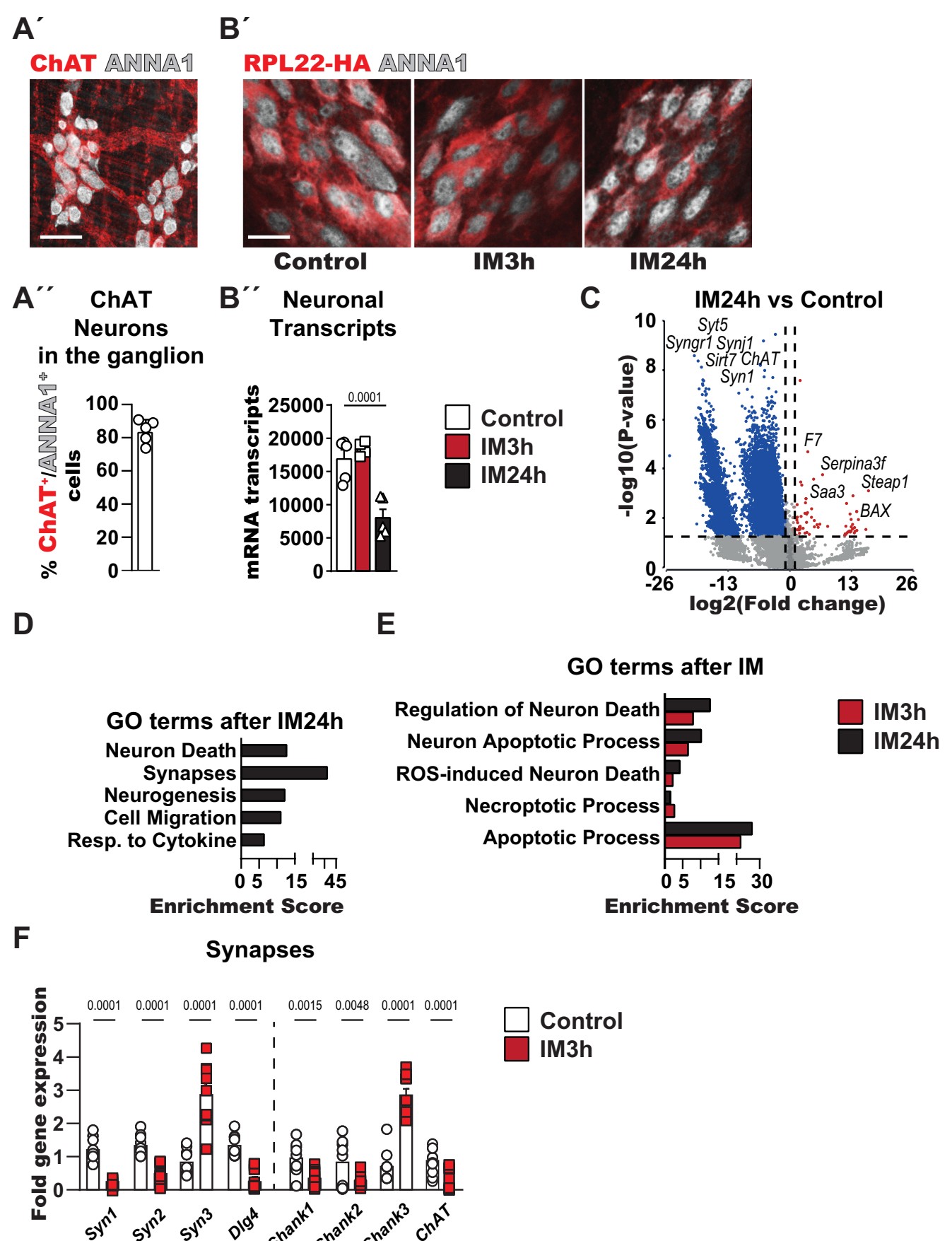

◀ **Figure EV3.** **Inflammation-triggered neuronal activation leads to enteric neurodegeneration.**

(**A′**) Immunohistochemistry analysis of myenteric neurons (ANNA1+, gray) and a neuronal subtype characterized by choline acetyltransferase expression (ChAT+, red) in jejunum *ME* whole-mount specimens. Scale bar 50 μm. (**A′′**) Quantification of ganglionic ANNA1+/ChAT+ cells. Around 80% of all ganglionic myenteric neurons in the healthy small bowel are ChAT+-neurons. Bar graphs show the mean % of double-positive cells normalized to the total number of ANNA1+ cells. $n = 5$. (**B–E**) ChAT^Cre^/Rpl22^HA/+^ mice were subjected to IM, and *ME* samples collected from IM3h, IM24h, and control animals for immunohistochemistry and RNA-Seq analysis. (**B′**) Immunohistochemistry analysis of myenteric neurons (ANNA1+, gray) and their expression of HA-tagged ribosomes (RPL22-HA, red). At all disease stages, HA-tagged ribosomes are expressed by ANNA1+ cells. Scale bar 50 μm. (**B′′**) Quantification of neuronal mRNA numbers detected by RNA-Seq analysis after IM. Bar graphs show the mean number of neuronal mRNA transcripts at the analyzed stages of POI development. $n = 5$. (**C**) Volcano plot shows significantly changed genes between the IM24h and control. The plot depicts 78 up- and 13238 downregulated genes with a fold change ≥1.5. $n = 5$ per group. (**D**) Gene ontology (GO) analysis of significantly changed genes ($P < 0.05$) shows strong induction of GO terms connected to "neuronal functions", "inflammation", and "proliferation" in IM24h mice. (**E**) GO analysis of significantly changed genes ($P < 0.05$) showing strong induction of GO terms connected to "neuronal death" in myenteric neurons during POI. (**F**) Gene expression analysis of factors involved in synaptic structures in IM3h and control animals. Bar graphs show the fold gene induction normalized to control mice. $n = 8$ (control), 9 (IM3h). Statistical analysis is based on Fisher's exact *t* test (**C–E**), one-way ANOVA (**B**), and Student's *t* test (**F**). Standard deviations are presented as SEM. Source data are available online for this figure.

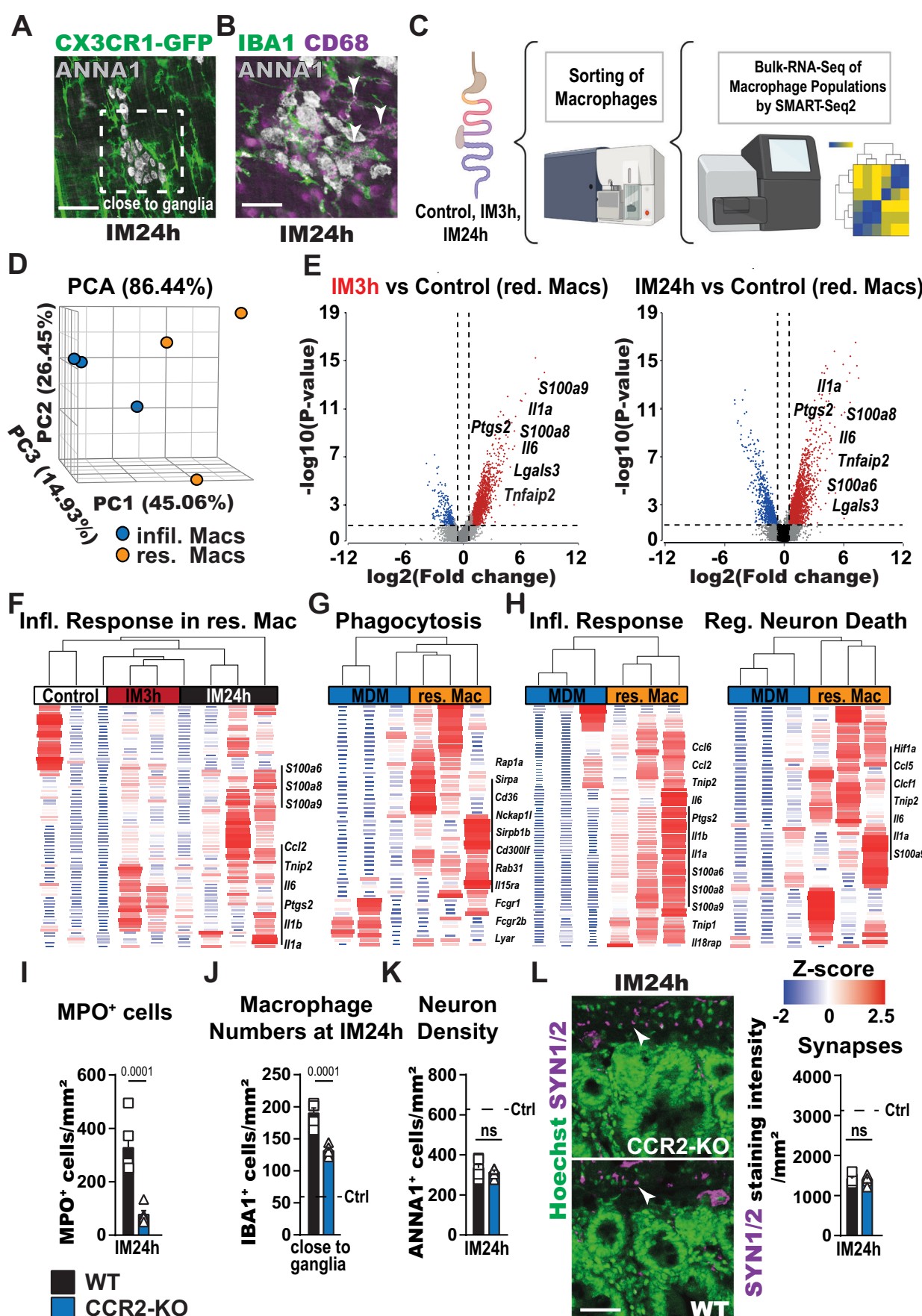

◀   **Figure EV4.   Inflammatory macrophages are involved in enteric neurodegeneration.**

(A) Immunohistochemistry analysis of myenteric neurons (ANNA1[+], gray) and resident macrophages (CX3CR1-GFP[+], green) 24 h post IM. The white dashed square illustrates the definition of macrophages close to ENS ganglia. Scale bar 50 μm. (B) Immunohistochemistry analysis of myenteric neurons (ANNA1[+], gray) and activated (CD68[+], violet, white arrowheads) resident macrophages (Iba1[+], green) 24 h post IM and in control. Scale bar 50 μm. (C) Schematic overview of the experimental setup to generate a transcriptome analysis in macrophages from POI mice. CX3CR1GFP/+ mice were subjected to IM, and *ME* samples were collected for FACS and subsequent RNA isolation from macrophages for SMART-Seq2 analysis. (D) PCA of samples from infiltrating monocyte-derived monocytes and resident macrophages at IM24h shows a separation of both cell types. (E) Volcano plots generated from the samples of resident macrophages show regulations (fold change >1.5) at IM3h with 2524 up- and 244 downregulated genes and at IM24h with 2010 up- and 856 downregulated genes. $n = 3$ per group. (F) The heatmap from resident macrophages presents significantly changed genes connected to the inflammatory response during POI. (G, H) The heatmaps from resident and infiltrating monocyte-derived macrophages show phagocyte genes (G), inflammatory response genes (H), and genes regulating neuron death (H) in resident compared to infiltrating cells. (I, L) Usage of CCR2[−/−] mice in the POI animal model. At IM24h, CCR2[−/−] mice have less infiltration of MPO[+] cells (I) and fewer IBA1[+] cells around ganglia (J). The reduced infiltration had no effect on neuronal numbers (K) and synaptic damage (L) in the gut. $n = 5$ (L) Immunohistochemistry analysis and quantification of synaptic structures (SYN1/2[+], violet, white arrowheads) and Hoechst as counterstaining (green) 24 h post IM in Swiss roles from the jejunum of WT and CCR2[−/−] mice. Bar graphs show the mean of staining intensity measurements in the indicated groups. $n = 5$. Scale bar 50 μm. Statistical analysis is based on Fisher's exact *t* test (E) and Student´s *t* test (I–L). Standard deviations are presented as SEM. Source data are available online for this figure.

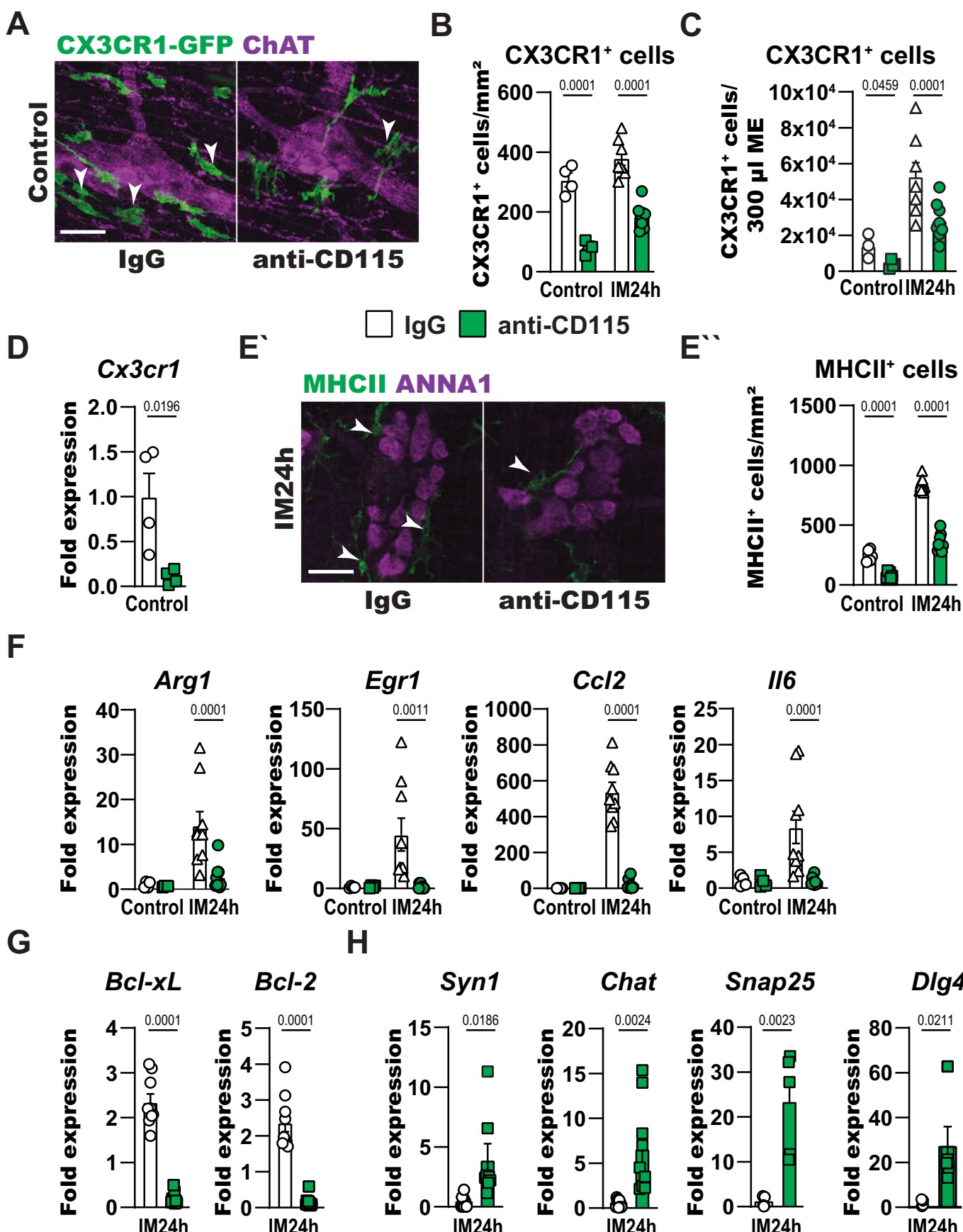

◀ **Figure EV5. Depleting macrophages in the inflamed gut reduces enteric neurodegeneration.**

(A) Immunohistochemistry analysis of myenteric neurons (ChAT+, violet) and resident macrophages (CX3CR1-GFP+, green, white arrowheads) in control mice after depletion treatment. Scale bar 50 μm. (B) Quantification of CX3CR1-GFP+ cells per mm² jejunum *ME* with and without depletion treatment in control and IM24h animals. Bar graphs show the mean CX3CR1-GFP+ cell number normalized to the *ME* area. $n = 4$ (both controls), 7 (IM24h_IgG), 8 (IM24h_CD115). (C) FACS analysis of CX3CR1-GFP+ cells per colon *ME* tissue weight in control and IM24h animals with and without depletion treatment. Bar graphs show the mean CX3CR1-GFP+ cell number normalized to the weight of colon *ME*. $n = 4$ (both controls), 7 (IM24h_IgG), 8 (IM24h_CD115). (D) Gene expression analysis of *Cx3cr1* in control animals undergoing anti-CD115 or IgG treatment. Bar graphs show the fold gene induction normalized to controls without depletion of macrophages. $n = 4$ (E′) Immunohistochemistry analysis of macrophages (MHCII+, green, white arrowheads) and myenteric neurons (ANNA1+, violet) in animals 24 post IM undergoing anti-CD115 or IgG treatment. Scale bar 50 μm. (E′′) Quantification of MHCII+ cells per mm² *ME* with and without CD115-depletion in control and IM24h animals. Bar graphs show the mean MHCII+ cell number normalized to the jejunum *ME* area. $n = 6$ (both controls), 9 (IM24h_IgG), 9 (IM24h_CD115). (F–H) Gene expression analysis of prominent factors involved in POI development (F), cell death marker genes (G), and synapses (H) in IM and control animals. Bar graphs show the fold gene induction normalized to their respective IgG-treated counterparts. $n = 4$ (both controls), 9 (IM24h_IgG), 9 (IM24h_CD115). Statistical analysis is based on one-way ANOVA (B, C, E, F) and Student's *t* test (D, G, H). Standard deviations are presented as SEM. Source data are available online for this figure.

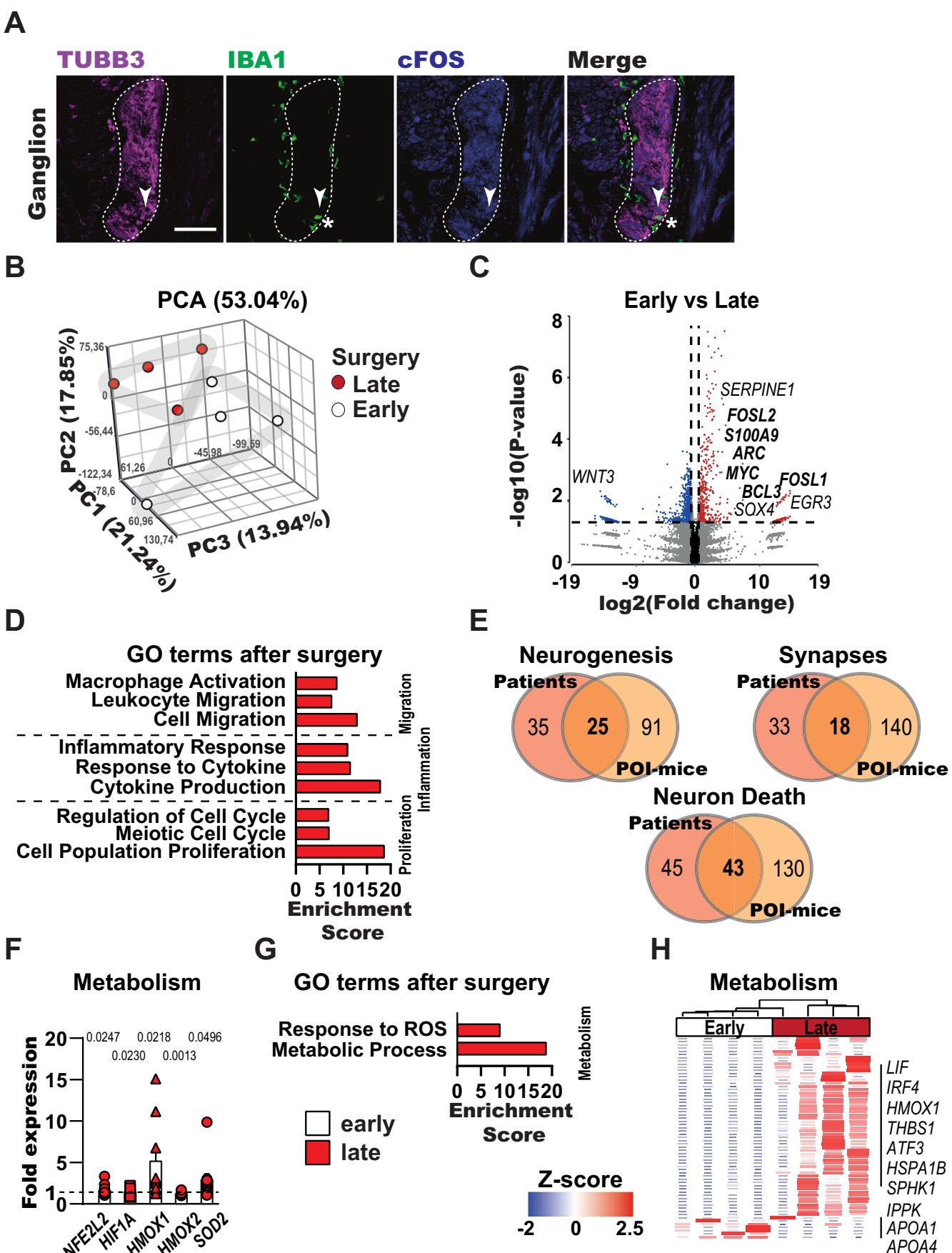

Figure EV6. **Gut surgical trauma causes enteric neurodegeneration in patients.**

(A) Immunohistochemistry analysis of activated (cFOS+, blue) myenteric neurons (TUBB3+, violet) and surrounding resident macrophages (IBA1+, green, white asterisks) in the *ME* of early human jejunum specimens. After surgery, double-positive neurons (white arrowheads) are visible in the enteric ganglia (white dashed line). Scale bar 50 μm. (B–D) 3′ Bulk RNA-Seq of human jejunal samples collected early and late during a pancreatectomy. (B) Principal component analysis (PCA) of the samples from early and late jejunum *ME* samples, representing a separation of both groups. (C) Volcano plot showing significantly changed genes between the early and late jejunal specimens. The plot depicts 488 up- and 891 downregulated genes with a fold change ≥1.5. $n = 4$ per group. (D) GO analysis of significantly changed genes ($P < 0.05$) shows induction of GO terms connected to POI hallmarks "migration", "inflammation", and "proliferation". (E) Venn diagrams show overlapping genes between surgical samples and POI mice connected to "neurogenesis", "synapses", and „neuron death". (F) Gene expression analysis shows genes associated with metabolic processes in human jejunum *ME* samples. Bar graphs show the fold gene induction normalized to early jejunum *ME* samples. $n = 9$. (G) GO analysis of significantly changed genes ($P < 0.05$) shows a strong induction of GO terms connected to metabolism. Bar graphs present the fold gene induction normalized to early jejunum *ME* samples. (H) Heatmap of genes connected to metabolic changes in early and late jejunal specimens. Statistical analysis is based on Fisher's exact *t* test (C, D, G) and Student´s *t* test (F). Standard deviations are presented as SEM. Source data are available online for this figure.

