## [Peer Review File · EMBO Molecular Medicine]

Macrophage-induced enteric neurodegeneration leads to motility impairment during gut inflammation

Mona Breßer, Kevin Siemens, Linda Schneider, Jonah Lunnebach, Patrick Leven, Tim Glowka, Kristin Oberländer, Elena De-Domenico, Joachim Schultze, Joachim Schmidt, Jörg Kalf, Anja Schneider, Sven Wehner, and Reiner Schneider

Corresponding author: Reiner Schneider (Reiner.schneider@ukbonn.de)

Review Timeline:

Submission Date:	14th May 24
Editorial Decision:	27th Jun 24
Revision Received:	25th Oct 24
Editorial Decision:	27th Nov 24
Revision Received:	17th Dec 24
Accepted:	17th Dec 24

Editor: Jingyi Hou

Transaction Report:

27th Jun 2024

Dear Dr. Schneider,

Thank you again for submitting your work to EMBO Molecular Medicine. First of all, please accept my apologies for the delay in getting back to you, which was due to the late arrival of referees' reports. As you will see from the reports below, the referees find your study of potential interest. However, they raise a series of concerns, which should be convincingly addressed in a major revision of the present manuscript.

The referees' recommendations are relatively straightforward, so there is no need for me to reiterate the points listed below. All the issues raised by the referees need to be carefully addressed. In particular, during our cross-commenting process (in which the referees are given the chance to make additional comments, including on each other's reports), Referee #3 indicated that for the manuscript to be accepted, it would be essential to "minimize data overinterpretation, provide at least partial validation of the gene expression data, provide some important quantifications in their image analysis and resolve the relationship between enteric neurons and macrophages at the early time point".

We would welcome the submission of a revised version within three months for further consideration. As you may already know, our editorial policy allows in principle a single round of major revision, and it is therefore essential to provide responses to the reviewers' comments that are as complete as possible.

Please also contact us as soon as possible if similar work is published elsewhere. If other work is published, we may not be able to extend the revision period beyond three months.

I look forward to seeing a revised form of your manuscript as soon as possible.

Use this link to login to the manuscript system and submit your revision: <https://embomolmed.msubmit.net/cgi-bin/main.plex>

Kind regards,
Jingyi

Jingyi Hou
Editor
EMBO Molecular Medicine

We require:

- 1) A .docx formatted version of the manuscript text (including legends for main figures, EV figures and tables). Please make sure that the changes are highlighted to be clearly visible.
- 2) Individual production quality figure files as .eps, .tif, .jpg (one file per figure). For guidance, download the 'Figure Guide PDF': (<https://www.embopress.org/page/journal/17574684/authorguide#figureformat>).
- 3) A .docx formatted letter INCLUDING the reviewers' reports and your detailed point-by-point responses to their comments. As part of the EMBO Press transparent editorial process, the point-by-point response is part of the Review Process File (RPF), which will be published alongside your paper.

- 4) A complete author checklist, which you can download from our author guidelines (<https://www.embopress.org/page/journal/17574684/authorguide#submissionofrevisions>). Please insert information in the checklist that is also reflected in the manuscript. The completed author checklist will also be part of the RPF.
- 5) Please note that all corresponding authors are required to supply an ORCID ID for their name upon submission of a revised manuscript.
- 6) It is mandatory to include a 'Data Availability' section after the Materials and Methods. Before submitting your revision, primary datasets produced in this study need to be deposited in an appropriate public database, and the accession numbers and database listed under 'Data Availability'. Please remember to provide a reviewer password if the datasets are not yet public (see <https://www.embopress.org/page/journal/17574684/authorguide#dataavailability>).
- In case you have no data that requires deposition in a public database, please state so in this section. Note that the Data Availability Section is restricted to new primary data that are part of this study.
- 7) For data quantification: please specify the name of the statistical test used to generate error bars and P values, the number (n) of independent experiments (specify technical or biological replicates) underlying each data point and the test used to calculate p-values in each figure legend. The figure legends should contain a basic description of n, P and the test applied. Graphs must include a description of the bars and the error bars (s.d., s.e.m.). See also 'Figure Legend' guidelines: <https://www.embopress.org/page/journal/17574684/authorguide#figureformat>
- 8) At EMBO Press we ask authors to provide source data for the main manuscript figures. Our source data coordinator will contact you to discuss which figure panels we would need source data for and will also provide you with helpful tips on how to upload and organize the files.
- 9) Our journal encourages inclusion of *data citations in the reference list* to directly cite datasets that were re-used and obtained from public databases. Data citations in the article text are distinct from normal bibliographical citations and should directly link to the database records from which the data can be accessed. In the main text, data citations are formatted as follows: "Data ref: Smith et al, 2001" or "Data ref: NCBI Sequence Read Archive PRJNA342805, 2017". In the Reference list, data citations must be labeled with "[DATASET]". A data reference must provide the database name, accession number/identifiers and a resolvable link to the landing page from which the data can be accessed at the end of the reference. Further instructions are available at .
- 10) We replaced Supplementary Information with Expanded View (EV) Figures and Tables that are collapsible/expandable online. A maximum of 5 EV Figures can be typeset. EV Figures should be cited as 'Figure EV1, Figure EV2" etc... in the text and their respective legends should be included in the main text after the legends of regular figures.
- For the figures that you do NOT wish to display as Expanded View figures, they should be bundled together with their legends in a single PDF file called *Appendix*, which should start with a short Table of Content. Appendix figures should be referred to in the main text as: "Appendix Figure S1, Appendix Figure S2" etc.
 - Additional Tables/Datasets should be labeled and referred to as Table EV1, Dataset EV1, etc. Legends have to be provided in a separate tab in case of .xls files. Alternatively, the legend can be supplied as a separate text file (README) and zipped together with the Table/Dataset file.

12) For more information: There is space at the end of each article to list relevant web links for further consultation by our

readers. Could you identify some relevant ones and provide such information as well? Some examples are patient associations, relevant databases, OMIM/proteins/genes links, author's websites, etc...

13) Author contributions: You will be asked to provide CRediT (Contributor Role Taxonomy) terms in the submission system. These replace a narrative author contribution section in the manuscript.

14) A Conflict of Interest statement should be provided in the main text.

16) All Materials and Methods need to be described in the main text using our 'Structured Methods' format, which is now required for all research articles. According to this format, the Methods section includes a Reagents and Tools Table (listing key reagents, experimental models, software and relevant equipment and including their sources and relevant identifiers) followed by a Methods and Protocols section describing the methods using a step-by-step protocol format. The aim is to facilitate adoption of the methodologies across labs. More information on how to adhere to this format as well as a downloadable template (.docx) for the Reagents and Tools Table can be found in our author guidelines: <https://www.embopress.org/page/journal/17574684/authorguide#structuredmethods>.

An example of a paper with Structured Methods can be found here: <https://www.embopress.org/doi/10.15252/msb.20178071>.

**** Reviewer's comments ****

Referee #1 (Comments on Novelty/Model System for Author):

The study is very detailed and presents novel findings that are clinically relevant. Another strength is that the findings from the animal model are accompanied by a translational study using intestinal tissue from patients who have undergone abdominal surgery for pancreatectomy.

Referee #1 (Remarks for Author):

This manuscript reports on abdominal surgery-induced activation of enteric nerves and the involvement of resident macrophages in surgery-induced neurodegeneration in a post-operative ileus model in mice. The animal results are accompanied by a study of enteric neurodegeneration in jejunal samples taken at early and late stages of surgery from patients undergoing pancreatectomy.

The study is very detailed and presents novel and interesting results. The manuscript is well written and the results are clearly presented and discussed. On reading the paper, the following comments were made which may improve the manuscript:

The legend to Figure 1 mentions that Student's t-test was used for statistical analysis of the data but a Student t-test is not allowed when comparing more than two groups. The same comment applies to the results shown in Figure 2. Please explain.

Statistical analysis page 32 mentions that some "experiments were repeated with more samples when the result was close to statistical significance". This is an odd statement because it is not clear whether the number n of mice was increased or whether additional tissue samples from the same animal were included. Also, when experiments are replicated, one would expect all groups to be replicated: for example, Figure 6c to 6f shows a significant difference in n in the groups compared, so I wonder if in this case only the groups that were close to the significant difference were replicated. Please explain.

Following the above comment, the authors could include the exact number n of mice in the figures. I am aware that the symbols in the figures refer to the number n, but many symbols overlap, making it impossible to deduce the exact number n of each experimental group. For example, in Figures 1c', 2b and 4b I cannot deduce from the symbols how many mice were included in the controls, IM3h and IM24h groups.

Figure 1f and 4f: Gene ontology analysis, enrichment score: are these results presented as an enrichment against the controls? Please clarify.

Page 26: Which manipulation did the sham mice receive? Were control mice also treated with analgesics? What is the effect of Tramadol on inflammatory and other (i.e. intestinal transit) outcome parameters?

Methods page 27 mentions that mouse colon samples were taken for FACS analysis but I did not find results from the colon in the manuscript

Page 13, line 14 from bottom: the close proximity of macrophages and neurons does not really prove that there is a direct exchange of signals between both cell types, so I would rewrite this statement.

Is there a reason why only male mice were used for the intestinal manipulation animal model while male and female mice were used for the macrophage depletion experiments (methods, page 26)

Referee #2 (Comments on Novelty/Model System for Author):

The novelty of the current study is low, however the manuscript could present a useful resource if revised appropriately. The study includes RNA-sequencing datasets of medium quality but these need to be confirmed on the protein level. Please see below for detailed major and minor concerns.

Referee #2 (Remarks for Author):

In their study entitled "Macrophage-induced enteric neurodegeneration leads to motility impairment during gut inflammation", Breßer and colleagues aim to better characterize the inflammatory response during the onset and progression of post-operative ileus (POI), a gastrointestinal disorder characterized by impaired intestinal motility and/or transit. During surgical manipulation of the intestine, macrophages in the muscularis externa (i.e. muscularis macrophages) initiate an inflammatory cascade of events that includes the upregulation of several transcription factors (i.e. NF- κ B, STAT etc), induction of pro-inflammatory gene expression and the release of chemokines and cytokines (i.e. IL-1 β , MCP1, IL-6, TNF α etc). This leads to the upregulation of adhesion molecules and recruitment of circulating leukocytes such as neutrophils and monocytes, which eventually contribute to impaired intestinal motility by inflammatory effects on smooth muscle and both intrinsic and extrinsic nerves. The fundamental role of macrophages in this process has been demonstrated, for example in macrophage-ablated Csf1op/op mice that are protected against POI.

In the current study, the authors aimed to further characterize the enteric nervous system in response to POI. They performed bulk RNA sequencing at 3h and 24h post intestinal manipulation and found upregulation of gene clusters that suggest the loss of neurons and activation of inflammatory responses. They further characterized synaptic and neuronal degeneration in the myenteric plexus and observed a decrease in HuC/HuD, PSD95 and ChAT expression at 24h post manipulation, suggestive of neurodegenerative response and synaptic loss. Next, the authors performed RNA sequencing on ribosomal translatomes isolated from ChATCre^{+/+}/Rpl22HA^{+/+} mice using the RiboTag approach and found upregulation of transcripts that suggest the loss of enteric neurons, i.e. genes related to cell death and neurogenesis. They further found increased macrophage numbers around myenteric neurons and SMART-Seq analysis revealed that these cells upregulate a "neurodegenerative phenotype". Further, the authors observed that depleting muscularis macrophages using anti-CSF1R prior to intestinal manipulation abrogated neuronal death (HuC/HuD counting), reduced immune cell infiltration (MPO counting) and improved gastrointestinal motility, indicating that targeting muscularis macrophages could mitigate POI symptoms and neurodegeneration. The authors conclude with the analysis of human gut samples obtained from patients undergoing pancreatectomy, observing similar genetic programs that indicate inflammation and neuronal loss.

Although the conclusions of the current study are not necessarily novel or innovative, the findings provide molecular insights into the role of muscularis macrophages and enteric neurons in POI and could represent a useful resource for the field. However, my main concern is the lack of validation of the transcriptome data in the current manuscript. For example, bulk-RNA seq experiments (Figure 1d-..) show many transcripts that can be either of neuronal (Bdnf), glial, fibroblast, macrophage origin, or even caused by tissue isolation/digestion artifacts (Fos, Arc). The authors should further acknowledge that only a minor number of observed transcriptional changes will be effectively translated and therefore biological relevant in the disease model. RNA-seq datasets in figures 3 and 4 are other examples where confirmation on the mRNA and/or protein level is needed. The authors could make use of a recent protocol established by the Pachnis lab (Francis Crick Institute, London, UK) that allows the characterization of mRNA transcripts in the muscularis externa by RNAscope (Obata Y et al., 2022; PMID 35676375) to verify expression patterns of some of their genes of interest in neurons and other cell types such as macrophages, glial cells, fibroblasts,.. Alternatively, if antibodies are available, the authors could perform immunohistochemistry to confirm observations on the protein level. Without these confirmation studies, many conclusions of the authors throughout the manuscripts are overstated and not justified.

Other major points:

- In Figure 2, the authors investigated synaptic deterioration that likely reflects enteric neurodegeneration in POI. However, the synaptic staining in the myenteric plexus is unconvincing. First, the authors show only low-resolution and zoomed-out immunofluorescence. Synapses, which are 70nm- 2µm, cannot be properly examined this way. Synapse numbers should be examined with super-resolution imaging. Second, the authors should perform a synaptic quantification. For this, they should examine at least one pair of pre and post synaptic markers. This will be a much more accurate characterization of synapse loss compared to mRNA levels (Figure 2b). Related to the suggested synapse loss, could the authors explain why many synapse-related transcripts are in fact increased in POI (Figure 1i and Figure 3e)?
- On page 13 and related to Figure 4, the authors claim that muscularis macrophages cause enteric neurodegeneration in POI. This is presumable, considering the well-described inflammatory roles of muscularis macrophages in POI, however it is not convincingly shown by the authors in the current manuscript. Gene ontology terms are not sufficient to conclude that macrophages are involved in neurodegeneration (Figure 4f). First, CD68 is not necessarily a marker of inflammation, but is a lysosomal marker in myeloid cells including gut macrophages and microglia. Are there alternative ways to show that the muscularis macrophages become 'inflammatory'? For example, intracellular FACS staining for cytokines or FACS sorting muscularis macrophages and performing qPCR for inflammatory markers? Alternatively, the authors could check whether the muscularis macrophages phagocytose more synapses (PMID: 37316669) or have engulfed apoptotic neurons. Second, it is well known that resident and infiltrating macrophages have different roles in the progression of POI (PMID: 28615302 and others). The authors acknowledge this in Figure 4 by FACS sorting and sequencing infiltrating monocytes vs resident muscularis macrophages, based on Ly6C expression. However, this discrimination is not mentioned or considered in their further analyses, except for a single heatmap (Extended Figure 4d)? This is confusing to me. Do infiltrating vs resident macrophages have different transcriptomes, and if so, could the authors confirm this on the gene expression (RNAscope) or protein (IHC or flow cytometry) level? I could have missed this but there is also no list, heatmap or volcano plot provided with differentially expressed genes, other than the 9 genes mentioned in Figure 4g/h? Again, are these expression levels from the infiltrating or resident macrophages? Please clarify.
- In Figure 5, the authors investigated whether depletion of muscularis macrophages prevented neuronal loss after intestinal manipulation. They use a single injection of anti-CD115 antibodies to deplete resident muscularis macrophages, however this does not prevent the influx of monocytes into the intestine that will replace the empty niche of the previously depleted muscularis macrophages. As mentioned earlier and noted by the authors in the introduction, there is a clear functional difference in resident muscularis macrophages and infiltrating monocytes in the progression of POI. Hence, it would be of great interest if the authors could test the effect of blocking resident vs infiltrating monocytes in their model. This could be achieved by anti-CCR2 administration (monoclonal antibody MC-21), that will prevent the infiltration of monocytes. Via the administration of either anti-CD115, anti-CCR2 or anti-CD115 in combination with anti-CCR2 the authors could test the effect of macrophage depletion (anti-CD115), monocyte depletion (anti-CCR2) or muscularis macrophage depletion with prevention of monocyte infiltration (anti-CD115/CCR2), respectively. In line, Figure 5b, right panel, shows muscularis macrophages with limited extensions and ramifications, likely to be monocytes.

Minor remarks/ suggestions:

- As mentioned earlier, there are many overstatements and over-interpretations in the current manuscript. A few examples include:
 - Page 11: "The neuron-specific transcriptome analysis confirmed POI-induced enteric neurodegeneration and suggested a direct interaction between enteric neurons and their inflammatory environment"
There is no experimental data in this manuscript that suggests the direct interaction of enteric neurons with their environmental niche.
 - Title of Figure 4: "Inflammatory macrophages are causing enteric neurodegeneration"
There is no data in this figure that suggests that macrophages are the cause of enteric neuronal degeneration.
 - Page 13: "Neurodegeneration in the CNS is caused by reactive microglia"
Microglia are increasingly recognized to be involved in the pathogenesis of several CNS disorders, but not all neurodegenerative processes are caused by reactive microglia. Please use tone down this statement. Also, what does 'reactive' mean?
 - Page 12: "At IM24h, the ganglia-associated macrophages assembling around ANNA1+ enteric neurons strongly increased (Figure 4a+b and S4a), suggesting a direct exchange between both cell types."
An increase in macrophage numbers does necessarily mean a direct exchange between cell types. Of note, what do the authors mean with 'exchange'?
- The authors could reference more accurate papers in sections of their manuscript. Few examples:
 - Page 3: "Recent advances expanded this view by investigating the role of enteric glial cells, a neurosupportive cell type of the ENS, revealing their impact on disease development (16) and their essential part in the inflammatory response of the gut (17-

20)" Please include PMID: 34671159.

-Page 13: Please include the reference for Cx3cr1gfp/+ mice: PMID:10805752

-Page 16 and Figure 5. Please include reference PMID:25036630. Muller and colleagues were the first to use anti-CD115 to deplete muscularis macrophages.

-Page 8: "Synapsins (Syn1-3), postsynaptic protein 95 (PSD95), and choline acetyltransferase (ChAT) are essential proteins participating in synaptic function (28)"

This reference does not demonstrate that Syn1-3, PSD95 and ChAT are essential proteins in synaptic function.

- The authors are interested in how altered neuroimmune communication between muscularis macrophages and enteric neurons could modulate the onset of POI. Is there any molecular pathway (i.e. cytokine, chemokine, neuropeptide,..) that appears in their ribotag data (Figure 3) that could reflect altered macrophage-enteric neuron communication? This might be of great interest to follow up.

- Could the authors explain or speculate on the variation observed in the control Ribotag samples in Figure 3b?

- Please mention in the figure legends that the myenteric plexus was investigated, this is more accurate as compared to 'enteric neurons'.

- Please include adjusted P-values in GO term analyses throughout the manuscript.

- Page 13 and Figure 4: Can the authors more clearly describe what they mean with macrophages 'close to ganglia'? Are these outside the ganglia or within the ganglia? Is there a certain distance the authors consider? How is this calculated?

- Could the authors clearly state in the figure legends which statistical tests they have used? I am not a biostatistician, but I believe that the 1-way ANOVA is the correct statistical test for Figures 1c, 1j, 1k, 2b, 2c', 2d" etc where multiple groups are compared with 1 variable. However, in the figure legend the Student's t-test is mentioned.

- Could the authors more clearly describe how they obtained the gene set on page 18? "Similar to the murine data sets, we detected significant upregulation of genes associated with cell death (BCL2, BCL2A1, CASP3, CASP4)," It is unclear where these genes come from.

Minor corrections/ adaptations of the main text:

-Page 5: "These data indicate a simultaneous induction of neuronal apoptotic cell death and proliferative programs."

Proliferation of enteric neurons in the adult mouse is controversial, please tone down this statement, unless backed with experimental data (for example, BrdU incorporation). This also applies to page 23 in the discussion: "Compensation of the enteric neuron loss during POI by generation of new neurons is very likely, as we found induction of neuronal proliferation and neurogenesis gene patterns in our sequencing data of the diseased mice" These are statements that need to be supported by experimental evidence. The observation of some upregulated genes related to neurogenesis and proliferation is not sufficient.

-Page 5: "Neuroinflammation in the gut involves and is regulated by various cell types (25)".

Neuroinflammation is somewhat a vague concept that has been introduced in the 1990s to define the pathological mechanisms in multiple sclerosis, but has now been adapted to many diseases that involve the central nervous system and microglia. In this context, what do the authors mean with 'neuroinflammation'? Is there involvement of the adaptive immune system in POI? If not sure, please use the term 'inflammation'.

-Page 5: "Our analysis focused on two time points, 3h, and 24h, after the intestinal manipulation, resulting in data disease onset and manifestation, respectively".

-Page 14: "These findings indicate a high production of "neuro-destructive" factors in close proximity to enteric neurons"

Please use another term and tone down this statement. In fact, the suggested cytokines are not necessarily "neuro-destructive". For example, IL6 regulates regulatory T cell differentiation to maintain homeostasis with the microbiota (PMID: 33691135).

-Page 16: "We quantified the neuron numbers after IM24h and, agreeing with our hypothesis, found more ANNA1+ neurons in the ME of CD115-depleted animals, indicating less neurodegeneration"

It is a bit odd to say that there are more neurons, as this indicates proliferation? I suggest to change: "We observed decreased loss of neurons".

-Page 23: "In PD, for example, enteric neurodegeneration has been introduced to describe synaptic damage and neuron death in the ENS (77) corresponding to the CNS phenotype (85), discovering novel similarities between the injured brain and gut"

Please tone down, enteric neuronal loss has not been clearly defined in PD and different studies report contradicting results. Conclusions from one paper using the Rotenone model (reference #77) should not be used to make a general statement of

neuronal loss in PD.

Referee #3 (Comments on Novelty/Model System for Author):

The study by Breber et al. entitled "Macrophage-induced enteric neurodegeneration leads to motility impairment during gut inflammation" examines early transcriptional changes in enteric neurons and muscularis macrophages based on transcriptional profiling of both cell types at two time points after induction of post-operative ileus (POI). Some of the findings are validated in a model of macrophage depletion in vivo. Although conceptually the study is not particularly novel, because the relationship between myenteric neurons and muscularis macrophages during POI have been studied in the past, including this group, this study provides a comprehensive analysis of gene expression datasets of both neurons and macrophages to get a better mechanistic understanding of the relationship between two cell types. Although the study is viewed as significant, some weaknesses came to the attention of this reviewers.

Major conceptual weaknesses:

It remains unclear what is the most upstream signal that drives myenteric plexitis after surgical manipulations. The authors generated comprehensive gene expression datasets at early (3 hrs) and late (24 hrs) time points. For neurons, they show the analysis for both time points, but for macrophage responses, they focus on the late response (24 hrs). Why? Is it because the 3 hr time point is not as impressive or significant? If this is the case, it seems to be a missed opportunity to understand "the chicken and the egg" relationship between two cell types. Muscularis macrophages are present within the myenteric plexus and not exposed to the abdominal environment. Do abdominal manipulations activate peritoneal macrophages or serosal macrophages which then transmit the signal to enteric neurons and then neurons pass it to muscularis macrophages? Peritoneal macrophages are likely to be affected by CD115-antibody treatment. Can it be explored, e.g., by administering the antibody i.v. vs i.p. and by looking at early time points more closely? Either outcome of this analysis will be informative for our understanding of the POI pathogenesis.

Other points:

"A prominent example of an acute neuroinflammatory disease in the gut is postoperative ileus (POI)". - It seems unappropriated to call POI a disease. It is not an independent clinical entity, but a pathological condition that develops as a complication of surgery.

Fig. 1c and all other images showing myenteric ganglia: show lower magnification images to demonstrate whether there is any heterogeneity in neuronal activation among different ganglia and fields. This will be very informative for our understanding of POI pathogenesis.

All figures: no quantification of neuronal loss or cFos+ neurons, apoptotic neurons etc. This would be informative to understand the overall impact of intestinal manipulation on neuronal activation, death etc, and also on heterogeneity of the response across ganglia as mentioned above.

Fig. 2d: CC3 staining is not convincing based on the images provided.

Page 11, final line: "suggested a direct interaction between enteric neurons and their inflammatory environment". - it is not clear how the authors arrived to this conclusion based on the data that they provide.

Page 13, middle: "suggesting a direct exchange between both cell types". - the conclusion is unclear. Exchange of what? Based on what was it concluded?

Page 14, last sentence: "Our results point out that activated resident macrophages create and maintain a neurodegenerative environment by causing inflammatory and metabolic stress for enteric neurons after POI induction, altering their gene expression profiles, and inducing neuronal death programs". - it should be clarified that the data are suggestive or predictive rather than confirmatory. The wording makes it sound like it is a statement.

Fig. 4d: is quantification of macrophages automated or manual?

Page 16: "Accordingly, expression levels of prominent POI disease genes (Arg1, Egr1, Ccl2, Il6, (17, 19)) showed less upregulation at IM24h in depleted animals and no changes in controls (Figure S5f)". - What about n3 hrs POI? The peak of neuronal loss is at 3 hrs but the data for macrophages are shown at 24 hrs. Is there a reason for that?

Fig. 6: what types of jejunal biopsies are shown - transmural or mucosal + submucosal? When showing neurons (Fig. 6b) what are we looking at - myenteric neurons or submucosal neurons?

In all figure legends, it should be clearly stated what region of the gut is being analyzed. Do the authors always look at the same region?

Referee #3 (Remarks for Author):

The study by Breber et al. entitled "Macrophage-induced enteric neurodegeneration leads to motility impairment during gut inflammation" examines early transcriptional changes in enteric neurons and muscularis macrophages based on transcriptional profiling of both cell types at two time points after induction of post-operative ileus (POI). Some of the findings are validated in a model of macrophage depletion in vivo. Although conceptually the study is not particularly novel, because the relationship between myenteric neurons and muscularis macrophages during POI have been studied in the past, including this group, this study provides a comprehensive analysis of gene expression datasets of both neurons and macrophages to get a better mechanistic understanding of the relationship between two cell types. Although the study is viewed as significant, some weaknesses came to the attention of this reviewers.

Major conceptual weaknesses:

It remains unclear what is the most upstream signal that drives myenteric plexitis after surgical manipulations. The authors generated comprehensive gene expression datasets at early (3 hrs) and late (24 hrs) time points. For neurons, they show the analysis for both time points, but for macrophage responses, they focus on the late response (24 hrs). Why? Is it because the 3 hr time point is not as impressive or significant? If this is the case, it seems to be a missed opportunity to understand "the chicken and the egg" relationship between two cell types. Muscularis macrophages are present within the myenteric plexus and not exposed to the abdominal environment. Do abdominal manipulations activate peritoneal macrophages or serosal macrophages which then transmit the signal to enteric neurons and then neurons pass it to muscularis macrophages? Peritoneal macrophages are likely to be affected by CD115-antibody treatment. Can it be explored, e.g., by administering the antibody i.v. vs i.p. and by looking at early time points more closely? Either outcome of this analysis will be informative for our understanding of the POI pathogenesis. A possible biological relevance of this inflammatory pathway should be also discussed.

Other points:

"A prominent example of an acute neuroinflammatory disease in the gut is postoperative ileus (POI)". - It seems unappropriated to call POI a disease. It is not an independent clinical entity, but a pathological condition that develops as a complication of surgery.

Fig. 1c and all other images showing myenteric ganglia: show lower magnification images to demonstrate whether there is any heterogeneity in neuronal activation among different ganglia and fields. This will be very informative for our understanding of POI pathogenesis.

All figures: no quantification of neuronal loss or cFos+ neurons, apoptotic neurons etc. This would be informative to understand the overall impact of intestinal manipulation on neuronal activation, death etc, and also on heterogeneity of the response across ganglia as mentioned above.

Fig. 2d: CC3 staining is not convincing based on the images provided.

Page 11, final line: "suggested a direct interaction between enteric neurons and their inflammatory environment". - it is not clear how the authors arrived to this conclusion based on the data that they provide.

Page 13, middle: "suggesting a direct exchange between both cell types". - the conclusion is unclear. Exchange of what? Based on what was it concluded?

Page 14, last sentence: "Our results point out that activated resident macrophages create and maintain a neurodegenerative environment by causing inflammatory and metabolic stress for enteric neurons after POI induction, altering their gene expression profiles, and inducing neuronal death programs". - it should be clarified that the data are suggestive or predictive rather than confirmatory. The wording makes it sound like it is a statement.

Fig. 4d: is quantification of macrophages automated or manual?

Page 16: "Accordingly, expression levels of prominent POI disease genes (Arg1, Egr1, Ccl2, Il6, (17, 19)) showed less upregulation at IM24h in depleted animals and no changes in controls (Figure S5f)". - What about n3 hrs POI? The peak of neuronal loss is at 3 hrs but the data for macrophages are shown at 24 hrs. Is there a reason for that?

Fig. 6: what types of jejunal biopsies are shown - transmural or mucosal + submucosal? When showing neurons (Fig. 6b) what are we looking at - myenteric neurons or submucosal neurons?

In all figure legends, it should be clearly stated what region of the gut is being analyzed. Do the authors always look at the same region?

Breßer and Siemens et al., Point by Point

We thank the reviewers for their detailed comments and hope our revision will answer their questions and discuss all open points adequately. Major attention and efforts have been spent on new experiments, allowing us to substantiate our hypotheses and conclusions.

Please find our point-by-point response in the following part:

Referee #1 (Remarks for Author):

This manuscript reports on abdominal surgery-induced activation of enteric nerves and the involvement of resident macrophages in surgery-induced neurodegeneration in a post-operative ileus model in mice. The animal results are accompanied by a study of enteric neurodegeneration in jejunal samples taken at early and late stages of surgery from patients undergoing pancreatectomy.

The study is very detailed and presents novel and interesting results. The manuscript is well written and the results are clearly presented and discussed. On reading the paper, the following comments were made which may improve the manuscript:

We thank the reviewer for the comments and hope our response to all questions is satisfying.

1) The legend to Figure 1 mentions that Student's t-test was used for statistical analysis of the data but a Student t-test is not allowed when comparing more than two groups. The same comment applies to the results shown in Figure 2. Please explain.

As we only compared 2 groups (control with one disease time point or the disease points with each other) we thought a simple Student *t*-test was adequate. Nevertheless, we now validated the data of Figures 1 and 2 with a one-way ANOVA test to ensure the significance of our data. We changed the figure legends and included the new significance levels of the ANOVA test accordingly.

2) Statistical analysis page 32 mentions that some "experiments were repeated with more samples when the result was close to statistical significance". This is an odd statement because it is not clear whether the number *n* of mice was increased or whether additional tissue samples from the same animal were included.

Importantly, all our *n*-numbers correspond to the number of mice used in the study, not to multiple tissue samples from the same mouse.

The mentioned sentence describes our procedure in the case of *in vitro* studies, where we would perform additional experiments to clarify if a tendency in the data can result in a clear phenotype. By mistake, we left this sentence in the manuscript from a previous version of our study. We deleted it from our manuscript and apologize for the misunderstanding.

3) Also, when experiments are replicated, one would expect all groups to be replicated: for example, Figure 6c to 6f shows a significant difference in *n* in the groups compared, so I wonder if in this case only the groups that were close to the significant difference were replicated. Please explain.

In Figure 6, we used only patient samples collected early and late during surgery for all plots. In Figure 6c-f, we present the data as fold changes between the late and the early specimens. For all plots, the same number of data points is shown; however, the values are sometimes very close together and overlap. Therefore, we included the corresponding *n*-

numbers in the figure legends for the reader's convenience. Additionally, we included all the raw data files in this submission to enable cross-referencing.

4) Following the above comment, the authors could include the exact number n of mice in the figures. I am aware that the symbols in the figures refer to the number n , but many symbols overlap, making it impossible to deduce the exact number n of each experimental group. For example, in Figures 1c', 2b and 4b I cannot deduce from the symbols how many mice were included in the controls, IM3h and IM24h groups.

See also Question 3.

We agreed with the reviewer and included all n -numbers for mice and patient samples in the corresponding figure legends.

5) Figure 1f and 4f: Gene ontology analysis, enrichment score: are these results presented as an enrichment against the controls? Please clarify.

For the gene ontology analysis, we used all significantly induced genes between control mice and mice from the indicated disease time points. These lists of genes were used to apply the gene ontology analysis. Therefore, the results of Figures 1f and 4f indeed present the enrichment of gene expression in the experimental group against the respective controls. We included some additional information to clarify our procedure for this analysis.

“To understand these neuronal changes, we performed gene ontology (GO) analysis of DEGs in mice undergoing surgery at specific time points compared to their control (naïve) mice ...”

6) Page 26: Which manipulation did the sham mice receive? Were control mice also treated with analgesics? What is the effect of Tramadol on inflammatory and other (i.e. intestinal transit) outcome parameters?

In this study, we have compared POI mice to naïve animals that did not receive analgesics (Tramadol). One of our previous studies showed that sham mice (only laparotomy with analgesics (Tramadol), without intestinal manipulation) develop a mild inflammatory phenotype with a minor change in GI-transit time and almost no leukocyte infiltration (PMID: 37941007). Therefore, we decided to use naïve mice for this study. We are aware that sham mice already differ from naïve mice, but the difference is neglectable compared to mice suffering from POI. In a comparative study (PMID: 30907232), our group investigated different analgesics and showed that Tramadol is preferable and did not alter cytokine levels significantly.

7) Methods page 27 mentions that mouse colon samples were taken for FACS analysis but I did not find results from the colon in the manuscript

In supplementary figure S5c, we used colon ME tissue for the FACS analysis to quantify the depletion of macrophages. This is mentioned in the figure legend and now also shortly in the text. As the small intestinal tissue was used for qPCR and histological analysis, we had to resort to colonic tissue from the same mice to ensure matching quantification. Since macrophage depletion is systemically applied, we do not expect any differences between the small and large intestine with regard to the depletion efficacy.

8) Page 13, line 14 from bottom: the close proximity of macrophages and neurons does not really prove that there is a direct exchange of signals between both cell types, so I would rewrite this statement.

We agree with the reviewer and rephrased that statement.

“At IM24h, the ganglia-associated macrophages assembling around ANNA1+ enteric neurons significantly increased (Figure 4a+b and EV4a), suggesting a possible interaction between both cell types.”

9) Is there a reason why only male mice were used for the intestinal manipulation animal model while male and female mice were used for the macrophage depletion experiments (methods, page 26)

For the depletion experiments, we only used CX3CR1-GFP transgenic mice. As we needed many mice with the same age and genotype, we used female mice to establish the correct conditions for the depletion (antibody concentration and time points). For the POI experiments, with and without depletion, we only used male Cx3xr1^{gfp/+} and for this revision neuronal *RiboTag* mice. By this, we could reduce the number of needed mice, according to the animal laws in Germany.

We included these details in the methods part to prevent any confusion.

“...and mice of both sexes (10-20 weeks) were used for macrophage depletion experiments (female mice only for establishing the depletion method, while only male mice were used for POI experiments) carried out under German federal law (AZ.: 81 02.04.2020.A357).“

Referee #2 (Comments on Novelty/Model System for Author):

The novelty of the current study is low, however the manuscript could present a useful resource if revised appropriately. The study includes RNA-sequencing datasets of medium quality but these need to be confirmed on the protein level. Please see below for detailed major and minor concerns.

We presented our results at several international meetings, including this year's and last year's Digestive Disease Week (DDW), one of the largest conferences for gut research. From the general program and discussions, we had the clear impression that enteric neurodegeneration is quite an important topic and not only relevant to people working in the ENS field but also to those interested in intestinal acute and chronic inflammation. Related to the particular topic of POI, it is indeed known that local neuroinflammation is responsible for the resulting functional motility disturbances but the detection of neuronal loss, synaptic degradation as well as the control of this neurodegeneration by macrophages is to the best of our knowledge unknown. Therefore, we disagree that the novelty of our study is low. Nevertheless, we felt encouraged by the substantial number of detailed remarks made by this reviewer, including a series of new experiments and a multitude of changes throughout the manuscript to address all points raised by reviewer 2. We hope our responses are satisfying and the reviewer agrees that the efforts spent and the numerous additional experiments improved the manuscript, making it suitable for publication.

Referee #2 (Remarks for Author):

In their study entitled "Macrophage-induced enteric neurodegeneration leads to motility impairment during gut inflammation", Breßer and colleagues aim to better characterize the inflammatory response during the onset and progression of post-operative ileus (POI), a gastrointestinal disorder characterized by impaired intestinal motility and/or transit. During surgical manipulation of the intestine, macrophages in the muscularis externa (i.e. muscularis macrophages) initiate an inflammatory cascade of events that includes the upregulation of several transcription factors (i.e. NF- κ B, STAT etc), induction of pro-inflammatory gene expression and the release of chemokines and cytokines (i.e. IL-1 β , MCP1, IL-6, TNF α etc). This leads to the upregulation of adhesion molecules and recruitment of circulating leukocytes such as neutrophils and monocytes, which eventually contribute to impaired intestinal motility by inflammatory effects on smooth muscle and both intrinsic and extrinsic nerves. The fundamental role of macrophages in this process has been demonstrated, for example in macrophage-ablated *Csf1op/op* mice that are protected against POI.

In the current study, the authors aimed to further characterize the enteric nervous system in response to POI. They performed bulk RNA sequencing at 3h and 24h post intestinal manipulation and found upregulation of gene clusters that suggest the loss of neurons and activation of inflammatory responses. They further characterized synaptic and neuronal degeneration in the myenteric plexus and observed a decrease in HuC/HuD, PSD95 and ChAT expression at 24h post manipulation, suggestive of neurodegenerative response and synaptic loss. Next, the authors performed RNA sequencing on ribosomal translatomes isolated from ChATCre^{+/+}/Rpl22HA^{+/+} mice using the RiboTag approach and found upregulation of transcripts that suggest the loss of enteric neurons, i.e. genes related to cell death and neurogenesis. They further found increased macrophage numbers around myenteric neurons and SMART-Seq analysis revealed that these cells upregulate a "neurodegenerative phenotype". Further, the authors observed that depleting muscularis macrophages using anti-CSF1R prior to intestinal manipulation abrogated neuronal death (HuC/HuD counting), reduced immune cell infiltration (MPO counting) and improved gastrointestinal motility, indicating that targeting muscularis macrophages could mitigate POI symptoms and neurodegeneration. The authors conclude with the analysis of human gut

samples obtained from patients undergoing pancreatectomy, observing similar genetic programs that indicate inflammation and neuronal loss.

Although the conclusions of the current study are not necessarily novel or innovative, the findings provide molecular insights into the role of muscularis macrophages and enteric neurons in POI and could represent a useful resource for the field.

1) However, my main concern is the lack of validation of the transcriptome data in the current manuscript. For example, bulk-RNA seq experiments (Figure 1d-..) show many transcripts that can be either of neuronal (Bdnf), glial, fibroblast, macrophage origin, or even caused by tissue isolation/digestion artifacts (Fos, Arc).

We appreciated the comment that transcriptional data needs validation and included transcriptional results by an unbiased proteomics analysis (Figure 1j-k, EV1g+h, 2c+g, EV2d+g). We included a mass spectrometry analysis of the *muscularis externa* of control and POI mice at the discussed disease time points. Therein, we gained insight into expressional changes on protein level after inducing POI.

We understand the reviewer's concern about the cellular origin of transcripts in our bulk-RNA sequencing experiments. However, we believe this approach is a useful method for gleaning a general overview of the transcriptional changes related to neuronal molecular pathways after intestinal manipulation. Furthermore, we deepened our understanding of neurons and macrophages, using neuronal *RiboTag* data in Figure 3 and Smart-Seq2 bulk mRNAseq data from sorted macrophage populations in Figure 4, pinpointing the origin of those transcripts by cell-specific analyses. Broader approaches, including single-cell analysis and single nuclei RNA sequencing, might, of course, increase the amount of transcriptional input and identify the cellular sources within all cells. However, these single-cell-based sequencing techniques regularly result in reduced numbers of detected genes. Furthermore, due to the immediate induction of neuronal death, the number of identified neuronal transcripts might be even more reduced than it is any way by this technique. We, therefore, believe that the *Ribotag* approach gives enough insight into enteric neurons to claim that neurons undergo a massive degenerative process.

2) The authors should further acknowledge that only a minor number of observed transcriptional changes will be effectively translated and therefore biological relevant in the disease model.

We are not sure how reviewer 2 knows that only a "minor number" of the documented transcriptional changes are, in the end, biologically relevant. Nevertheless, we validated the changes in the transcript levels by a corresponding unbiased mass spectrometry analysis and detected similar enriched signaling pathways (See Figure 1k+EV1h+S1e) and functions with a clear down-regulation in synaptic proteins (see Figure 2c), indicating that the mRNA expression changes are translated to the protein level. We included these new results in our manuscript and discussed them accordingly.

Enriched Pathways after IM

Enriched Pathways after IM

3) RNA-seq datasets in figures 3 and 4 are other examples where confirmation on the mRNA and/or protein level is needed. The authors could make use of a recent protocol established by the Pachnis lab (Francis Crick Institute, London, UK) that allows the characterization of mRNA transcripts in the muscularis externa by RNAscope (Obata Y et al., 2022; PMID 35676375) to verify expression patterns of some of their genes of interest in neurons and other cell types such as macrophages, glial cells, fibroblasts, ..

See also Question 1+2.

To validate changes on the protein level, we also performed mass spectrometry analysis of the *muscularis externa* of control and POI mice at the disease time points that were previously discussed (see Figures 1 and 2). We are currently interested in characterizing enteric neurodegeneration as a general phenotype after surgery, and no specific targets have been chosen for in-depth analysis. We do not believe that the underlying molecular mechanisms differ between the enteric neuronal subpopulations due to the simultaneous and sheer amount of neuronal and synaptic loss. Nevertheless, studies about distinct pathways might be interesting for research interested in the biology of specific neuronal subpopulations, but they go far beyond the scope of our study.

Concerning the specificity of our transcriptional data, we think that the neuronal *RiboTag* data in Figure 3 and the Smart-Seq2 data in Figure 4 from sorted macrophage populations is an excellent starting point. RNAscope is a valuable technique to validate the location of certain expression changes, but it would still generate mRNA data without confirming any changes on the protein level. As we did not claim a certain role of individual mediators but rather identified overarching processes of neurodegeneration, we also do not see a benefit for this study (over the *Ribotag* approach), when including spatial expression patterns of a few genes by RNAscope.

4) Alternatively, if antibodies are available, the authors could perform immunohistochemistry to confirm observations on the protein level. Without these confirmation studies, many conclusions of the authors throughout the manuscripts are overstated and not justified.

To confirm our transcriptional data, we included data from a mass spectrometry analysis from *muscularis externa* samples of control and POI mice. As we are not claiming any detailed molecular mechanism, we are unsure what specific molecules we should target at the moment. However, we performed new experiments including immunohistochemistry, FACS, and transcriptional analysis to strengthen our conclusions, which are listed in this short overview:

- I. Enteric neurons are activated, damaged/die during POI: Transcriptional data from total tissue and *Ribotag* mice, IHC for cFOS, ANNA1 and cleaved-Casp3 in whole mounts (WMs) and swiss roles (new data), Blood plasma analysis for increase of neuronal markers as a sign of neurodegeneration (new data).

- II. Synaptic damage/dysregulation: Transcriptional data from total tissue and *Ribotag* mice, IHC for synaptic proteins in WMs, and Swiss roles (new data) with quantification (new data).
- III. Resident macrophages develop an inflammatory phenotype in POI: IHC of WMs, FACS analysis (new data), and Transcriptional data from sorted cells.
- IV. Resident macrophages develop a neurodegenerative profile and are one driver of enteric neurodegeneration: Transcriptional data from sorted cells, macrophage-depletion experiments in POI with validation of neuronal numbers and synaptic gene expression, usage of CCR2-KO mice in the POI model (new data), FACS data for phagocytosis of neurons in POI (new data), Transcriptional data from neuronal *Ribotag* mice after macrophage depletion (new data).

Other major points:

5) In Figure 2, the authors investigated synaptic deterioration that likely reflects enteric neurodegeneration in POI. However, the synaptic staining in the myenteric plexus is unconvincing. First, the authors show only low-resolution and zoomed-out immunofluorescence. Synapses, which are 70nm- 2µm, cannot be properly examined this way.

To get a professional opinion on our work about synaptic damage, we contacted Prof. Anja Schneider from the DZNE in Bonn. She and her group are experts in neurodegeneration in the brain, mainly in the context of Alzheimer's Disease. Seeking their advice on detecting and quantifying neurodegeneration, they recommended additional antibodies (targets: Synapsin1 and postsynaptic density protein 95) for quantifying synaptic damage. For the brain, their strategy is to compare the intensity of the staining and the co-localization of both markers to quantify synaptic structures. Following their recommendation, we adopted this procedure for the gut with the same antibodies/synaptic markers and performed immunohistochemistry in WMs and Swiss roles of control and diseased mice (Figure 2a and EV4I).

Additionally, we established a common staining for neuronal morphology for gut tissue, the Golgi-cox staining (PMID 28447990). This method highlights the neurite processes of stained cells, allowing the complexity of dendritic branching to be visualized. Adapting the protocol commonly used in CNS research to the intestine was complicated and took much time, so we limited the analysis to only comparing control and IM24h gut samples. Those analyses provided important new data, as we detected reduced dendritic branching and fewer

neuronal structures in mice after surgery, supporting our other results on the morphological levels. We included images of the Golgi-Cox staining as proof of concept in our manuscript (Figure EV2b).

Moreover, Anja Schneider's group used a novel blood plasma analysis method using the highly sensitive Quanterix technique (see also methods part) to detect neuro- and astroglia degeneration in mice by quantifying markers such as GFAP, neurofilament (NfL), and amyloid beta (Ab40, Ab42) in mice suffering from Alzheimer's disease (AD). To validate the neurodegeneration detected in our POI mouse model, we collected blood plasma samples from control and POI mice and measured serum levels of GFAP, Ab40+42, and Nfl by this technique. We detected elevated NfL levels in the plasma of POI mice compared to naïve animals (Figure 2i) but did not find any differences in Ab40+42 and GFAP (Supplementary Figure S2d). We included the measurement in our manuscript.

6) Synapse numbers should be examined with super-resolution imaging. Second, the authors should perform a synaptic quantification. For this, they should examine at least one pair of pre and post synaptic markers. This will be a much more accurate characterization of synapse loss compared to mRNA levels (Figure 2b).

Please see also Question 5.

Super-resolution imaging is a very interesting technique, and the images that have been published are spectacular. Sadly, this imaging method has not been established for murine gut tissue yet, and after reviewing some publications from experts in this field, we did not see any feasibility of establishing this technique in our lab or obtaining a new collaborator in the time frame of this revision. As mentioned above, we quantified synapses and included them in the results (Figure 2+EV4, see above), but performing/establishing a completely new method for the gut was not an option. Notably, we had already discussed this point with the editor and agreed that the super-resolution imaging technique exceeded the revision tasks.

7) Related to the suggested synapse loss, could the authors explain why many synapse-related transcripts are in fact increased in POI (Figure 1i and Figure 3e)?

Unfortunately, we do not have a definite answer for that. However, we speculate that this phenotype could be defined as "counterregulation" of synaptic gene expression in POI. On the one hand, we see a synaptic loss on the protein level for prominent targets like synapsins, which could be a compensatory mechanism after the damage. On the other hand,

the up-regulation could be connected to the occurring neuroinflammation, which triggers the induction of certain synaptic genes in the gut, there is no direct evidence yet available. Our data only clearly state that there are strong changes in the synaptic gene expression after surgery and that it arises in parallel with the POI hallmarks.

8) On page 13 and related to Figure 4, the authors claim that muscularis macrophages cause enteric neurodegeneration in POI. This is presumable, considering the well-described inflammatory roles of muscularis macrophages in POI, however it is not convincingly shown by the authors in the current manuscript. Gene ontology terms are not sufficient to conclude that macrophages are involved in neurodegeneration (Figure 4f). First, CD68 is not necessarily a marker of inflammation, but is a lysosomal marker in myeloid cells including gut macrophages and microglia. Are there alternative ways to show that the muscularis macrophages become 'inflammatory'? For example, intracellular FACS staining for cytokines or FACS sorting muscularis macrophages and performing qPCR for inflammatory markers?

We agree with the reviewer that more data about the inflammatory activation of resident macrophages is needed to substantiate the macrophage reactivity. For additional transcriptional information, we included more data from the Smart-Seq2 analysis (Figure EV4d-h and S3c), and to better characterize the cells, we performed a new FACS experiment from the ME of IM3h and control mice to define the activation through additional inflammatory markers. The presented markers are validated with the latest literature about macrophages and microglia in disease (Figure 4e). We tried the cytokine stainings, too, but were unsuccessful, so we included cell surface markers. All new data show a clear inflammatory phenotype in resident macrophages after the induction of POI at the early IM3h time point, supporting our conclusion.

9) Alternatively, the authors could check whether the muscularis macrophages phagocytose more synapses (PMID: 37316669) or have engulfed apoptotic neurons.

We thank the reviewer for suggesting this straightforward approach. We used transgenic CX3CR1-GFP Chat-Cre Ai14-floxed mice, which have green, GFP⁺ macrophages and red, tdTomato⁺ enteric neurons, to perform these additional FACS experiments in control and POI animals and cited the suggested publication (PMID: 37316669). We could detect double-positive macrophages in control and POI mice with the FACS, showing a significant increase in this number after surgery. We saw no difference between resident and infiltrating monocyte-derived macrophages in the phagocytotic potential, but on the gene expression level, resident cells showed a far stronger expression of genes related to phagocytosis. Additionally, we visualized the tdTomato⁺/GFP⁺ macrophages by confocal microscopy (Figure S3e). We included all the data in Figure 4h+i, EV4, and the supplementary figures.

10) Second, it is well known that resident and infiltrating macrophages have different roles in the progression of POI (PMID: 28615302 and others). The authors acknowledge this in Figure 4 by FACS sorting and sequencing infiltrating monocytes vs resident muscularis macrophages, based on Ly6C expression. However, this discrimination is not mentioned or considered in their further analyses, except for a single heatmap (Extended Figure 4d)? This is confusing to me.

See also Question 12.

We apologize for the confusion. As we see a clear induction of the neurodegenerative profile in resident macrophages, we focus mainly on them rather than confuse the reader with too much information about other cells. After reviewing all comments, we realized that additional

data from the infiltrating cells might help to emphasize the specific role of resident macrophages. Consequently, we included additional data from monocyte-derived macrophages generated by the Smart-Seq2 analysis in Figure EV4i-l. Moreover, to highlight the impact of resident macrophages, we analyzed neuronal death in CCR2-KO mice that underwent intestinal manipulation (IM). The CCR2-KO mice were already used in POI studies (PMIDs: 21113155+28615302+30581430), showing that after IM, these mice have no visible monocyte infiltration into the *muscularis externa*, attenuated inflammatory reactions, and prolonged motility impairment. In this experimental setup, we compared the impact of resident macrophages on enteric neurons in POI without the participation of infiltrating cells and confirmed the dominant role of resident macrophages in neurodegeneration. We included these experiments in Figure EV4, underlining the detrimental role of resident macrophages.

11) Do infiltrating vs resident macrophages have different transcriptomes, and if so, could the authors confirm this on the gene expression (RNAscope) or protein (IHC or flow cytometry) level? I could have missed this but there is also no list, heatmap or volcano plot provided with differentially expressed genes, other than the 9 genes mentioned in Figure 4g/h? Again, are these expression levels from the infiltrating or resident macrophages? Please clarify.

See also Question 10+12.

Yes, these populations differ transcriptionally massively (See Figure EV4d), which is not an unexpected finding. To emphasize this difference, we have included additional data from the Smart-Seq2 analysis, including a PCA plot, a volcano plot, and heat maps (See Figure 4 + EV4 + supplementary data). As our data mentioned above from CCR2-KO mice showed no significant impact of the infiltrating monocytes on neurodegeneration, we further characterized the resident cells by more experiments as we were limited by time and mice and rather used our resources for the many other open questions. For example, we performed a new FACS analysis at IM3h to better define the inflammatory activation of resident macrophages.

In our first submission, figure 4g+h displays the expression from resident macrophages, while data from infiltrating cells is only shown in Figure S4d. For clarification, we highlighted the origin of the macrophage subtypes in the respective figure legends.

12) In Figure 5, the authors investigated whether depletion of muscularis macrophages prevented neuronal loss after intestinal manipulation. They use a single injection of anti-CD115 antibodies to deplete resident muscularis macrophages, however this does not prevent the influx of monocytes into the intestine that will replace the empty niche of the previously depleted muscularis macrophages. As mentioned earlier and noted by the authors in the introduction, there is a clear functional difference in resident muscularis macrophages and infiltrating monocytes in the progression of POI. Hence, it would be of great interest if the authors could test the effect of blocking resident vs infiltrating monocytes in their model. This could be achieved by anti-CCR2 administration (monoclonal antibody MC-21), that will prevent the infiltration of monocytes. Via the administration of either anti-CD115, anti-CCR2

or anti-CD115 in combination with anti-CCR2 the authors could test the effect of macrophage depletion (anti-CD115), monocyte depletion (anti-CCR2) or muscularis macrophage depletion with prevention of monocyte infiltration (anti-CD115/CCR2), respectively. In line, Figure 5b, right panel, shows muscularis macrophages with limited extensions and ramifications, likely to be monocytes.

See also Questions 10 and 11.

We agree with the reviewer that this topic is important. As mentioned above, we did experiments in CCR2-KO mice to validate enteric neurodegeneration in POI without the influence of infiltrating monocytes on the inflammatory cascade in the gut (PMIDs: 21113155+28615302+30581430). CCR2-KO mice showed the same neuronal death and synaptic damage as wt controls at IM24h (Figure EV4), indicating that resident cells are more responsible for the discovered neurodegeneration in POI. Based on these results, we decided that additional animal experiments did not help provide us with more insight. The recommended antibody treatments from Rewiever 2 were a good experimental idea, but as the CCR2-KO mice, a very good substitute for the injections of CCR2-antibody, show no improvement concerning neuronal damage and death, we think that a combination of CD115-antibody and CCR2-KO mice was not necessary anymore, because the effect should be similar to CD115-antibody treatments alone. So, we would rather save animals and resources, as the animal laws of the government demand us to do so.

Minor remarks/ suggestions:

13) As mentioned earlier, there are many overstatements and over-interpretations in the current manuscript. A few examples include:

-Page 11: "The neuron-specific transcriptome analysis confirmed POI-induced enteric neurodegeneration and suggested a direct interaction between enteric neurons and their inflammatory environment"

There is no experimental data in this manuscript that suggests the direct interaction of enteric neurons with their environmental niche.

We reworded that sentence.

"The neuron-specific transcriptome analysis confirmed POI-induced enteric neurodegeneration and suggested an impact of the inflammatory environment on enteric neuron homeostasis."

14) Title of Figure 4: "Inflammatory macrophages are causing enteric neurodegeneration"
There is no data in this figure that suggests that macrophages are the cause of enteric neuronal degeneration.

We have now included additional data further supporting our hypothesis. Nevertheless we reworded this figure title.

"Inflammatory macrophages are involved in enteric neurodegeneration"

15) Page 13: "Neurodegeneration in the CNS is caused by reactive microglia"
Microglia are increasingly recognized to be involved in the pathogenesis of several CNS disorders, but not all neurodegenerative processes are caused by reactive microglia. Please use tone down this statement. Also, what does 'reactive' mean?

We agree that microglia are not the only cause of neurodegeneration in the CNS, but we think it is important to mention that they are able to induce and extend it. For the term "reactivity", we included a new review (PMID: 33171230) in our study explaining reactivity in microglia during neurodegeneration.

We toned our statement down as follows:

"Neurodegeneration in the CNS **can be** caused by reactive microglia"

16) Page 12: "At IM24h, the ganglia-associated macrophages assembling around ANNA1+ enteric neurons strongly increased (Figure 4a+b and S4a), suggesting a direct exchange between both cell types."
An increase in macrophage numbers does necessarily mean a direct exchange between cell types. Of note, what do the authors mean with 'exchange'?

We reworded this sentence.

"At IM24h, the ganglia-associated macrophages assembling around ANNA1⁺ enteric neurons strongly increased (Figure 4a+b and EV4a), suggesting a possible interaction between both cell types."

17) The authors could reference more accurate papers in sections of their manuscript. Few examples:

-Page 3: "Recent advances expanded this view by investigating the role of enteric glial cells, a neurosupportive cell type of the ENS, revealing their impact on disease development (16) and their essential part in the inflammatory response of the gut (17-20)" Please include PMID: 34671159.

We included this reference.

18) Page 13: Please include the reference for Cx3cr1^{gfp/+} mice: PMID:10805752

We included the suggested reference.

19) Page 16 and Figure 5. Please include reference PMID:25036630. Muller and colleagues were the first to use anti-CD115 to deplete muscularis macrophages.

We included this reference.

20) Page 8: "Synapsins (Syn1-3), postsynaptic protein 95 (PSD95), and choline acetyltransferase (ChAT) are essential proteins participating in synaptic function (28)" This reference does not demonstrate that Syn1-3, PSD95 and ChAT are essential proteins in synaptic function.

We apologize for the mistake and have included the correct references (PMID 25429258+34408655) in our manuscript.

21) The authors are interested in how altered neuroimmune communication between muscularis macrophages and enteric neurons could modulate the onset of POI. Is there any molecular pathway (i.e. cytokine, chemokine, neuropeptide,..) that appears in their ribotag data (Figure 3) that could reflect altered macrophage-enteric neuron communication? This might be of great interest to follow up.

The reviewer is correct; certain pathways are highly activated in enteric neurons during POI. We are in the process of submitting a grant proposal and an additional paper on this topic.

22) Could the authors explain or speculate on the variation observed in the control Ribotag samples in Figure 3b?

We think there are two possible reasons. First, the *Ribotag* approach is a long procedure, and there could be deviations from the many steps (detailed protocol in PMID: 34939271). Secondly, the homeostatic phase and ratios of analyzed neurons might affect the transcriptome, as only actively translated mRNA from the small intestine *muscularis externa* is visualized. This could lead to a variation in the PCA plot, as the amount of specific mRNA is critical for the patterning.

23) Please mention in the figure legends that the myenteric plexus was investigated, this is more accurate as compared to 'enteric neurons'.

We included the term "myenteric plexus" in the figure legends.

24) Please include adjusted P-values in GO term analyses throughout the manuscript.

To analyze our RNA-Seq data, we use a software called *PARTEK* (<https://www.partek.com/>), an R-based tool that helps us with all the calculations and comparisons. We contacted *PARTEK* support to discuss the reviewer's question about the GO term analysis and the resulting enrichment scores. The support explained that there is no adjusted p-value for the enrichment score, as this analysis includes no multiple testing to generate gene ontology terms. Notably, for the gene lists that were used for the GO term analysis, we applied multiple testing corrections.

25) Page 13 and Figure 4: Can the authors more clearly describe what they mean with macrophages 'close to ganglia'? Are these outside the ganglia or within the ganglia? Is there a certain distance the authors consider? How is this calculated?

As mentioned in Figure 4 and EV, we estimated the size of enteric ganglia and used a box with a certain size (500px x 500px) to count all macrophages inside of this box. These cells were then named "close to ganglia" macrophages. The calculation was done by image and normalized to the area (mm²). We did not calculate any distance or define the position in more detail, as we had no other results pointing to a specific macrophage type. By screening all reviewer's comments, we realized that this analysis could be improved. Consequently, to clarify and better define macrophages in close contact to ganglia, we now included stainings and quantifications of F11R⁺ macrophages during POI in Figures 4c. In a recent publication by the Boeckxstaens group, these cells were verified to be associated with neurons in the gut

(PMID: 37316669). The quantification underlined our previously shown increase in “close to ganglia” macrophages, as “neuron-associated” macrophages are also enriched during the disease.

We think the new stainings and quantifications align well with the previous data and provide a gainful addition to our study.

26) Could the authors clearly state in the figure legends which statistical tests they have used? I am not a biostatistician, but I believe that the 1-way ANOVA is the correct statistical test for Figures 1c, 1j, 1k, 2b, 2c', 2d" etc where multiple groups are compared with 1 variable. However, in the figure legend the Student's t-test is mentioned.

As we only compared 2 groups (control with one disease time point or the disease points with each other), we thought a simple Student *t*-test was adequate. Nevertheless, we now validated the data of Figures 1 and 2 with a one-way ANOVA test to ensure the significance of our data. We changed the figure legends and included the new significance levels of the ANOVA test accordingly.

27) Could the authors more clearly describe how they obtained the gene set on page 18? "Similar to the murine data sets, we detected significant upregulation of genes associated with cell death (BCL2, BCL2A1, CASP3, CASP4)," It is unclear where these genes come from.

The mentioned genes are prominent markers for cell death. We required the gene list by reviewing various publications (PMID: 24355989, PMID: 25526085) about neuronal death under different conditions *in vivo*.

Minor corrections/ adaptations of the main text:

28) Page 5: "These data indicate a simultaneous induction of neuronal apoptotic cell death and proliferative programs."

Proliferation of enteric neurons in the adult mouse is controversial, please tone down this statement, unless backed with experimental data (for example, BrdU incorporation). This also applies to page 23 in the discussion: "Compensation of the enteric neuron loss during POI by generation of new neurons is very likely, as we found induction of neuronal proliferation and neurogenesis gene patterns in our sequencing data of the diseased mice" These are statements that need to be supported by experimental evidence. The observation of some upregulated genes related to neurogenesis and proliferation is not sufficient.

To support our statements, we included data from IM21days mice, animals that underwent intestinal manipulation 21 days before the sacrifice. There, we could no longer detect any disease hallmarks, and there was no difference in neuron numbers. So, we conclude that enteric neurodegeneration is a transient post-operative aspect of POI pathophysiology that is resolved 3 weeks after surgery with normal levels of enteric neuron numbers and no synaptic damage compared to naive control animals. We included the data in Figure 2h+I+EV2h and expanded the discussion part.

29) Page 5: "Neuroinflammation in the gut involves and is regulated by various cell types (25)".

Neuroinflammation is somewhat a vague concept that has been introduced in the 1990s to define the pathological mechanisms in multiple sclerosis, but has now been adapted to many diseases that involve the central nervous system and microglia. In this context, what do the authors mean with 'neuroinflammation'? Is there involvement of the adaptive immune system in POI? If not sure, please use the term 'inflammation'.

We were unaware of the history behind the term "neuroinflammation"; we mainly use it to highlight that we have inflammation around neuronal structures in the gut. As we have used this term already in former publications and other groups seem to concur (some examples: PMID: 30116771, 20615234, 30381426), we would like to keep but extend it with the word "intestinal" to pronounce that it relates to the intestine and not the CNS.

-Page 5: "Our analysis focused on two time points, 3h, and 24h, after the intestinal manipulation, resulting in data disease onset and manifestation, respectively".

We corrected this sentence.

"Our analysis focused on two time points, 3h, and 24h after the intestinal manipulation, generating data from the disorder onset and manifestation, respectively."

30) Page 14: "These findings indicate a high production of "neuro-destructive" factors in close proximity to enteric neurons" Please use another term and tone down this statement. In fact, the suggested cytokines are not necessarily "neuro-destructive". For example, IL6 regulates regulatory T cell differentiation to maintain homeostasis with the microbiota (PMID: 33691135).

We changed the sentence, but we would like to point out that the mentioned factors can all harm neurons (we cited studies for every factor), and there is the possibility that they can damage enteric neurons, too. The mentioned factor IL-6 has many roles in many tissues, and its function varies greatly, so a harmful role for neurons is possible, as seen here (PMID:

34058569).

31) Page 16: "We quantified the neuron numbers after IM24h and, agreeing with our hypothesis, found more ANNA1+ neurons in the ME of CD115-depleted animals, indicating less neurodegeneration"

It is a bit odd to say that there are more neurons, as this indicates proliferation? I suggest to change: "We observed decreased loss of neurons".

We agree with the reviewer and changed this sentence.

"We quantified the neuron numbers after IM24h and, in line with our hypothesis, found more ANNA1+ neurons in the ME of CD115-depleted animals, indicating a smaller decrease in neuronal numbers and less neurodegeneration."

32) Page 23: "In PD, for example, enteric neurodegeneration has been introduced to describe synaptic damage and neuron death in the ENS (77) corresponding to the CNS phenotype (85), discovering novel similarities between the injured brain and gut" Please tone down, enteric neuronal loss has not been clearly defined in PD and different studies report contradicting results. Conclusions from one paper using the Rotenone model (reference #77) should not be used to make a general statement of neuronal loss in PD.

We mentioned PD only as an example; however, we do not work in this scientific field. Some additional research showed that the discussion of enteric neuronal loss in PD is still ongoing, but there is some convincing data already available, so we cited more work on this topic. We modified this sentence to indicate that enteric neuronal loss is still under investigation in PD.

"In PD, for example, enteric neurodegeneration has been recently introduced to describe synaptic damage and neuron death in the ENS (92, 100–103) corresponding to the CNS phenotype (104); however, more studies are needed to confirm the enteric neuron phenotype and discover novel similarities between the injured brain and gut."

Referee #3 (Comments on Novelty/Model System for Author):

The study by **Breber et al.** entitled "Macrophage-induced enteric neurodegeneration leads to motility impairment during gut inflammation" examines early transcriptional changes in enteric neurons and muscularis macrophages based on transcriptional profiling of both cell types at two time points after induction of post-operative ileus (POI). Some of the findings are validated in a model of macrophage depletion in vivo. Although conceptually the study is not particularly novel, because the relationship between myenteric neurons and muscularis macrophages during POI have been studied in the past, including this group, this study provides a comprehensive analysis of gene expression datasets of both neurons and macrophages to get a better mechanistic understanding of the relationship between two cell types. Although the study is viewed as significant, some weaknesses came to the attention of this reviewers.

We thank the reviewer for the comments and hope our response to all questions is satisfying. To start, we would like to point out that our group has never studied the relationship between neurons and macrophages in the past. In the last few years, we have focused mainly on the role of enteric glial cells in POI, and before, the immune cells were of specific interest to us.

Major conceptual weaknesses:

It remains unclear what is the most upstream signal that drives myenteric plexitis after surgical manipulations. The authors generated comprehensive gene expression datasets at early (3 hrs) and late (24 hrs) time points.

We are indeed still unaware of the whole mechanism for POI development. As our research continues, we discover more aspects of the inflammatory cascade during POI and its complexity. However, our current study was not focused on understanding the upstream signal or the driver of the inflammatory reaction; we were interested in the role of enteric neurons in the ongoing processes.

1) For neurons, they show the analysis for both time points, but for macrophage responses, they focus on the late response (24 hrs). Why? Is it because the 3 hr time point is not as impressive or significant? If this is the case, it seems to be a missed opportunity to understand "the chicken and the egg" relationship between two cell types. Muscularis macrophages are present within the myenteric plexus and not exposed to the abdominal environment.

It seems that we need to clarify some data about the macrophages. In Figure 4, we show mainly data about resident macrophages at 2 time points (IM3h and IM24h). We now have also included additional data for infiltrating immune cells (primarily monocyte-derived macrophages) to highlight the resident cells' crucial and neurotoxic role in POI (see Figure EV4). In Figure 5, we decided to focus on the IM24h time point because, for our hypothesis, we wanted to analyze the impact of macrophage depletion on the neurodegenerative phenotype at the peak of the disease, which was previously shown to be at IM24h (PMID 37941007, 33332729, 33977334). At IM24h, we could validate neuronal loss and synaptic damage.

Additionally, we included now data from neuronal *RiboTag* mice (also used in Figure 3) in the depletion model at IM24h to get specific mRNA expression data from enteric neurons in the absence of resident macrophages (see Figure 5g+h). Consequently, we are not planning any analysis from the IM3h time point. We understand the interest in the “chicken and egg” question during POI, but for our study, we focused on the outcome of the macrophage depletion on neuronal function/death and not on the initiation process of POI.

2) Do abdominal manipulations activate peritoneal macrophages or serosal macrophages which then transmit the signal to enteric neurons and then neurons pass it to muscularis macrophages?

Regarding peritoneal macrophages, we talked with various collaboration partners in the POI research field. So far, this line of thought (see question 2) has not been followed up. We agree that the peritoneal macrophages may play a role after surgery, but focusing on the mechanical stimulus from the intestinal manipulation (IM), we are quite confident that the majority of the inflammatory reaction initiates in the muscle layer of the small intestine. Our studies and research of many other groups showed data of several cell types being activated by IM in the *muscularis externa* in the early stages of POI.

Regarding serosal macrophages, we honestly never thought about them. We think they are not directly connected to the ENS, as our confocal images showed only a few neuronal and glial cells when focused on their cell layer (See confocal image below). However, they could be activated by IM, and they could transmit this signal to other macrophages closer to the ENS. At the moment, we cannot prove or disprove anything in this regard, but maybe future work will shed light on this. In agreement with the editor, we would like to focus our study on the role of enteric neurons and the occurring enteric neurodegeneration.

3) Peritoneal macrophages are likely to be affected by CD115-antibody treatment. Can it be explored, e.g., **by administering the antibody i.v. vs i.p. and by looking at early time points more closely?** Either outcome of this analysis will be informative for our understanding of the POI pathogenesis.

See also the above comment to Question 1.

We think that the CD115-antibody treatment will affect peritoneal macrophages as well, but their role is not discussed in our study, and one additional *in vivo* experiment will not clarify their role in POI. It rather will open up more questions for a topic we are not addressing in our manuscript at the moment.

In general, the topic of peritoneal macrophages is interesting, but in agreement with the editor, we would like to keep it separate from our current study.

However, we share an interest in deciphering the role of different macrophage populations, which was why, for our study, comparing resident and infiltrating cells seemed crucial. Therefore, to better define the impact of resident cells, we now analyzed neuronal death in CCR2-KO mice that underwent intestinal manipulation (IM). The CCR2-KO mice were already used in POI studies, and their phenotype is published (PMIDs: 21113155+28615302+30581430), showing that after IM, these mice have no visible leukocyte infiltration into the *muscularis externa*, an attenuated inflammatory reaction, and a prolonged impairment of the GI transit. In this experimental setup, we compared the impact of resident macrophages on enteric neurons in POI without infiltrating monocytes. We included the new data in **Figures EV4**, confirming the crucial role of resident macrophages in enteric neurodegeneration.

Other points:

4) "A prominent example of an acute neuroinflammatory disease in the gut is postoperative ileus (POI)". - It seems unappropriated to call POI a disease. It is not an independent clinical entity, but a pathological condition that develops as a complication of surgery.

We agreed with the reviewer and rephrased this sentence accordingly.

"A prominent example of an acute neuroinflammatory condition in the gut is postoperative ileus (POI)."

5) Fig. 1c and all other images showing myenteric ganglia: show lower magnification images to demonstrate whether there is any heterogeneity in neuronal activation among different ganglia and fields. This will be very informative for our understanding of POI pathogenesis.

We tried to show a clear co-localization or possible interaction by the cell localization with our images. However, we agree that overview image is of interest to the readers, so we added a stitched image (3x3 with 200x magnification) of the FOS staining and included it in the supplementary figures (see supplementary figure S1d).

6) All figures: no quantification of neuronal loss or cFos+ neurons, apoptotic neurons etc. This would be informative to understand the overall impact of intestinal manipulation on neuronal activation, death etc, and also on heterogeneity of the response across ganglia as mentioned above.

We agree with the reviewer, so we included quantifications of neuronal activation (cFOS⁺) in Figure 1c, neuronal loss (Figure 2c), and neuronal apoptosis (Figure 2d). Additionally, we performed new analyses to quantify the synaptic damage during POI in Figure 2a and EV4l. In Figures 4 and 5, we also quantified the macrophage numbers to show increased macrophage numbers around the ganglia and the success of the depletion. To better define macrophages around ganglia, we included stainings and quantifications of F11R-positive macrophages during POI in Figures 4c. In a recent publication by the Boeckxstaens group, these cells were verified to be associated with neurons in the gut (PMID: 37316669) so we could compare the numbers of "close" and "associated" macrophages during the disease.

7) Fig. 2d: CC3 staining is not convincing based on the images provided.

Sadly, we agree with the reviewer, as the staining for cleaved-caspase 3 in the whole mount specimens is not ideal. For clarification, we performed caspase 3 stainings in paraffin and cryosections to show co-localization of cleaved-caspase3 and neurons. The new images of stained ganglia were included in **Figure S2b**.

8) Page 11, final line: "suggested a direct interaction between enteric neurons and their inflammatory environment". - it is not clear how the authors arrived to this conclusion based on the data that they provide.

Our data shows that enteric neurons are activated (cFOS⁺), dying (cleaved Caspase3⁺), and strongly change their transcriptional profile (*RiboTag* data from Figure 3). In POI, a strong inflammatory cascade is induced, and we hypothesize that enteric neurons play a part in this reaction and interact with their environment. But to prevent any confusion, we changed the wording in the sentence:

"... and **suggested an impact of the inflammatory environment on enteric neurons.**"

9) Page 13, middle: "suggesting a direct exchange between both cell types". - the conclusion is unclear. Exchange of what? Based on what was it concluded?

We agree with the reviewer and rephrased this sentence to describe our conclusion in more detail. In our opinion, macrophages release several factors, such as cytokines and growth factors, that can activate/harm surrounding cells, in our case, enteric neurons.

"At IM24h, the ganglia-associated macrophages assembling around ANNA1⁺ enteric neurons strongly increased (Figure 4a+b and EV4a), suggesting a possible interaction between both cell types."

10) Page 14, last sentence: "Our results point out that activated resident macrophages create and maintain a neurodegenerative environment by causing inflammatory and metabolic stress for enteric neurons after POI induction, altering their gene expression profiles, and inducing neuronal death programs". - it should be clarified that the data are suggestive or predictive rather than confirmatory. The wording makes it sound like it is a statement.

We agree with the reviewer and rephrased this sentence. However, our new data support our initial hypothesis. Especially our newly included comparison between infiltrating and resident macrophages highlights the neurodegenerative role of the resident cells quite clearly.

"Our results suggest that activated resident macrophages are able to create and maintain a neurodegenerative environment for enteric neurons after POI induction, altering their gene expression profiles and inducing neuronal death programs."

11) Fig. 4d: is quantification of macrophages automated or manual?

For Figure 4b, the macrophage number was quantified from whole-mount specimens stained for macrophage markers (MHCII) or GFP (CX3CR1-GFP mice) in a blinded manner by two group members independently (authors Breßer and Siemens).

12) Page 16: "Accordingly, expression levels of prominent POI disease genes (Arg1, Egr1, Ccl2, Il6, (17, 19)) showed less upregulation at IM24h in depleted animals and no changes in controls (Figure S5f)". - What about n3 hrs POI? Th peak of neuronal loss is at 3 hrs but the data for macrophages are shown at 24 hrs. Is there a reason for that?

See also the above comment to question 1.

The peak of cleaved-caspase3⁺ and cFOS⁺ neurons is at IM3h, but the phenotype of enteric neurodegeneration is at IM24h. There, we see the neuronal loss and the strong synaptic damage. Additionally, the hallmarks of POI (impaired GI transit, high immune cell infiltration, and strong neuroinflammation) are strongest at IM24h. As macrophage depletion should have an impact on the complete progression of POI, we hypothesized that we would detect bigger differences at the later stages. Our data from Figure 5 confirmed this hypothesis, showing significant changes in all POI hallmarks.

We agree that the IM3h time point is interesting, but to validate possible treatment options, we needed data on the outcome of POI with a focus on motility impairment. All effects of neurodegeneration are prolonged during the disease, so the later stages were, for our study, the reasonable choice.

13) Fig. 6: what types of jejunal biopsies are shown - transmural or mucosal + submucosal? When showing neurons (Fig. 6b) what are we looking at - myenteric neurons or submucosal neurons?

In Figure 6, we work with *muscularis externa* samples of jejunal gut samples. We took the muscle layer because it is more relevant for POI studies. Figures 6b and S6a show histological images of myenteric ganglia in the *muscularis externa*. We included more information in the figure legends to clarify the region and better describe the human samples.

14) In all figure legends, it should be clearly stated what region of the gut is being analyzed. Do the authors always look at the same region?

We agreed with the reviewer and included information about the gut region in all figure legends. For all histological analyses in mice, we always focused on areas at the end of the jejunum and the beginning of the Ileum. For the workup of murine samples, we always divide the whole gastrointestinal tract (from stomach to colon) into 15 sections to choose the same sections for analysis. A Recent publication shows our procedure very well (PMID: 39072642).

27th Nov 2024

Dear Reiner,

Thank you for submitting your revised manuscript to EMBO Molecular Medicine. We have now received the enclosed report from the three referees who re-assessed your work. As you will see, the referees think the manuscript has significantly improved after revision. However, several relatively minor concerns still remain, and we would ask you to address them in a revision.

On a more editorial level:

1. Please remove figures from the manuscript file. The legends for main and EV figures should stay in the manuscript file.
 2. In the manuscript file, please provide author names as first name, last name, instead of last name, first name.
 3. Please remove the authors' contribution section from the manuscript file.
 4. Funding information needs to be part of "Acknowledgements".
 5. Please rename "Materials and Methods" to "Methods".
 6. In Methods, include a statement that informed consent was obtained from all subjects and that the experiments conformed to the principles set out in the WMA Declaration of Helsinki and the Department of Health and Human Services Belmont Report.
 7. The references need to be formatted according to the EMBO Molecular Medicine reference style.
 - Please list up to 10 co-authors of a paper before adding et al. to the reference list.
 - Citations should be listed in alphabetical order.
 - Year should be in brackets.
 - 'Literature Cited' should be renamed to 'References'.
 8. Appendix
 - The supplementary figures and tables need to be bundled together with their legends in a single PDF file called *Appendix*.
 - The title page should include a Table of Content with page numbers of all the items.
 - Each legend should be below the corresponding Figure/Table in the Appendix.
 - The nomenclature should be "Appendix Figure S1, Appendix Figure S2, Appendix Table S1" etc. The callouts in the manuscript should be updated accordingly.
 - EV figure legends should be removed from the Appendix and placed in the main manuscript file.
 9. Data Availability
 - please include the specific URLs for (GSE198889, GSE205610, GSE134943, GSE149181) datasets in the data availability statement.
 - remove "will be" from the statement and make sure these datasets are publicly accessible upon the acceptance of the manuscript.
 10. Every published paper now includes a 'Synopsis' to further enhance discoverability. Synopses are displayed on the journal webpage and are freely accessible to all readers. They include a short stand first (maximum of 300 characters, including space) as well as 2-5 one-sentences bullet points that summarizes the paper. Please write the bullet points to summarize the key NEW findings. They should be designed to be complementary to the abstract - i.e. not repeat the same text. We encourage inclusion of key acronyms and quantitative information (maximum of 30 words / bullet point). Please use the passive voice.
- Please attach these in a separate file or send them by email, we will incorporate them accordingly.
11. Source data need to be reorganized - SD for the main figures should be uploaded as one folder per figure. The source data for EV and Appendix figures should be grouped together and uploaded as one folder.
 12. When submitting your revised manuscript, please do not include the Reagents and Tools Table in the Methods section of the manuscript but upload it as a separate file choosing the file type "Reagent Table". The Reagents and Tools Table template (.docx) can be found in our author guidelines:
<https://www.embopress.org/page/journal/17574684/authorguide#structuredmethods>
 13. The manuscript sections should be in the following order: Title page - Abstract & Keywords - Introduction - Results -

Discussion - Methods - Data Availability - Acknowledgments - Disclosure Statement & Competing Interests - References - Figure Legends -(Main Tables with legends if applicable) - Expanded View Figure Legends.

14. Please address the following issues in the Figure Legends :

- Please note that the black arrowheads are mislabeled as white arrowhead in the legend of figure 5d'. This needs to be rectified.
- Please note that the exact p values are not provided in the legends of figures 1c"; 2a"-c, d", e", f-i; 3f-g; 4b, c", e, h; 5c, d"-e, f"-h; 6c-f, j-k; EV 1d-f; EV 2c, e"-h; EV 3b", f; EV 4i-j; EV 5b-d, e"-h; EV 6f.
- Please indicate the statistical test used for data analysis in the legends of figures 1d-f, j-k; 3c-d; 4g; 6g; EV 1h; EV 2d; EV 3c-e; EV 4e; EV 5d; EV 6c-d, g.
- Please note that in figures 1c"; 5c, d"-e, f"-h; 3b", f; EV 4i-l; there is a mismatch between the annotated p values in the figure legend and the annotated p values in the figure file that should be corrected.
- Please note that information related to n is missing in the legends of figures 3c; EV 3c; EV 4e; EV 6c.
- Although 'n' is provided, please describe the nature of entity for 'n' in the legends of figures 2d", h; 4c".
- Please note that for heatmap present in figures 1g-i; 3e; 6h-i; EV1c; EV 4f-h; EV 6h; a numbered scale bar is not provided. This needs to be rectified.
- Please note that the white arrowheads are not defined in the legends of figures 5b. This needs to be rectified.

Please attach a .docx formatted letter INCLUDING the reviewers' reports and your detailed point-by-point responses to their comments.

I look forward to seeing a revised form of your manuscript as soon as possible.

Use this link to login to the manuscript system and submit your revision: <https://embomolmed.msubmit.net/cgi-bin/main.plex>

Kind regards,
Jingyi

Jingyi Hou
Editor
EMBO Molecular Medicine

***** Reviewer's comments *****

Referee #1 (Comments on Novelty/Model System for Author):

The authors have responded appropriately to my comments and have revised the manuscript accordingly. I have no further comments.

Referee #1 (Remarks for Author):

The authors have responded appropriately to my comments and have revised the manuscript accordingly. I have no further comments.

Referee #2 (Comments on Novelty/Model System for Author):

The current manuscript has a medium medical impact but novelty is low. The role of muscularis macrophages in POI has been described previously. I acknowledge that some aspects were previously unexplored (e.g. synaptic loss, macrophage in-depth profiling) but I feel there is a missed opportunity here to really advance the field, for example by identifying and validating pathways or mechanisms by which muscularis MF promote neurodegeneration.

Referee #2 (Remarks for Author):

I appreciate the additional experiments conducted to strengthen the manuscript. However, I strongly encourage the authors to address a few remaining concerns before final publication. I have raised these previously but feel they have been misunderstood.

First, I recommended including validation experiments following transcriptomic or proteomic analyses. The authors responded

by performing another profiling experiment (mass spectrometry), but this misses the point. Top hits from unbiased sequencing experiments should always be validated at the mRNA and/or protein level using targeted approaches, and not by conducting yet another profiling experiment. RNA abundance often poorly correlates with functional protein levels due to post-transcriptional regulation and technical artifacts inherent to high-throughput techniques. This is why only a small subset of transcriptional changes is biologically relevant. Moreover, the mechanical processing and enzymatic digestion steps, prolonged incubations and complex procedures (as described by the authors) introduce risks of non-specific or stress-induced transcriptional artifacts and experimental variability, as evidenced by the RiboTag data (Figure 3b). Validation methods such as immunohistochemistry, FACS, or RNAscope could confirm the quantitative presence and spatial localization of the identified candidates within neurons, which is particularly important given the focus on neurodegeneration.

Minor additional points, please include a higher resolution image of the synaptic staining experiments (Figure 2a and others)? They really seem of poor quality. Second, apologies if I missed this, but I cannot locate representative flow cytometry plots for figure 4e. Third, I would recommend excluding the blood plasma assay from the manuscript. There is absolutely no rationale in measuring Ab40/42 levels in a model of POI.

Referee #3 (Comments on Novelty/Model System for Author):

The models are appropriate.

Referee #3 (Remarks for Author):

The manuscript by Breber et al. entitled "Macrophage-induced enteric neurodegeneration leads to motility impairment during gut inflammation" improved significantly after revision, however, some important weaknesses remain, mostly of editorial nature.

Although the study design is straightforward, it is quite difficult to read the manuscript due to the reasons described below.

1) The clinical relevance and significance of the study are not clearly presented.

For example, the title "Macrophage-induced enteric neurodegeneration leads to motility impairment during gut inflammation" is misleading. In scientific literature, "gut inflammation" is applied to mucosal or transmural inflammation caused by inflammatory bowel disease (IBD) or enteric infection. In contrast, postoperative ileus, modeled in this study, is a transient disturbance of GI function in the originally non-inflamed intestine due to a transient inflammatory response limited to the myenteric plexus. Similarly, the Introduction section is very broad, with some generalizations that do not make sense in the context of POI pathogenesis and are confusing.

With this in mind, the abstract and Introduction should be focused on summarizing the published literature relevant to POI pathogenesis while highlighting the unknowns, hypothesizing on POI pathogenesis and providing a concise summary of the experimental design and findings of the study.

2) The language of the manuscript is not precise, with many generalisations, over-interpretations and strong statements based on suggestive or predictive transcriptomics data. Some sentences are hard to make sense of. Below are some examples.

3) All figure legends should exclude data interpretations. Instead, more information should be provided about experimental design and readouts. E.g., "at IM3h, a clear population of double positive cells (white arrowheads) was detected". Instead, "white arrowhead points to cFos-positive cells among the Anna1+ myenteric neurons". Arrows could be made smaller not to interfere with the images. Also suggested: "C' is a representative image for c".

Suggested corrections:

Abstract: "followed by transcriptional and translational signatures of neuronal proliferation and death coinciding with synaptic degradation." - This sentence needs to be revisited.

Abstract: "In human gut samples taken early and late during abdominal surgery, we substantiated the murine data and found reactive and apoptotic neurons with dysregulated synaptic signaling, discovering a species' overarching mechanism." - This sentence needs to be revisited.

Introduction: "Although multiple forms of intestinal pathologies are known to impair ENS functions (6), detailed investigations focusing particularly on the role or response of enteric neurons in inflammatory mechanisms are still lacking (7)." - This sentence needs to be revisited. POI is one of the few intestinal pathologies known to impair ENS function, but it is not an inflammatory disease on its own.

Results: "POI is a frequent transient GI-motility disorder and complication of abdominal surgery." - suggested "POI is a frequent complication of abdominal surgery". POI is not an independent disease or disorder, but a transient condition complicating surgery.

It is suggested that the authors avoid overstated phrases like "found striking evidence of strong enteric neuronal activity".

"To create this unique intestinal environment and focus on enteric neurons therein, we applied the mouse model of postoperative ileus (POI)." - Suggested: "To understand the pathogenesis of surgery-associated GI dysfunction, we applied a mouse model of postoperative ileus to study the impact of intestinal manipulation during surgery on myenteric neurons".

"Induction of POI is a standardized model using surgical trauma by intestinal manipulation (IM)". - This sentence needs to be revisited.

Legend Fig. 1a: "At IM24h, the disease peak, an immense immune cell infiltrate is present in the ME and, in concert with the ENS, evokes typical symptoms of POI." - This sentence needs to be revisited. "Immense" should be removed. A reference to a supplemental figure where this data is shown should be made.

Fig. 2c: "Control and IM24h mice showed almost no double-positive cells, whereas, at IM3h, many activated neurons (myenteric plexus) were detected." - let the reader decide how to interpret the data. Instead, this information could be provided in the text. Is quantification in c done per field, square mm or something else? There is no legend. C" - it is not indicated how groups are compared to each other, especially for group 3 on the right (24hrs post-IM)? It looks similar to controls, but p value shows high significance.

Fig. 1f: There is no legend.

Fig. 1k: remove "shows strong induction of pathways connected to neuronal functions."

Fig. 1j: Is it possible to indicate specific proteins to demonstrate a correlation between gene and protein expression?

Fig. S1a: "Bar graphs show the mean CD45+ cell number normalized to the ME tissue weight." - This is not a common way of normalizing flow cytometry data. Usually, normalization is done to total viable cells or total viable singlets for relative numbers. Absolute counts are determined by multiplying relative numbers to total cell numbers in each sample.

Fig. S1b and other figures below: How long does it take to resolve myenteric plexitis following IM? The data below show post-IM 24 hrs and day 21, but no time points in between.

Fig. S1d: "immunohistochemistry" should be replaced with "immunofluorescence" (and in all other figures). In figure S1d, there is no indication of Anna1 staining in grey.

Fig. 2: how homogeneous is the effect in Fig. 2 among the ganglia? Kulkarni et al initially showed a high rate of neurogenesis in normal small bowel, however, later with more careful analysis they found a lot of heterogeneity in the rate of neurogenesis among different ganglia? Are the ganglia shown in Fig. 2 represent the ganglia most dramatically impacted by IM?

Fig. 2e: CC3 staining. Grey color for Anna1 is not the best option as it looks like every single neuron is pink, i.e., CC3+. Fig. S2b is more convincing. Also, statistically, it looks like in some mice up to 50% of neurons are CC3+ which is hard to believe, while the image on the left points to only 3 neurons out of 12 being CC3+. This will be 25%. CC3 antibody staining of gut tissue gives a lot of background and should be interpreted with caution.

Similarly, Figure EV2e: It looks like all Anna1 positive cells are Ki67+ (up to 90% when quantified). This is surprising even for inflamed myenteric plexus. According to Kulkarni et al (PMID: 28420791), mature enteric neurons do not proliferate. Is there a variability in Ki67 labeling of myenteric neurons among the ganglia. Low-magnification high-resolution images would be helpful in addition to high-resolution images of selective ganglia.

Fig. 2b: gene abbreviation for mouse gene would be *Bax* (italic), for human - *BAX* (italic).

Fig. 2: "Following up on the detrimental enteric neuronal phenotype in POI, we explored synaptic deterioration by immunohistochemistry and gene expression analyses." - Images provided do not show synaptic deterioration. Instead, they show expression patterns of proteins associated with synaptic function.

"but a substantial decay at IM24h" - replace with "reduced expression".

"Neurodegeneration" may not be the best term describing the changes in myenteric neurons after IM. Neurodegeneration implies a progressive nature, while it looks like changes in enteric neurons in POI are transient and with no long-term functional outcomes.

Figure 3:

"CX3CR1+/F11R+ macrophage numbers increased during POI, indicating possible inflammation-induced communication between enteric neurons and macrophages." - Based on the images provided there is no evidence that macrophages and neurons communicate. Did the authors mean "cell-cell interaction"? There is also a possibility that macrophages upregulate F11R expression upon IM.

"while genes associated with synaptic transmission (e.g., Arc, Shank1, Nrnx1) were differentially regulated, hinting at a more engaged neuronal phase in the early disease state". - It is unclear what the authors are trying to say. This sentence needs to be revisited.

"demonstrating functional changes in neuronal transmission in the inflamed ME." - The authors probably meant "predicting", not demonstrating. This sentence needs to be revisited.

"our transcriptional data hint at a severe inflammatory and neuro-deteriorating phenotype". - "severe" and "neuro-deteriorating" should be removed.

"neurotoxic actions in POI". - "neurotoxic" is a very specific term not relevant in this context.

Genes highlighted under the category "neuroinflammation" are non-specific inflammatory genes. They should be listed as "inflammation".

Discussion:

It looks like neuronal damage by IM manipulation is transient and does not have any significant functional consequences in mice. Is it the case in patients who underwent abdominal surgery complicated by POI?

The discussion should highlight the limitations of the study.

Other comments: in figure legends "enteric neurons (myenteric plexus)" could be replaced with "myenteric neurons".

Point by Point

Referee #1 (Comments on Novelty/Model System for Author):

The authors have responded appropriately to my comments and have revised the manuscript accordingly. I have no further comments.

Referee #1 (Remarks for Author):

The authors have responded appropriately to my comments and have revised the manuscript accordingly. I have no further comments.

We thank reviewer1 for his work.

Referee #2 (Comments on Novelty/Model System for Author):

The current manuscript has a medium medical impact but novelty is low. The role of muscularis macrophages in POI has been described previously. I acknowledge that some aspects were previously unexplored (e.g. synaptic loss, macrophage in-depth profiling) but I feel there is a missed opportunity here to really advance the field, for example by identifying and validating pathways or mechanisms by which muscularis MF promote neurodegeneration.

Reviewer2 is correct. The pathways or mechanisms by which muscularis macrophages promote enteric neurodegeneration are very interesting. Therefore, we are already focusing on this aspect in a followup study.

Referee #2 (Remarks for Author):

I appreciate the additional experiments conducted to strengthen the manuscript. However, I strongly encourage the authors to address a few remaining concerns before final publication. I have raised these previously but feel they have been misunderstood.

First, I recommended including validation experiments following transcriptomic or proteomic analyses. The authors responded by performing another profiling experiment (mass spectrometry), but this misses the point. Top hits from unbiased sequencing experiments should always be validated at the mRNA and/or protein level using targeted approaches, and not by conducting yet another profiling experiment. RNA abundance often poorly correlates with functional protein levels due to post-transcriptional regulation and technical artifacts inherent to high-throughput techniques. This is why only a small subset of transcriptional changes is biologically relevant. Moreover, the mechanical processing and enzymatic digestion steps, prolonged incubations and complex procedures (as described by the authors) introduce risks of non-specific or stress-induced transcriptional artifacts and experimental variability, as evidenced by the RiboTag data (Figure 3b). Validation methods such as immunohistochemistry, FACS, or RNAscope could confirm the quantitative presence and spatial localization of the identified candidates within neurons, which is particularly important given the focus on neurodegeneration.

As mentioned before, we are currently interested in characterizing enteric neurodegeneration as a general phenotype after surgery, and no specific targets have been chosen for in-depth analysis. For all our conclusions, we tried to gather different types of data sets for validation.

We showed neuronal activation and death by histology and transcriptomics, including cell-specific *RiboTag* data. We described synaptic damage in POI using various histology approaches and confirmed it through gene and protein expression analyses. Additionally, we looked at resident macrophages by histology, defined their gene profile by SMART-Seq2, and compared their impact on POI to the infiltrating cells by using antibody-depletion and knockout mouse approaches.

At this stage, we are not sure which specific targets we should validate by other techniques to strengthen our hypothesis. We scoured our manuscript again and tried to obtain certain genes or proteins of interest that were highlighted in the study but could ultimately not decide on a specific candidate. We hope that followup work by us or others will present specific targets in neurons or macrophages and validate them in new mouse studies to build upon the groundwork we have supplied here.

Minor additional points, please include a higher resolution image of the synaptic staining experiments (Figure 2a and others)? They really seem of poor quality.

As we aimed to show larger areas of synaptic damage, the resolution was initially low. While the images are still not perfect, we increased the resolution of the confocal images in Figure 2a to the maximum of the microscope (8192x8192) and hope it sufficiently displays synaptic damage. Additionally, we conferred with our collaborators from the group of Anja Schneider to get their opinions on our synaptic staining in comparison to similar stainings in the brain. We agreed that brain tissue offers higher quality and clearer images than gut tissue, but the signal is comparable overall.

Second, apologies if I missed this, but I cannot locate representative flow cytometry plots for figure 4e.

The reviewer2 is correct; we forgot to include these plots. In the new version, you can find them in Appendix Figure S3a.

Third, I would recommend excluding the blood plasma assay from the manuscript. There is absolutely no rationale in measuring Ab40/42 levels in a model of POI.

Reviewer2 is correct, Ab40/42 is not connected to POI; we only included these data as a positive control for the Alzheimer mouse model. We excluded these plots from the Appendix Figures.

Referee #3 (Comments on Novelty/Model System for Author):

The models are appropriate.

Referee #3 (Remarks for Author):

The manuscript by Breber et al. entitled "Macrophage-induced enteric neurodegeneration leads to motility impairment during gut inflammation" improved significantly after revision, however, some important weaknesses remain, mostly of editorial nature.

We thank reviewer3 for the comments and hope our response to all questions is satisfying.

Although the study design is straightforward, it is quite difficult to read the manuscript due to the reasons described below.

1) The clinical relevance and significance of the study are not clearly presented.

For example, the title "Macrophage-induced enteric neurodegeneration leads to motility impairment during gut inflammation" is misleading. In scientific literature, "gut inflammation" is applied to mucosal or transmural inflammation caused by inflammatory bowel disease (IBD) or enteric infection. In contrast, postoperative ileus, modeled in this study, is a transient disturbance of GI function in the originally non-inflamed intestine due to a transient inflammatory response limited to the myenteric plexus. Similarly, the Introduction section is very broad, with some generalizations that do not make sense in the context of POI pathogenesis and are confusing.

Over the past two decades, POI has been a reliable and well-studied model, which has enabled us to start our investigation of the neuronal role in gut inflammation. We agree that other models induce stronger gut inflammation and can give more insights into the connection between neuroinflammation and ENS function, but due to the acute and transient nature of the POI model, it is especially suited to provide information about early neuronal reactivity after a singular inflammatory stimulus. As our data suggests, enteric neuronal reactivity needs even further investigation and tailored models to thoroughly describe this exciting cell type. Accordingly, POI is not the central focus of this study but rather a previously well-established and -defined instrument to study neuronal function in acute neuroinflammation. As we predicted these cells to be involved in the inflammatory cascade and thus POI pathogenesis, we believe this model to be a good basis for the initial description of acute neuronal reactivity in the ENS.

With this in mind, the abstract and Introduction should be focused on summarizing the published literature relevant to POI pathogenesis while highlighting the unknowns, hypothesizing on POI pathogenesis and providing a concise summary of the experimental design and findings of the study.

We made all the suggested changes below and revised the Introduction and the abstract accordingly.

2) The language of the manuscript is not precise, with many generalisations, over-interpretations and strong statements based on suggestive or predictive transcriptomics data. Some sentences are hard to make sense of. Below are some examples.

In the revision, we added many experiments to validate our transcriptomic data, e.g., many histological stainings, mass-spec analysis, and *in vivo* data of CCR2-KO mice. All new experiments underlined our transcriptomic data and supported the interpretations and statements in the manuscript. While not all questions were answered fully, we have used a variety of different methods at our disposal to provide sufficient data to convince the reader and the reviewers and still offer avenues to be explored in the future.

3) All figure legends should exclude data interpretations. Instead, more information should be provided about experimental design and readouts. E.g., "at IM3h, a clear population of double positive cells (white arrowheads) was detected". Instead, "white arrowhead points to cFos-positive cells among the Anna1+ myenteric neurons". Arrows could be made smaller not to interfere with the images. Also suggested: "C' is a representative image for c".

We agreed with reviewer3 and modified the figure legends and the arrowheads in the images.

Suggested corrections:

Abstract: "followed by transcriptional and translational signatures of neuronal proliferation and death coinciding with synaptic degradation." - This sentence needs to be revisited.

We revised the sentence.

Abstract: "In human gut samples taken early and late during abdominal surgery, we substantiated the murine data and found reactive and apoptotic neurons with dysregulated synaptic signaling, discovering a species' overarching mechanism." - This sentence needs to be revisited.

We revised the sentence.

Introduction: "Although multiple forms of intestinal pathologies are known to impair ENS functions (6), detailed investigations focusing particularly on the role or response of enteric neurons in inflammatory mechanisms are still lacking (7)." - This sentence needs to be revisited. POI is one of the few intestinal pathologies known to impair ENS function, but it is not an inflammatory disease on its own.

We are unsure how to revise the sentence properly. We do not mention POI as an inflammatory disease; we only point out that enteric neurons should be investigated in context to inflammatory mechanisms. One aspect of POI is defined by an inflammation-based mechanism that leads to motility impairment, and thus, it is an appropriate model to study the role of enteric neurons in gut inflammation.

Results: "POI is a frequent transient GI-motility disorder and complication of abdominal surgery." - suggested "POI is a frequent complication of abdominal surgery". POI is not an independent disease or disorder, but a transient condition complicating surgery.

We revised the sentence.

It is suggested that the authors avoid overstated phrases like "found striking evidence of strong enteric neuronal activity".

We revised the sentence accordingly.

"To create this unique intestinal environment and focus on enteric neurons therein, we applied the mouse model of postoperative ileus (POI)." - Suggested: "To understand the pathogenesis of surgery-associated GI dysfunction, we applied a mouse model of postoperative ileus to study the impact of intestinal manipulation during surgery on myenteric neurons".

We revised the sentence accordingly.

"Induction of POI is a standardized model using surgical trauma by intestinal manipulation (IM)". - This sentence needs to be revisited.

We revised the sentence:

"Induction of POI can be accomplished in a standardized manner using surgical trauma evoked by intestinal manipulation (IM) to trigger acute intestinal inflammation, ..."

Legend Fig. 1a: "At IM24h, the disease peak, an immense immune cell infiltrate is present in the ME and, in concert with the ENS, evokes typical symptoms of POI." - This sentence

needs to be revisited. "Immense" should be removed. A reference to a supplemental figure where this data is shown should be made.

We agreed with reviewer3 and modified the text.

Fig. 2c: "Control and IM24h mice showed almost no double-positive cells, whereas, at IM3h, many activated neurons (myenteric plexus) were detected." - let the reader decide how to interpret the data. Instead, this information could be provided in the text. Is quantification in c done per field, square mm or something else? There is no legend. C" - it is not indicated how groups are compared to each other, especially for group 3 on the right (24hrs post-IM)? It looks similar to controls, but p value shows high significance.

We think reviewer3 is talking about Figure 1c, as 2c is the new mass spec data for synaptic proteins and there are no other images in this figure.

We modified the figure legends and included more data about the quantification. IM24h was compared with IM3h and showed significance.

Fig. 1f: There is no legend.

We included an additional legend for Figure 1f.

Fig. 1k: remove "shows strong induction of pathways connected to neuronal functions."

We agreed with reviewer3 and modified the figure legend.

Fig. 1j: Is it possible to indicate specific proteins to demonstrate a correlation between gene and protein expression?

We included some proteins of interest in the volcano plots to show correlations between gene and protein expression in POI and modified the results section accordingly.

Fig. S1a: "Bar graphs show the mean CD45+ cell number normalized to the ME tissue weight." - This is not a common way of normalizing flow cytometry data. Usually, normalization is done to total viable cells or total viable singlets for relative numbers. Absolute counts are determined by multiplying relative numbers to total cell numbers in each sample.

In our studies, we normalized the cell number by adding counting beads to the samples. These beads serve as a stopping gate for the immediate flow cytometry measurement. As soon as 4000 beads have been measured, the analysis is stopped. Of course, the weight of the inserted muscularis externa (*ME*) has to be taken into account in the following analysis, as we know that the *ME* has a different weight depending on the age of the animal and the time of manipulation. The mentioned counting beads can be used to normalize/calculate the number of cells depending on the initial weight of the *ME*. Based on the number of cells detected by the analyzer after 4000 beads, the weight of the *ME* can be calculated using a simple rule of three. Of course, "Normalization to total viable cells or total viable singlets for relative numbers or determining absolute counts by multiplying relative numbers to total cell numbers" also represents a possible approach. However, we prefer to use the method mentioned above in order to be able to normalize the weight of the material used and keep the data comparable to our former FACS studies, as we used this normalization in the last 15 years for all experiments.

Fig. S1b and other figures below: How long does it take to resolve myenteric plexitis following IM? The data below show post-IM 24 hrs and day 21, but no time points in between.

For motility disturbance and immune cell infiltration, 72h post-surgery already shows almost naïve levels, while seven days after IM, we experience a complete recovery. We showed these data in older studies (PMID: 30581430) but never focused on the neuronal phenotype. We have some additional data on other time points, but we decided that the late time point (21 days) was sufficient to prove the recovery of the mice. For followup work, we will investigate more time points to better understand the recovery phase.

Fig. S1d: "immunohistochemistry" should be replaced with "immunofluorescence" (and in all other figures). In figure S1d, there is no indication of Anna1 staining in grey.

We changed the wording in the whole text and we modified the figure legend in Appendix Figure S1d.

Fig. 2: how homogeneous is the effect in Fig. 2 among the ganglia? Kulkarni et al initially showed a high rate of neurogenesis in normal small bowel, however, later with more careful analysis they found a lot of heterogeneity in the rate of neurogenesis among different ganglia? Are the ganglia shown in Fig. 2 represent the ganglia most dramatically impacted by IM?

We did not see a big variability of the ganglia with these effects. There are sometimes technical issues with the preparation, as the inflamed tissue can be tricky to prepare, but for our counting, we took images of five random areas per small bowel muscle layer whole mount and counted all visible ganglia for double-positive cells.

However, we always analyzed the same area of the gut and never checked in other parts, so regional differences might occur and have not been analyzed in this study. Importantly, during POI, the entire small intestine is subjected to trauma through IM.

Fig. 2e: CC3 staining. Grey color for Anna1 is not the best option as it looks like every single neuron is pink, ie., CC3+. Fig. S2b is more convincing. Also, statistically, it looks like in some mice up to 50% of neurons are CC3+ which is hard to believe, while the image on the left points to only 3 neurons out of 12 being CC3+. This will be 25%. CC3 antibody staining of gut tissue gives a lot of background and should be interpreted with caution.

We agree with reviewer3 that the CC3+ staining in the whole mount specimens is not perfect. However, we repeated the stainings several times, and the signal quality and intensity were always similar. To validate double positive cells, we performed the same staining in paraffin and cryosections of the small bowel and detected CC3+ neurons throughout the ENS.

The counting was done by two people in a blinded manner, and it resulted in similar numbers for the double-positive cells.

We realize that the counting of CC3+ cells alone is not enough to substantiate our hypothesis of neuronal death in POI, so we also quantified ANNA1+ cells throughout POI progression and looked for the regulation of neuronal death signatures in bulk and *RiboTag* RNA-Seq data sets.

Similarly, Figure EV2e: It looks like all Anna1 positive cells are Ki67+ (up to 90% when quantified). This is surprising even for inflamed myenteric plexus. According to Kulkarni et al (PMID: 28420791), mature enteric neurons do not proliferate. Is there a variability in Ki67

labeling of myenteric neurons among the ganglia. Low-magnification high-resolution images would be helpful in addition to high-resolution images of selective ganglia.

The high counts of Ki67-positive neurons are very difficult to interpret. We discussed this exact point with Prof. Kulkarni at the DDW this year. We saw the induction of neuronal proliferation on the transcript level (*RiboTag* data) and in the histology analysis, but these high numbers of double-positive cells would indicate a far too strong cell division of neurons in the inflamed gut. In theory, we would end up with more neurons after the insult than in a healthy mouse. Therefore, we discussed the option of upregulating Ki67 in neurons independently of proliferation. We included more literature and some additional sentences in our discussion.

“Compensation of the enteric neuron loss during POI by generation of new neurons is very likely, as we found induction of neuronal proliferation by upregulation of Ki67-positive myenteric neurons and neurogenesis gene patterns in our sequencing data of the diseased mice. Interestingly, recent publications also discuss additional functions of Ki67, such as its role in cell cycle arrest and/or cell synchronization (Miller et al, 2018; Sun & Kaufman, 2018). These functions may play an alternative role in the recovery of neurons during intestinal inflammation in POI.”

We saw no big variability of the ganglia in the Ki67 stainings, but the intensity of the Ki67 signal varied between the neurons and other cell types. This could be a technical or biological reason. Therefore, the low-magnification images are difficult to generate, as the signal is sometimes very weak and will not result in a high-quality image for publication.

Fig. 2b: gene abbreviation for mouse gene would be *Bax* (italic), for human - *BAX* (italic).

We changed the abbreviation in the text and figure.

Fig. 2: "Following up on the detrimental enteric neuronal phenotype in POI, we explored synaptic deterioration by immunohistochemistry and gene expression analyses." - Images provided do not show synaptic deterioration. Instead, they show expression patterns of proteins associated with synaptic function.

We agreed with reviewer3 and changed the wording from “deterioration” to “function”.

"but a substantial decay at IM24h" - replace with "reduced expression".

We agreed with reviewer3 and modified the wording.

"Neurodegeneration" may not be the best term describing the changes in myenteric neurons after IM. Neurodegeneration implies a progressive nature, while it looks like changes in enteric neurons in POI are transient and with no long-term functional outcomes.

Other groups, such as (PMID: 33414704 + PMID: 30259276 + PMID: 31622607 + PMID: 33871822), have been using the term to describe neuronal changes in the ENS, and it encompasses the alterations studies have shown for the CNS. While our model only induces transient changes, the mechanism we are investigating leads to death and disruption of neurons and their processes, which is, per definition, a type of neurodegeneration and, therefore, a proper use for this term.

"CX3CR1+/F11R+ macrophage numbers increased during POI, indicating possible inflammation-induced communication between enteric neurons and macrophages." - Based on the images provided there is no evidence that macrophages and neurons communicate. Did the authors mean "cell-cell interaction"? There is also a possibility that macrophages upregulate F11R expression upon IM.

Reviewer3 is correct; we have no data on direct communication between these two cell types, so we changed the wording.

"CX3CR1+/F11R+ macrophage numbers increased during POI, indicating possible inflammation-induced cell-cell interaction between enteric neurons and macrophages (Figure 4c)."

Yes, F11R can be upregulated by POI. According to Viola et al. (PMID: 37316669), F11R-positive macrophages are in contact with enteric neurons, so we chose this marker to better underline the possibility of neuron-macrophage communication in POI.

"while genes associated with synaptic transmission (e.g., Arc, Shank1, Nrnx1) were differentially regulated, hinting at a more engaged neuronal phase in the early disease state". - It is unclear what the authors are trying to say. This sentence needs to be revisited.

We agreed with reviewer3 and revised the wording.

"demonstrating functional changes in neuronal transmission in the inflamed ME." - The authors probably meant "predicting", not demonstrating. This sentence needs to be revisited.

We agreed with reviewer3 and revised the wording.

"our transcriptional data hint at a severe inflammatory and neuro-deteriorating phenotype". - "severe" and "neuro-deteriorating" should be removed.

We agreed with reviewer3 and modified the wording.

"neurotoxic actions in POI". - "neurotoxic" is a very specific term not relevant in this context.

Neurotoxicity is a unique form of toxicity that is defined as a biological, chemical, or physical agent able to damage the structure and function of neurons. Although we do not know the mechanism behind the neuronal death during POI, this process is likely caused by some type of neurotoxic agent acting on enteric neurons. Future studies should clarify this and hopefully find some therapeutic way to treat surgical-induced neuronal damage and counteract the responsible agents.

Genes highlighted under the category "neuroinflammation" are non-specific inflammatory genes. They should be listed as "inflammation".

As we discussed these inflammatory genes in the context of enteric neurons, we would like to keep the term "neuroinflammation" to highlight that this specific inflammation affects neurons. As there are many similar examples in CNS studies available, we would like to adapt them for the ENS.

Discussion:

It looks like neuronal damage by IM manipulation is transient and does not have any significant functional consequences in mice. Is it the case in patients who underwent

abdominal surgery complicated by POI?
The discussion should highlight the limitations of the study.

The significant functional consequence of neuronal damage is motility impairment, which, luckily for the patient, is, in most cases, a transient occurrence. So far, we do not have data on potential subsequent impairments or symptoms in POI patients. However, we currently collect an increasing number of different patient samples to gain an initial insight into the impact of abdominal surgery on patients at later stages. While POI hallmarks completely recovered after 21 days, we ultimately cannot assess whether mice are healthy again or if intestinal neurodegeneration has specific side effects that are not immediately apparent and only develop under certain conditions.

We added more information about the limitations of the study to the discussion as well.

“In summary, our data show that enteric neurons participate in the inflammatory cascade after surgical gut trauma and are driven toward a neurodegenerative program by resident macrophages. Certainly, our study is only a starting point in defining mechanisms behind neurodegeneration in the gut and comprehending the recovery of neuronal function after POI. Followup work has to identify specific pathways and markers in neurons and resident macrophages to get the complete story. However, counteracting these neurodegenerative processes could be a new strategy to protect patients from surgery-induced ENS complications, thereby positioning enteric neuroprotection as a valuable intervention strategy for treating inflammatory diseases of the gut.”

Other comments: in figure legends "enteric neurons (myenteric plexus)" could be replaced with "myenteric neurons."

We agreed with reviewer3 and modified the wording in the figure legends.

17th Dec 2024

Dear Reiner,

We are pleased to inform you that your manuscript is accepted for publication and is now being sent to our publisher to be included in the next available issue of EMBO Molecular Medicine.

Yours sincerely,
Jingyi

Jingyi Hou
Editor
EMBO Molecular Medicine
